# Trajectory Data Suffices for Statistically Efficient Learning in Offline RL with Linear $q^\pi$-Realizability and Concentrability

**Volodymyr Tkachuk**
University of Alberta, Edmonton, Canada

**Gellért Weisz**
Google DeepMind, London, UK

**Csaba Szepesvári**
Google DeepMind, Edmonton, Canada
University of Alberta, Edmonton, Canada

## Abstract

We consider offline reinforcement learning (RL) in $H$-horizon Markov decision processes (MDPs) under the linear $q^\pi$-realizability assumption, where the action-value function of every policy is linear with respect to a given $d$-dimensional feature function. The hope in this setting is that learning a good policy will be possible without requiring a sample size that scales with the number of states in the MDP. Foster et al. [2021] have shown this to be impossible even under *concentrability*, a data coverage assumption where a coefficient $C_{\text{conc}}$ bounds the extent to which the state-action distribution of any policy can veer off the data distribution. However, the data in this previous work was in the form of a sequence of individual transitions. This leaves open the question of whether the negative result mentioned could be overcome if the data was composed of sequences of full trajectories. In this work we answer this question positively by proving that with trajectory data, a dataset of size $\text{poly}(d, H, C_{\text{conc}})/\epsilon^2$ is sufficient for deriving an $\epsilon$-optimal policy, regardless of the size of the state space. The main tool that makes this result possible is due to Weisz et al. [2023], who demonstrate that linear MDPs can be used to approximate linearly $q^\pi$-realizable MDPs. The connection to trajectory data is that the linear MDP approximation relies on "skipping" over certain states. The associated estimation problems are thus easy when working with trajectory data, while they remain nontrivial when working with individual transitions. The question of computational efficiency under our assumptions remains open.

## 1 Introduction

We study the offline reinforcement learning (RL) setting, where the objective is to derive a near-optimal policy for an $H$-horizon Markov decision process (MDP) using *offline data*. This contrasts with the online RL paradigm, where learners interact directly with an MDP – or its simulator – to collect new data. Offline RL is especially relevant when acquiring new data guided by the learner is infeasible or ill-advised for safety reasons.

Deriving a near-optimal policy is only possible from offline data that covers the MDP well enough. One way to formalize this as an assumption, which we adopt for this work, is called *concentrability*. This assumption posits that the offline data sufficiently covers the distribution of state-action pairs that are accessible through running any policy. Challenges also arise in MDPs characterized by large or infinite state spaces. In such scenarios, an efficient learner's data requirements cannot scale

| | | Assumptions | | |
|---|---|---|---|---|
| Work | Task | Data | Structural | Result |
| [Xiong et al., 2022] | $\pi$-opt | $\lambda_{\min}$ lower bound | Linear MDP | ✓ |
| [Chen and Jiang, 2019] | $\pi$-opt | Conc | Bellman complete | ✓ |
| [Xie and Jiang, 2021] | $\pi$-opt | Strong Conc | $q^\pi$ | ✓ |
| **[This work]** | $\pi$-opt | Conc & **Traj data** | $q^\pi$ | ✓ |
| [Foster et al., 2021] | $\pi$-opt or $\pi$-eval | Conc | $q^\pi$ | x |
| [Wang et al., 2020] | $\pi$-eval | $\lambda_{\min}$ lower bound | $q^\pi$ | x |
| [Jia et al., 2024] | $\pi$-eval | Conc | Restricted $q^\pi$ | x |

Table 1: Notation is defined as: $\pi$-opt = policy optimization, $\pi$-eval = policy evaluation, Conc = Concentrability, $q^\pi$ = linear $q^\pi$-realizability, Traj = Trajectory, ✓ = $\mathrm{poly}(d, H, C_{\mathrm{conc}}, 1/\epsilon)$ sample complexity, x = exponential lower bound in terms of one of $d, H, C_{\mathrm{conc}}$.

with the state space size. An approach to remove state space dependence is to assume that the state-action value function of any policy can be linearly represented using a $d$-dimensional feature map, an assumption known as *linear $q^\pi$-realizability*.

While linear $q^\pi$-realizability facilitates efficient online RL [Weisz et al., 2023], its applicability has been limited in offline contexts. For instance, Foster et al. [2021] proves that no learning algorithm can derive an $\epsilon$-optimal policy under linear $q^\pi$-realizability and concentrability bounded by $C_{\mathrm{conc}}$, with a $\mathrm{poly}(d, H, C_{\mathrm{conc}}, \epsilon^{-1})$ number of samples. However, their result does not apply to *trajectory data*, where the offline data contains full sequences of state, action, and reward tuples obtained by following some policy from the initial state to the terminal state. The following problem is left open:

*"Does there exist an efficient learner that outputs an $\epsilon$-optimal policy, under the assumptions of linear $q^\pi$-realizability, concentrability, and trajectory data?"*

Our findings affirmatively answer this question in terms of sample complexity, highlighting a notable distinction in the requirements for trajectory data versus general offline data for effective learning. This underscores the practical value of accumulating trajectory data whenever feasible.

## 2 Related Works

In Table 1 we provide a comparison of our result to the other works in offline RL discussed below.

**Lower bounds:** As we have already discussed in Section 1, the work of Foster et al. [2021] shows a lower bound that depends on the size of the state space (in the same setting as ours), except they do not assume access to trajectory data. The work by Jia et al. [2024] is perhaps the most relevent to ours. They show an exponential lower bound in the horizon for policy evaluation, under the assumptions of trajectory data, concentrability, and a *restricted* linear $q^\pi$-realizability where the value function of only the target policy is linear. While we anticipate that evaluating policies (their focus) is no more difficult than optimizing policies (our focus), our $q^\pi$ realizability is for all memoryless policies (Assumption 1), while theirs is restricted to the target policy. Zanette [2021] shows an exponential lower bound in terms of the feature dimension $d$, under linear $q^\pi$-realizability, and various other structural assumptions; however, their setting would result in a concentrability coefficient larger than the size of the state space. Wang et al. [2020], Amortila et al. [2020] show a lower bound that is exponential in the horizon, under linear $q^\pi$-realizability. However, they use a $\lambda_{\min}$ *lower bound* condition, which requires a lower bound on the minimum eigenvalue $\lambda_{\min}$ of the expected covariance matrix used for least-squares estimation. This is seen as a weaker condition than ours, as it only posits good coverage in terms of the feature space, not the (possibly much richer) state-action space.

**Upper bounds:** Chen and Jiang [2019], Munos and Szepesvári [2008] present efficient algorithms under concentrability and *Bellman completeness*, an assumption that the Bellman optimality operator outputs a linearly realizable function when its input is linearly realizable. As linear $q^\pi$-realizability does not imply Bellman completeness [Zanette et al., 2020], these results do not transfer

to our setting. Xie and Jiang [2021] show an upper bound under linear $q^\pi$-realizability, albeit, using a stronger notion of data coverage than concentrability, which we call *strong concentrability*. The work of Xie et al. [2021, 2022] give data-dependent sample complexity bounds that hold under both Bellman completeness and linear $q^\pi$-realizability even in the absence of explicit data coverage assumptions. Jin et al. [2021] assume a general function approximation setting and also provide data-dependent bounds, while Duan et al. [2020], Xiong et al. [2022] show upper bounds for *linear MDPs* (a stronger assumption than linear $q^\pi$-realizability [Zanette et al., 2020]) with the $\lambda_{\min}$ lower bound condition.

## 3   Setting

Throughout we fix the integer $d \geq 1$. Let $\vec{0} \in \mathbb{R}^d$ be the $d$-dimensional, all zero vector. For $L > 0$, let $\mathcal{B}(L) = \{x \in \mathbb{R}^d : \|x\|_2 \leq L\}$ denote the $d$-dimensional Euclidean ball of radius $L$ centered at the origin, where $\|\cdot\|_2$ denotes the Euclidean norm. The inner product $\langle x, y \rangle$ for $x, y \in \mathbb{R}^d$ is defined as the dot product $x^\top y$. Let $\mathbb{1}\{B\}$ be the indicator function of a boolean-valued (possibly random) variable $B$, taking the value 1 if $B$ is true and 0 if false. Let $\mathcal{M}_1(X)$ denote the set of probability distributions over the set $X$. Let $\mathbb{E}_{B\sim\mathcal{P}}$ denote the expectation of random variable $B$ under distribution $\mathcal{P}$. For integers $i, j$, let $[i] = \{1, 2, \ldots, i\}$ and $[i : j] = \{i, \ldots, j\}$. For a symmetric matrix $M \in \mathbb{R}^{d \times d}$ we write $\lambda_{\min}(M)$ and $\lambda_{\max}(M)$ for its minimum and maximum eigenvalue.

The environment is modeled by a finite horizon Markov decision process (MDP). Fix the horizon to $H$. This MDP is defined by a tuple $(\mathcal{S}, \mathcal{A}, P, \mathcal{R})$. Here, the state space $\mathcal{S}$ is finite[1], and organized by stages: $\mathcal{S} = \bigcup_{h\in[H+1]} \mathcal{S}_h$, starting from a designated initial state $s_1$ ($\mathcal{S}_1 = \{s_1\}$)[2], and culminating in a designated terminal state $s_\top$ ($\mathcal{S}_{H+1} = \{s_\top\}$)[3]. Without loss of generality, we assume $\mathcal{S}_h$ and $\mathcal{S}_{h'}$ for $h \neq h'$ are disjoint sets. Define the function $\text{stage} : \mathcal{S} \to [H + 1]$, such that $\text{stage}(s) = h$ if $s \in \mathcal{S}_h$. The action space $\mathcal{A}$ is finite. The transition kernel is $P : (\bigcup_{h\in[H]} \mathcal{S}_h) \times \mathcal{A} \to \mathcal{M}_1(\mathcal{S})$, with the property that transitions occur between successive stages. Specifically, for any $h \in [H]$, state $s_h \in \mathcal{S}_h$, and action $a \in \mathcal{A}$, $P(s_h, a) \in \mathcal{M}_1(\mathcal{S}_{h+1})$. The reward kernel is $\mathcal{R} : \mathcal{S} \times \mathcal{A} \to \mathcal{M}_1([0, 1])$. So that the terminal state $s_\top$ has no influence on the learner we force the reward kernel to deterministically give zero reward for all actions $a \in \mathcal{A}$ in $s_\top$ (i.e. $\mathcal{R}(s_\top, a)(r) = \mathbb{1}\{0 = r\}$). An agent interacts with this environment sequentially across an episode of $H + 1$ stages, by selecting an action $a \in \mathcal{A}$ in the current state. The environment (except at stage $H + 1$) then transitions to a subsequent state according to $P$ and provides a reward in $[0, 1]$ as specified by $\mathcal{R}$[4].

We define an agent's interaction with the MDP through a *policy* $\pi$, which assigns a probability distribution over actions based on the history of interactions (including states, actions, and rewards). For this work, we restrict policies to be *memoryless*, that is, their action distribution depends solely on the most recent state in the history. The set of all memoryless policies is $\Pi = \{\pi : \pi : \mathcal{S} \to \mathcal{M}_1(\mathcal{A})\}$. For $\pi \in \Pi$, we write $\pi(a|s)$ to denote the probability $\pi(s)$ assigns to action $a$. For deterministic policies only (i.e., those that for each state place a unit probability mass on some action) we sometimes abuse notation by writing $\pi(s)$ to denote $\arg\max_{a\in\mathcal{A}} \pi(a|s)$. Starting from any state $s$ within the MDP and using a policy $\pi$ induces a probability distribution over trajectories, denoted as $\mathbb{P}_{\pi,s}$. For any $a \in \mathcal{A}$, $\mathbb{P}_{\pi,s,a}$ is the distribution over the trajectories when first action $a$ is used in state $s$, after which policy $\pi$ is followed. Specifically, for some $h \in [H + 1]$ and $(s, a) \in \mathcal{S}_h \times \mathcal{A}$, we write $\text{Traj} \sim \mathbb{P}_{\pi,s,a}$ to denote that $\text{Traj} = (S_h, A_h, R_h, \ldots, S_{H+1}, A_{H+1}, R_{H+1})$ for a random trajectory that follows the distribution specified by $\mathbb{P}_{\pi,s,a}$, that is, $S_h = s$, $A_h = a$, $A_i \sim \pi(S_i)$ for $i \in [h + 1 : H + 1]$, $S_{i+1} \sim P(S_i, A_i)$ for $i \in [h : H]$, and $R_i \sim \mathcal{R}(S_i, A_i)$ for $i \in [h : H + 1]$. Writing $\text{Traj} \sim \mathbb{P}_{\pi,s}$ has an analogous meaning, with the only difference being that $A_h$ is not fixed,

---

[1]The state space is assumed to be finite to simplify presentation. Our results extend to infinite state spaces.

[2]A deterministic start state $s_1$ is added for simplicity of presentation. It is easy to show that adding an additional stage to the MDP allows for the transition dynamics to encode an arbitrary start state distribution.

[3]A terminal state $s_\top$ is added purely as a technical convenience for the analysis. We will focus on the interaction of learners for stages $h \in [H]$ (not $[H + 1]$), since the terminal state will have no affect on the learner.

[4]Here, the reward and next-state are independent, given the current state and last action. Independence is nonessential and is assumed only to simplify the presentation.

and instead $A_h \sim \pi(S_h)$. For $h \in [H+1]$, we write $\mathbb{P}^h_{\pi,s}$ (and $\mathbb{P}^h_{\pi,s,a}$) for the marginal distribution of $(S_h, A_h)$ (i.e., the state-action pair of stage $h$) under the joint distribution of $\mathbb{P}_{\pi,s}$ (and $\mathbb{P}_{\pi,s,a}$).

For $1 \le t \le t' \le H+1$, we use the notation $x_{t:t'} = (x_u)_{u \in [t:t']}$ throughout, except when $(x_u)_{u \in [t:t']}$ are a sequence of scalar rewards. In that case, for convenience, we write $r_{t:t'} = \sum_{u=t}^{t'} r_u$ and $R_{t:t'} = \sum_{u=t}^{t'} R_u$. The state-value and action-value functions $v^\pi$ and $q^\pi$ are defined as the expected total reward along the rest of the trajectory while $\pi$ is used:

$$v^\pi(s) = \mathop{\mathbb{E}}_{\mathrm{Traj} \sim \mathbb{P}_{\pi,s}} R_{\mathrm{stage}(s):H} \text{ for } s \in \mathcal{S} \quad \text{and} \quad q^\pi(s,a) = \mathop{\mathbb{E}}_{\mathrm{Traj} \sim \mathbb{P}_{\pi,s,a}} R_{\mathrm{stage}(s):H} \text{ for } (s,a) \in \mathcal{S} \times \mathcal{A}.$$

Let $\pi^\star \in \Pi$ be an optimal policy, satisfying $q^{\pi^\star}(s,a) = \sup_{\pi \in \Pi} q^\pi(s,a)$ for all $(s,a) \in \mathcal{S} \times \mathcal{A}$. Let $q^\star(s,a) = q^{\pi^\star}(s,a)$ and $v^\star(s) = \max_{a \in \mathcal{A}} q^\star(s,a)$ for all $(s,a) \in \mathcal{S} \times \mathcal{A}$. By definition, we have

$$v^\star(s_\top) = v^\pi(s_\top) = 0 \quad \text{and} \quad q^\star(s_\top, a) = q^\pi(s_\top, a) = 0 \qquad \text{for all } \pi \in \Pi, a \in \mathcal{A}. \quad (1)$$

### 3.1 Assumptions and Problem Statement

A feature map is defined as $\phi : \mathcal{S} \times \mathcal{A} \to \mathcal{B}(L_1)$ for some $L_1 > 0$. The representative power of a feature map for an MDP is described by the following assumption:

**Assumption 1** $((\eta, L_2)$-Approximately Linear $q^\pi$-Realizable MDP)**.** *For some $\eta \ge 0, L_2 > 0$, assume that the MDP (together with a feature map $\phi$) is such that*

$$\sup_{\pi \in \Pi} \min_{\theta_h \in \mathcal{B}(L_2)} \max_{(s_h, a_h) \in \mathcal{S}_h \times \mathcal{A}} |q^\pi(s_h, a_h) - \langle \phi(s_h, a_h), \theta_h \rangle| \le \eta \qquad \text{for all } h \in [H+1].$$

*For any $h \in [H+1]$, let $\psi_h : \Pi \to \mathcal{B}(L_2)$ be a mapping from policies to parameter values $\theta_h$ that attain the min in the above display. For $h = H+1$, we restrict this mapping to $\psi_{H+1}(\cdot) = \vec{0}$, which satisfies the above display by definition. We write $\psi_{h:t}(\pi)$ for $(\psi_h(\pi), \ldots, \psi_t(\pi))$.*

We also make the following assumptions on the offline data (the relationship to non-trajectory data and the negative result by Foster et al. [2021] is discussed in Appendix B) :

**Assumption 2** (Full Length Trajectory Data)**.** *Assume the learner is given a dataset of full length trajectories[5] and corresponding features of size $n \ge 1$:*

$$\left(\mathrm{traj}^1, \ldots, \mathrm{traj}^n\right) \quad \text{and} \quad \left((\phi(s_h^1, a))_{h \in [H], a \in \mathcal{A}}, \ldots, (\phi(s_h^n, a))_{h \in [H], a \in \mathcal{A}}\right),$$

*where for some "data collection policy" $\pi^0 \in \Pi$ unknown to the learner, $(\mathrm{traj}^j)_{j=1}^n$ are independent samples from $\mathbb{P}_{\pi^0, s_1}$ where $\mathrm{traj}^j = (s_t^j, a_t^j, r_t^j)_{t \in [H+1]}$. To simplify notation we write*

$$\phi_h^j = \phi(s_h^j, a_h^j) \quad \text{for all } h \in [H], j \in [n].$$

**Definition 1** (Admissible Distribution)**.** *A sequence of $H$ state-action distributions $\nu = (\nu_h)_{h \in [H]} \in (\mathcal{M}_1(\mathcal{S}_h \times \mathcal{A}))^H$ is admissible for an MDP if there exists a policy $\pi \in \Pi$ such that*

$$\nu_h(s_h, a_h) = \mathbb{P}^h_{\pi, s_1}(s_h, a_h) \quad \text{for all } (s_h, a_h) \in \mathcal{S}_h \times \mathcal{A}, \ h \in [H].$$

Define the state-action occupancy measure of the data collection policy $\pi^0$ as $\mu = (\mu_h)_{h \in [H]}$ such that

$$\mu_h(s_h, a_h) = \mathbb{P}^h_{\pi^0, s_1}(s_h, a_h) \quad \text{for all } (s_h, a_h) \in \mathcal{S}_h \times \mathcal{A}, \ h \in [H].$$

**Assumption 3** (Concentrability)**.** *Assume there exists a constant $C_{\mathrm{conc}} \ge 1$, such that for all admissible distributions $\nu = (\nu_h)_{h \in [H]}$*

$$\max_{h \in [H]} \max_{(s_h, a_h) \in \mathcal{S}_h \times \mathcal{A}} \left\{ \frac{\nu_h(s_h, a_h)}{\mu_h(s_h, a_h)} \right\} \le C_{\mathrm{conc}}.$$

**Problem 1.** *Let $\epsilon > 0$. Under Assumptions 1 to 3, does there exist a learner, with access to only $n = \mathrm{poly}(1/\epsilon, H, d, C_{\mathrm{conc}}, \log(1/\delta), \log(1/L_1), \log(1/L_2))$ full length trajectories (as defined in Assumption 2), that outputs a policy $\pi$ such that, with probability at least $1 - \delta$,*

$$v^\star(s_1) - v^\pi(s_1) \le \epsilon?$$

---

[5]Our learner does not require explicit knowledge of the states within each trajectory; the features alone are sufficient.

## 4 Result

We resolve Problem 1 in the positive by defining a learner (Algorithm 1) that: selects parameters optimistically from modified MDPs that "skip over" certain states while preserving tight $q$-value estimation guarantees (achieved by solving Optimization Problem 1); then, outputs a greedy policy $\pi'$ (defined in line 3) over the selected parameters. This result is made formal in following theorem (proof in Section 5):

**Theorem 1.** *Let $\epsilon \in (0, H]$. Under Assumptions 1 to 3, if the number of full length trajectories $n = \tilde{\Theta}(C_{\mathrm{conc}}^4 H^7 d^4/\epsilon^2)$ and $\eta = \tilde{\mathcal{O}}\big(\epsilon^2/(C_{\mathrm{conc}}^2 H^5 d^2)\big)^6$, then, with probability at least $1 - \delta$, the policy $\pi'$ output by our learner (Algorithm 1) satisfies*

$$v^\star(s_1) - v^{\pi'}(s_1) \leq \epsilon,$$

where $\tilde{\Omega}, \tilde{\mathcal{O}}$ and $\tilde{\Theta}$ are the counterparts of $\Omega, \mathcal{O}$ and $\Theta$ from the big-Oh notation that hide polyloga-rithmic factors of the problem parameters $(1/\epsilon, 1/\delta, H, d, C_{\mathrm{conc}}, L_1, L_2)$. The following subsections focus on introducing the theory needed to formally present our learner, giving intuition behind our learner, and presenting our learner.

### 4.1 Background Theory

Our learner relies on the observation due to Weisz et al. [2023] that linearly $q^\pi$-realizable MDPs are linear MDPs, as long as they contain no low-range states. The *range* of a state is the largest possible regret from that state, that is, the largest difference in action-value that the choice of action in that state can make (up to misspecification):

$$\mathrm{range}(s) = \sup_{\pi \in \Pi} \max_{a, a' \in \mathcal{A}} \langle \phi(s, a, a'), \psi_{\mathrm{stage}(s)}(\pi) \rangle \quad \text{for all } s \in \mathcal{S}, \tag{2}$$

where $\phi(s, a, a') = \phi(s, a) - \phi(s, a')$ is a notation we use to denote feature differences. Intuitively, the choice of actions in low-range states are unimportant, as

$$|v^\pi(s) - q^\pi(s, a)| \leq \mathrm{range}(s) + 2\eta \quad \text{for any } \pi \in \Pi \text{ and all } (s, a) \in \mathcal{S} \times \mathcal{A}. \tag{3}$$

As a warm-up, consider the example MDPs shown in Fig. 1. We will transform the linearly $q^\pi$-realizable MDP on the left into a linear MDP on the right by "skipping over" the red low-range states. Let the features for both MDPs be $\phi(s_1, \cdot) = (1), \phi(s_3, \cdot) = (0.5), \phi(\cdot, \cdot) = (0)$ otherwise. Then the left MDP is $(0, 1)$-approximately $q^\pi$-realizable, with realizability parameter $\psi_h(\pi) = (1)$ for all $h \in [H+1], \pi \in \Pi$. However, it is not a linear MDP, since the rewards cannot be represented by a linear function of the features. To see this, notice that there exists no $\theta$ such that $\langle \phi(s_1, a_1), \theta \rangle = r(s_1, a_1) = 1$ and $\langle \phi(s_1, a_2), \theta \rangle = r(s_1, a_2) = 0.5$, since $\phi(s_1, a_1) = \phi(s_1, a_2) = (1)$. We modify this MDP on the left to "skip over" low-range red states, by automatically taking the first available action at such states, and summing up the rewards along skipped paths. This turns the MDP into the one on the right of Fig. 1, which is a linear MDP.

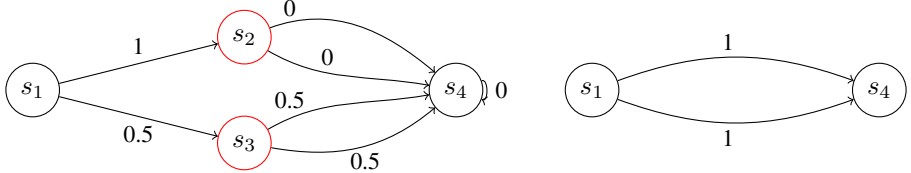

Figure 1: The features for both MDPs are $\phi(s_1, \cdot) = (1), \phi(s_3, \cdot) = (0.5), \phi(\cdot, \cdot) = (0)$ otherwise. **Left:** A $(0, 1)$-approximately $q^\pi$-realizable MDP. **Right:** Linear MDP, obtained by skipping low range (red) states in the left MDP. Source: Figure 1 from [Weisz et al., 2023].

The key fact about linear MDPs that we will use is that for any function $f : \mathcal{S} \to [0, H]$ (e.g., $v$-value approximators), and any $h \in [H]$, there is some parameter $\theta_h \in \mathbb{R}^d$ so that for *any* $(s, a) \in \mathcal{S}_h \times \mathcal{A}$, $\langle \phi(s, a), \theta_h \rangle$ gives the expectation of the reward plus $f$'s value on the next state. In our modified

---

$^6$The bound on $\eta$ is assumed for clarity of presentation, to avoid presenting two error terms in the final bound.

MDP this result transfers to the fact that the expected sum of rewards along a skipped path, plus $f$'s value on the next state after the skipped path, is linearly realizable. Before making this result formal in Lemma 4.2, we clarify the skipping behavior.

First, we address the fact that we need an approximate, parametric bound on range$(\cdot)$ with a parameter count that is independent of $|\mathcal{S}|$. For $h \in [2 : H]$, let $\Psi_h = \{\psi_h(\pi) : \pi \in \Pi\} \subseteq \mathcal{B}(L_2)$ be the (compact) set of parameter values corresponding to all policies. For all $h \in [2 : H]$, fix a subset $\bar{G}_h \subset \Psi_h$ of size $|\bar{G}_h| = d_0 := \lceil 4d \log \log(d) + 16 \rceil$ that is the basis of a near-optimal design for $\Psi_h$ (more precisely, satisfying Definition 3). The existence of such a near-optimal design follows from [Todd, 2016, Part (ii) of Lemma 3.9]. Let $\bar{G} = \bar{G}_{2:H}$, which we call the *true guess*. Now notice that $\bar{G} \in \mathbf{G}$ where

$$\mathbf{G} = (\mathcal{B}(L_2))^{[2:H] \times [d_0]}. \tag{4}$$

For $G \in \mathbf{G}$ we will use the notation that $G = G_{2:H}$, where $G_h = (\vartheta_h^i)_{i \in [d_0]} \in \mathcal{B}(L_2)^{d_0}$. Any $G = G_{2:H} \in \mathbf{G}$ can be used to define an approximate, low parameter-count "version" of range that is completely specified by $\tilde{O}(Hd^2)$ parameters:

$$\text{range}^G(s) = \max_{\vartheta \in G_h} \max_{a,a' \in \mathcal{A}} \langle \phi(s, a, a'), \vartheta \rangle \qquad \text{for all } h \in [2 : H], s \in \mathcal{S}_h. \tag{5}$$

As shown in Proposition 4.5 of [Weisz et al., 2023], range$^{\bar{G}}$ can be used to bound the true range:

**Lemma 4.1.** *For all $h \in [2 : H]$ and $s \in \mathcal{S}_h$, $\text{range}(s) \leq \sqrt{2d} \cdot \text{range}^{\bar{G}}(s)$.*

Based on any $G \in \mathbf{G}$, we are interested in simulating a modified MDP that "skips over" states $s$ that have a low range$^G(s)$, by taking an action according to $\pi^0$, and presenting as the reward the summed up rewards along paths consisting of skipped states. This "modified MDP" only serves as intuition, and will not be formally defined or used in our formal arguments. Instead, we define the "skipping probability" parameter at state $s \in \mathcal{S}$, with $\alpha > 0$ (set later in Eq. (35)), as

$$\omega_G(s) = \begin{cases} 1 & \text{if } s \notin \mathcal{S}_1 \cup \mathcal{S}_{H+1} \text{ and } \text{range}^G(s) \leq \alpha/\sqrt{2d} \\ 2 - \sqrt{2d} \cdot \text{range}^G(s)/\alpha & \text{if } s \notin \mathcal{S}_1 \cup \mathcal{S}_{H+1} \text{ and } \alpha/\sqrt{2d} \leq \text{range}^G(s) \leq 2\alpha/\sqrt{2d} \\ 0 & \text{otherwise.} \end{cases} \tag{6}$$

The skipping behavior is probabilistic[7]: it never skips for stages 1 and $H + 1$ (where range$^G$ is not defined); it always skips for ranges lower than some threshold, never skips for ranges higher than twice this threshold, and linearly interpolates between the two in between the thresholds. For $h \in [H]$, and $1 \leq l \leq h$, let $\text{traj} = (s_t, a_t, r_t)_{l \leq t \leq H+1}$ be any fixed trajectory that starts from some stage $l$. Let $\tau \sim F_{G,\text{traj},h+1} \in \mathcal{M}_1([h+1 : H+1])$ be the random stopping stage, when starting from state $s_h$ and skipping subsequent states with probability $\omega_G(\cdot)$. Formally, for $t \in [h + 1 : H + 1]$ let $F_{G,\text{traj},h+1}(\tau = t) = (1 - \omega_G(s_t)) \prod_{u=h+1}^{t-1} \omega_G(s_u)$. We will often write $F_{G,h+1}^j$ to denote $F_{G,\text{traj}^j,h+1}$ where $\text{traj}^j = (s_t^j, a_t^j, r_t^j)_{t \in [H+1]}, j \in [n]$.

Next, we present a key tool derived from results of Weisz et al. [2023]: as long as the skips are informed by the true guess, the resulting MDP is approximately linear (proof in Appendix C):

**Lemma 4.2** (Approximate Linear MDP under the true guess). *Let $\eta \geq 0, L_2 > 0$. Let $M$ be an $(\eta, L_2)$-approximately linear $q^\pi$-realizable MDP (Assumption 1) with corresponding feature map $\phi$. Let $\tilde{L}_2 = L_2(8H^2 d_0/\alpha + 1)$. Then, for each $f : \mathcal{S} \to [0, H]$ with $f(s_\top) = 0$, policy $\pi \in \Pi$, and stage $h \in [H]$, there exists a parameter $\rho_h^\pi(f) \in \mathcal{B}(\tilde{L}_2)$ such that for all $(s, a) \in \mathcal{S}_h \times \mathcal{A}$,*

$$\left| \mathop{\mathbb{E}}_{\text{Traj} \sim \mathbb{P}_{\pi,s,a}} \mathop{\mathbb{E}}_{\tau \sim F_{\bar{G},\text{Traj},h+1}} [R_{h:\tau-1} + f(S_\tau)] - \langle \phi(s, a), \rho_h^\pi(f) \rangle \right| \leq \tilde{\eta},$$

*where $\tilde{\eta} = \eta(10H^2 d_0/\alpha + 1)$.*

## 4.2 The Benefit of Trajectory Data

Our learner will heavily rely on the result presented in Lemma 4.2. We will need to learn good estimates of the parameters $\rho_h^{\pi^0}(f)$, for any $f : \mathcal{S} \to [0, H], h \in [H]$. However, to estimate a $\rho_h^{\pi^0}(f)$

---

[7]The resulting smoothness of skipping behavior is beneficial for a later technical covering argument (Eq. (90) in Lemma I.4).

parameter well we will require least-squares targets that have bounded noise and expectation equal to $\langle \phi(s,a), \rho_h^\pi(f) \rangle$ for all $(s,a) \in \mathcal{S}_h \times \mathcal{A}$. Full trajectory data (Assumption 2) makes this possible. Each full length trajectory $\text{traj}^j = (s_t^j, a_t^j, r_t^j)_{t \in [H+1]} j \in [n]$ can be used to create the following least-squares target (which has the desired properties):

$$\mathbb{E}_{\tau \sim F_{\bar{G}, h+1}^j} \left[ r_{h:\tau-1}^j + f\left(s_\tau^j\right) \right].$$

Importantly, it is because we have full length trajectories that we can transform the data available to simulate arbitrary length skipping mechanisms.

## 4.3 Intuition Behind our Learner

Next, we describe the high-level intuition and ideas behind our learner. Consider the "modified" MDP where low-range states are skipped. As the learner has access to trajectory data (Assumption 2), it can transform this data accordingly to simulate trajectories from the modified MDP. Any near-optimal policy for the modified MDP is also near-optimal for the original MDP (due to Eq. (3)). Thus, our previous linear realizability property (Lemma 4.2) allows for an offline RL version of the algorithm ELEANOR [Zanette et al., 2020] to statistically efficiently derive a near-optimal policy for the modified MDP. Indeed, the optimization problem underlying ELEANOR serves as a starting point for Optimization Problem 1, which is at the heart of our learner.

The challenge is that the true guess $\bar{G}$ that Lemma 4.2 relies upon is not known to the learner. This means that the learner is not given any explicit information of what states to "skip over". To overcome this, we design a learner to output the policy $\pi'$ (defined in Algorithm 1) based on Optimization Problem 1, where the optimization problem considers all guesses for the possible values of $\bar{G}$. For each $G \in \mathbf{G}$, it considers the MDP that skips over low-range states when the range is calculated according to $G$. It then calculates sets $\Theta_{G,h}$ for each stage $h$, that are guaranteed (with high probability) to include the parameter $\psi_h(\pi_G^\star)$ realizing $q^{\pi_G^\star}$ (where $\pi_G^\star$, defined in Eq. (15), is the optimal policy in the MDP with skipping based on $G$). We achieve this by defining $\Theta_{G,h}$ backwards for $h = H, H-1, \ldots, 1$. By induction, if $\Theta_{G,h+1}, \ldots, \Theta_{G,H}$ all contain the desired parameter for their stage, then *some* parameter sequence in the Cartesian product $\Theta_{G,h+1} \times \cdots \times \Theta_{G,H}$ allows us to near-perfectly (up to some misspecification error) compute $q^{\pi_G^\star}$-values of stages $> h$. Therefore, the least-squares parameter based on this sequence will be near the true parameter for stage $h$. Defining $\hat{\Theta}_{G,h}$ to be all least-squares predictors for sequences in the aforementioned Cartesian product, and $\Theta_{G,h}$ to be unions of the confidence ellipsoids around these predictors ensures the true parameter $\psi_h(\pi_G^\star)$ realizing $q^{\pi_G^\star}$ for stage $h$ is included in $\Theta_{G,h}$. This argument is made precise in Lemma D.2.

There are two problems remaining. One is that some values of $G$ considered by Optimization Problem 1 lead to skipping over important large-value states, degrading the performance of the best policy $\pi_G^\star$ available under that skipping. The other problem is that at the expense of making sure the true parameters are included in the sets $\Theta_{G,h}$, these sets might become large, in the sense of containing parameters that lead to very different predictions. Avoiding the first problem would make $v^{\pi_G^\star}(s_1)$ nearly as large as $v^\star(s_1)$. Avoiding the second problem would lead to tight $q$-value estimators, and therefore to $v^{\pi_G'}(s_1)$ being nearly as large as $v^{\pi_G^\star}(s_1)$, for a policy $\pi_G'$ that is greedy with respect to our hypothetically tight $q$-value estimator. A key idea is to reject from consideration any $G \in \mathbf{G}$ that leads to $q$-estimations that are not sufficiently tight (Eq. (14)). The reason we can do this is because for $G = \bar{G}$ we can show that this condition passes (with high probability), and therefore we do not reject $\bar{G}$ (precise statement in Lemma D.3). We can show this since we have trajectory data (Assumption 2), allowing us to use least-squares targets of the form used in Eq. (12), which we know are linearly realizable when $G = \bar{G}$ (discussed in Section 4.2). Finally, we resolve the first problem by selecting among these tight estimators the one that guarantees the highest policy value from $s_1$, which can be no worse than the value guaranteed by the choice of $G = \bar{G}$, which itself can be seen to be close to $v^\star(s_1)$.

## 4.4 Learner

Next, we formally introduce our learner, at the heart of which lies Optimization Problem 1. We define various $q$ and $v$-value estimators that we use. For $x \in \mathbb{R}$, let $\text{clip}_{[0,H]} x = \max\{0, \min\{H, x\}\}$.

Then, for $h \in [H], s \in \cup_{t \in [h:H+1]} \mathcal{S}_t, a \in \mathcal{A}, \theta \in \mathbb{R}^d, \theta_{h:H+1} = (\theta_h, \ldots, \theta_{H+1}) \in \mathbb{R}^{d(H-h+2)}$, let

$$q_\theta(s,a) = \langle \phi(s,a), \theta \rangle, \quad q_{\theta_{h:H+1}}(s,a) = q_{\theta_{\text{stage}(s)}}(s,a), \tag{7}$$

$$v_\theta(s) = \max_{a \in \mathcal{A}} q_\theta(s,a), \quad v_{\theta_{h:H+1}}(s) = \max_{a \in \mathcal{A}} q_{\theta_{h:H+1}}(s,a), \tag{8}$$

$$\bar{q}_\theta(s,a) = \text{clip}_{[0,H]} q_\theta(s,a), \quad \bar{q}_{\theta_{h:H+1}}(s,a) = \bar{q}_{\theta_{\text{stage}(s)}}(s,a), \tag{9}$$

$$\bar{v}_\theta(s) = \text{clip}_{[0,H]} v_\theta(s), \quad \bar{v}_{\theta_{h:H+1}}(s) = \bar{v}_{\theta_{\text{stage}(s)}}(s). \tag{10}$$

**Optimization Problem 1.**

$$\arg\max_{G \in \mathbf{G}, \theta^\dagger_{1:H+1} \in \Theta_{G,1} \times \cdots \times \Theta_{G,H+1}} \bar{v}_{\theta^\dagger_1}(s_1) \qquad \text{subject to, for all } h \in [H]$$

$$X_h = \lambda I + \sum_{j \in [n]} \phi_h^j (\phi_h^j)^\top, \tag{11}$$

$$\hat{\Theta}_{G,h} = \left\{ X_h^{-1} \sum_{j \in [n]} \phi_h^j \mathbb{E}_{\tau \sim F_{G,h+1}^j} \left[ r_{h:\tau-1}^j + \bar{v}_{\theta_{h+1:H+1}}(s_\tau^j) \right] : \theta_{h+1:H+1} \in \underset{u=h+1}{\overset{H+1}{\times}} \Theta_{G,u} \right\}, \tag{12}$$

$$\Theta_{G,h} = \left\{ \theta_h \in \mathcal{B}(\tilde{L}_2) : \min_{\hat{\theta}_h \in \hat{\Theta}_{G,h}} \|\theta_h - \hat{\theta}_h\|_{X_h} \le \beta \right\}, \quad \Theta_{G,H+1} = \{\vec{0}\} \quad \beta \text{ defn Eq. (32)}, \tag{13}$$

$$\frac{1}{n} \sum_{j \in [n]} \left( \max_{\theta \in \Theta_{G,h}} \bar{q}_\theta(s_h^j, a_h^j) - \min_{\theta \in \Theta_{G,h}} \bar{q}_\theta(s_h^j, a_h^j) \right) \le \bar{\epsilon}, \qquad \bar{\epsilon} \text{ defn Eq. (29)}. \tag{14}$$

Let $(G', \theta'_{1:H+1})$ denote the solution to Optimization Problem 1. Notice that unlike ELEANOR, we optimize over all $G \in \mathbf{G}$, which can be seen as an optimization over all possible "modified MDPs" with different skipping mechanisms. Another observation is that apart from $h = 1$, the choice of $\theta'_h$ from $\Theta_{G',h}$ made by the optimization is arbitrary. Indeed, unlike ELEANOR, which chooses globally optimistic least-squares predictors for each stage, our optimization does not need to care about (or optimize for) the specific choice of $q$-value predictors from their respective confidence sets $\Theta_{G,h}$. We can be agnostic to the choice of $q$-value predictors because all choices lead to similar predictions due to Eq. (14), as will be shown formally, in Lemma 5.1. However, fixing an arbitrary concrete choice in the optimization allows us to define an output policy $\pi'$ that is parametrized only by these vectors, making both the memory and computational requirements of representing and executing $\pi'$ small.

Our learner (Algorithm 1) solves Optimization Problem 1 and uses $\theta'_{1:H+1}$ to output a greedy policy.

---

**Algorithm 1:** Learner

1: **input:** accuracy $\epsilon > 0$, failure probability $\delta > 0$, concentrability coefficient $C_{\text{conc}} < \infty$, trajectories $(\text{traj}^1, \ldots, \text{traj}^n)$, features $\left( (\phi(s_h^1, a))_{h \in [H], a \in \mathcal{A}}, \ldots, (\phi(s_h^n, a))_{h \in [H], a \in \mathcal{A}} \right)$, norm bounds $L_1, L_2$.

2: $G', \theta'_{1:H+1} \leftarrow$ solution to Optimization Problem 1

3: $\pi'(a|s) \leftarrow \mathbb{1}\left\{ a = \arg\max_{a' \in \mathcal{A}} \bar{q}_{\theta'_{\text{stage}(s)}}(s, a') \right\}$ for all $(s,a) \in \mathcal{S} \times \mathcal{A}$

4: **return** $\pi'$

---

## 5 Proof of Theorem 1

Before giving the proof, we formally define the optimal policy in the modified MDP that skips according to $\text{range}^G$, for any $G \in \mathbf{G}$ as

$$\pi_G^\star(a|s) = \pi^0(a|s)\omega_G(s) + \mathbb{1}\left\{ a = \arg\max_{a' \in \mathcal{A}} q^{\pi_G^\star}(s,a') \right\}(1 - \omega_G(s)). \tag{15}$$

Notice that for states $s \in \mathcal{S}_h$ for some $h \in [H]$, $\pi_G^\star(\cdot|s)$ in the above definition depends on the value of $q^{\pi_G^\star}(s', \cdot)$ for some $s' \in \mathcal{S}_{h+1}$. We can therefore interpret the above recursive definition as defining $\pi_G^\star(\cdot|s)$ for $s \in \mathcal{S}_h$, first for $h = H+1$, then $h = H$, etc., down to $h = 1$. Every time we define the policy for some stage $h$ in such a way, the policy and $q^{\pi_G^\star}(s', \cdot)$ are already defined on later stages, making the definition valid, and resolving the recursive nature of Eq. (15).

*Proof.* $v^\star(s_1) - v^{\pi'}(s_1)$ can be decomposed into the following error terms.

$$v^\star(s_1) - v^{\pi'}(s_1) = \underbrace{v^\star(s_1) - v^{\pi^\star_{\bar{G}}}(s_1)}_{(I)} + \underbrace{v^{\pi^\star_{\bar{G}}}(s_1) - v_{\theta'_1}(s_1)}_{(II)} + \underbrace{v_{\theta'_1}(s_1) - v^{\pi'}(s_1)}_{(III)}.$$

The remainder of the proof focuses on bounding these error terms. Following the intuition described in Section 4, showing that terms (I) and (II) are small can be seen as addressing the first problem of potentially skipping over large-value states, while showing that term (III) is small can be seen as addressing the second problem of $\pi'$ being greedy w.r.t. to a potentially inaccurate estimates $\theta'_{1:H+1}$.

**Bounding** $(I) = v^\star(s_1) - v^{\pi^\star_{\bar{G}}}(s_1)$: This term cannot be too large since the range$^{\bar{G}}$ function is approximately correct (Lemma 4.1), and we only skip over states with low range$^{\bar{G}}$ (Eq. (6)), implying the action we take doesn't affect the value function much (Eq. (3)). In Appendix D.1 we formalize this intuition, and show the following result.

$$(I) = v^\star(s_1) - v^{\pi^\star_{\bar{G}}}(s_1) \leq H(2\alpha + 2\eta). \tag{16}$$

**Bounding** $(II) = v^{\pi^\star_{\bar{G}}}(s_1) - \bar{v}_{\theta'_1}(s_1)$: This term can be bounded by approximately zero due to Optimization Problem 1 being optimistic from the start state. First, note that $v^{\pi^\star_{\bar{G}}}$ is approximately equal to $\bar{v}_{\psi_1(\pi^\star_{\bar{G}})}$ (Assumption 1). Then, in Lemma D.2 we show that $\psi_1(\pi^\star_{\bar{G}}) \in \Theta_{\bar{G},1}$, and in Lemma D.3 we show that $\bar{G}$ is a feasible solution to Optimization Problem 1. Since $(G', \theta'_{1:H+1})$ is the solution to Optimization Problem 1, it holds that $\bar{v}_{\theta'_1}(s_1) \geq \bar{v}_\theta(s_1)$ for any $\theta \in \Theta_{G,1}$ where $G$ is a feasible solution to Optimization Problem 1. Thus $\bar{v}_{\theta'_1}(s_1) \geq v_{\psi_1(\pi^\star_{\bar{G}})}(s_1)$. In Appendix D.2 we formalize this intuition, and show that with probability at least $1 - \delta$,

$$(II) = v^{\pi^\star_{\bar{G}}}(s_1) - \bar{v}_{\theta'_1}(s_1) \leq \eta. \tag{17}$$

**Bounding** $(III) = \bar{v}_{\theta'_1}(s_1) - v^{\pi'}(s_1)$: To bound term (III) we will first show in Lemma 5.1 that value estimates in terms of $\theta'_h$ and $\psi_h(\pi^\star_{G'})$ are close with high probability for all $h \in [H+1]$. This lemma allows us to relate $\bar{v}_{\theta'_1}$ to $\bar{v}_{\psi_1(\pi^\star_{G'})}$ and then Assumption 1 relates $\bar{v}_{\psi_1(\pi^\star_{G'})}$ to $v^{\pi^\star_{G'}}$. We are then left with relating $v^{\pi^\star_{G'}}$ to $v^{\pi'}$. To do this we claim that $\pi'$ is an approximate policy improvement step w.r.t. $v^{\pi^\star_{G'}}$, which can be seen by recalling that $\pi'$ is greedy w.r.t. $\bar{q}_{\theta'_{1:H+1}}$, and as we mentioned a couple sentences ago, $\bar{v}_{\theta'_h}$ and $\bar{v}_{\psi_h(\pi^\star_{G'})}$ are close for all $h \in [H+1]$.

To formalize this intuition we begin by decomposing $\bar{v}_{\theta'_1}(s_1) - v^{\pi'}(s_1)$ into the following error terms

$$\bar{v}_{\theta'_1}(s_1) - v^{\pi'}(s_1) = \bar{v}_{\theta'_1}(s_1) - \bar{q}_{\psi_1(\pi^\star_{G'})}(s_1, \pi'(s_1)) + \bar{q}_{\psi_1(\pi^\star_{G'})}(s_1, \pi'(s_1)) - v^{\pi'}(s_1). \tag{18}$$

To bound $\bar{v}_{\theta'_1}(s_1) - \bar{q}_{\psi_1(\pi^\star_{G'})}(s_1, \pi'(s_1))$ we introduce a useful lemma (proof in Appendix F).

**Lemma 5.1.** *There is an event $\mathcal{E}_2$, that occurs with probability at least $1 - \delta/3$, such that under event $\mathcal{E}_2$, for all $G \in \mathbf{G}$ that are feasible solutions to Optimization Problem 1, for all $h \in [H]$, for all $(\theta_{s,a})_{(s,a) \in \mathcal{S}_h \times \mathcal{A}}$ and $(\check{\theta}_{s,a})_{(s,a) \in \mathcal{S}_h \times \mathcal{A}} \in \Theta_{G,h}^{\mathcal{S}_h \times \mathcal{A}}$, and for all admissible distributions $\nu = (\nu_t)_{t \in [H]}$, it holds that*

$$\mathbb{E}_{(S,A) \sim \nu_h}\left[\bar{q}_{\theta_{S,A}}(S, A) - \bar{q}_{\check{\theta}_{S,A}}(S, A)\right] \leq \tilde{\epsilon} \qquad \tilde{\epsilon} \text{ defn Eq. (30)}.$$

To use Lemma 5.1 we must show that $\bar{v}_{\theta'_1}(s_1) - \bar{q}_{\psi_1(\pi^\star_{G'})}(s_1, \pi'(s_1))$ satisfies its requirements. First, note that $\bar{v}_{\theta'_1}(s_1) = \bar{q}_{\theta'_1}(s_1, \pi'(s_1))$, by definition of $\pi'$ (line 3). Second, $G'$ is the solution to Optimization Problem 1, thus, a feasible solution. Third, by Lemma D.2, there is an event $\mathcal{E}_1$, which occurs with probability at least $1 - \delta/3$, such that under event $\mathcal{E}_1$, $\psi_1(\pi^\star_{G'}) \in \Theta_{G',1}$, and by definition $\theta'_1 \in \Theta_{G',1}$. Let $\nu_h(s, a) = \mathbb{P}^h_{\pi', s_1}(s, a)$ for all $h \in [H], (s, a) \in \mathcal{S}_h \times \mathcal{A}$. Clearly $\nu = (\nu_h)_{h \in [H]}$ is an admissible distribution by Definition 1. Thus, under event $\mathcal{E}_1 \cap \mathcal{E}_2$, by Lemma 5.1,

$$\bar{v}_{\theta'_1}(s_1) - \bar{q}_{\psi_1(\pi^\star_{G'})}(s_1, \pi'(s_1)) = \mathbb{E}_{(S,A) \sim \nu_1}\left[\bar{q}_{\theta'_1}(S, A) - \bar{q}_{\psi_1(\pi^\star_{G'})}(S, A)\right] \leq \tilde{\epsilon}.$$

It is left to bound $\bar{q}_{\psi_1(\pi^\star_{G'})}(s_1, \pi'(s_1)) - v^{\pi'}(s_1)$ in Eq. (18). To do this, first note that

$$\bar{q}_{\psi_1(\pi^\star_{G'})}(s_1, \pi'(s_1)) - v^{\pi'}(s_1) \leq q^{\pi^\star_{G'}}(s_1, \pi'(s_1)) - v^{\pi'}(s_1) + \eta\,.$$

where the inequality holds since we have approximate linear $q^\pi$-realizability (Assumption 1). To bound $q^{\pi^\star_{G'}}(s_1, \pi'(s_1)) - v^{\pi'}(s_1)$ notice that $v^{\pi'}(s_1) = q^{\pi'}(s_1, \pi'(s_1))$, which implies that

$$q^{\pi^\star_{G'}}(s_1, \pi'(s_1)) - v^{\pi'}(s_1) = q^{\pi^\star_{G'}}(s_1, \pi'(s_1)) - q^{\pi'}(s_1, \pi'(s_1)) = \mathop{\mathbb{E}}_{\mathrm{Traj}\sim\mathbb{P}_{\pi',s_1}}\left[v^{\pi^\star_{G'}}(S_2) - v^{\pi'}(S_2)\right]\,.$$

Next, we give a bound on $\mathbb{E}_{\mathrm{Traj}\sim\mathbb{P}_{\pi',s_1}}\left[v^{\pi^\star_{G'}}(S_2) - v^{\pi'}(S_2)\right]$ (proof in Appendix D.3):

**Lemma 5.2.** *Under event $\mathcal{E}_1 \cap \mathcal{E}_2$, for any $h \in [2 : H+1]$, it holds that*

$$\mathop{\mathbb{E}}_{\mathrm{Traj}\sim\mathbb{P}_{\pi',s_1}}\left[v^{\pi^\star_{G'}}(S_h) - v^{\pi'}(S_h)\right] \leq 2(H - h + 2)(\eta + \tilde{\epsilon})\,.$$

Intuitively, the above lemma holds since $\pi'$ can be thought of as an approximate policy improvement step w.r.t. $v^{\pi^\star_{G'}}$. To see this, recall that $\pi'$ is greedy w.r.t. $\bar{q}_{\theta'_{1:H+1}}$ (line 3). Then, with Lemma 5.1, we can show $\bar{v}_{\theta'_h}$ and $\bar{v}_{\psi_h(\pi^\star_{G'})}$(which is close to $v^{\pi^\star_{G'}}$ (Assumption 1)) are close for all $h \in [H+1]$. The above bounds imply that under event $\mathcal{E}_1 \cap \mathcal{E}_2$, which occurs with probability at least $1 - 2\delta/3$,

$$(\mathrm{III}) = \bar{v}_{\theta'_1}(s_1) - v^{\pi'}(s_1) \leq 2H(\eta + \tilde{\epsilon}) + \tilde{\epsilon} + \eta\,. \tag{19}$$

**Combining the Bounds:** To finish the proof we combine the bounds on all three terms (Eqs. (16), (17) and (19)), to get that under event $\mathcal{E}_1 \cap \mathcal{E}_2 \cap \mathcal{E}_3$, which occurs with probability at least $1 - \delta$,

$$v^\star(s_1) - v^{\pi'}(s_1) \leq H(2\alpha + 2\eta) + \eta + 2H(\eta + \tilde{\epsilon}) + \tilde{\epsilon} + \eta \leq 4(H+1)(\alpha + \eta + \tilde{\epsilon})\,.$$

To bound the above display by $\epsilon$ we set $\alpha = \epsilon/(12(H+1)) < 1$. If $n = \tilde{\Theta}\big(C_{\mathrm{conc}}^4 H^7 d^4/\epsilon^2\big)$ and $\eta = \tilde{\mathcal{O}}\big(\alpha/\sqrt{nH}\big)$ (Eq. (26)), we show that $\tilde{\epsilon} = \tilde{\mathcal{O}}\big(C_{\mathrm{conc}}^2 H^{5/2} d^2/\sqrt{n}\big)$ (Eq. (44)). This implies that

$$4(H+1)(\alpha + \eta + \tilde{\epsilon}) \leq \epsilon\,. \qquad \square$$

# 6   Limitations and Conclusions

In this work we resolved an open problem in the positive, by presenting the first statistically efficient learner (Section 4.4) that outputs a near optimal policy in the offline RL setting with approximate linear $q^\pi$-realizability (Assumption 1), trajectory data (Assumption 2), and concentrability (Assumption 3). One limitation of this work is that we are not aware of any computationally efficient implementation of Optimization Problem 1, which is at the heart of our learner. As such, it is left as an open problem whether computationally efficient learning is possible in the setting we considered. Another limitation is that we are not sure if our statistical rate in Theorem 1 is optimal. Showing a matching lower bound or improving the rate is left for future work.

Another limitation of our work originates from our setting underpinning our result (Section 4), namely the three assumptions: approximate linear $q^\pi$-realizability, trajectory data, and concentrability. Approximate linear $q^\pi$-realizability requires the value function of all memoryless policies to be linear in a fixed and known $d$-dimensional feature map. While strictly weaker than the linear MDP assumption [Zanette et al., 2020], this assumption is still strong. Trajectory data requires full sequences of interactions with an environment to be collected by a single policy. For long horizon problems this can be practically challenging. Concentrability requires the state and action spaces to be well-covered. This can be challenging to guarantee since often the state and action spaces are unknown at the time of data collection. Further, since we require the trajectory data to be collected by a single policy, it may be the case that no single policy exists that covers the state and action spaces well, and a mixture of policies must be considered, which our current result does not immediately hold for. Although the assumptions appear strong, a justification for them is that under many variations of weaker assumptions (for instance: general data, or linear $q^\pi$-realizability of only one policy, or only coverage of the feature space), polynomial statistical rates have been shown to be impossible to achieve by any learner (Table 1).

Since this work is focused on foundational theoretical research it is unlikely to have any direct and immediate societal impacts.

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

# Appendix

## A Parameter Settings and Notation

$$n = \tilde{\Theta}\left(\frac{C_{\text{conc}}^4 H^7 d^4}{\epsilon^2}\right) \qquad\qquad \text{Set at the end of Section 5}$$

$$\tag{20}$$

$$d_0 = \lceil 4d \log \log(d) + 16 \rceil \qquad\qquad \text{Defined above Eq. (4)} \qquad (21)$$

$$L_1 = \text{Upper bound on 2-norm of features } \phi \qquad\qquad \text{Defined above Assumption 1}$$

$$\tag{22}$$

$$L_2 = \text{Upper bound on 2-norm of true parameters } \psi_h, h \in [H] \qquad \text{Assumption 1} \qquad (23)$$

$$\tilde{L}_2 = L_2(8H^2 d_0/\alpha + 1) \qquad\qquad \text{Defined in Lemma 4.2} \qquad (24)$$

$$\sqrt{\lambda} = H^{3/2} d / \tilde{L}_2 \qquad\qquad \text{Defined in Eq. (57)} \qquad (25)$$

$$\eta \leq \frac{H^{3/2} d}{\sqrt{n}(10H^2 d_0/\alpha + 1)} = \tilde{\Theta}\left(\frac{\alpha}{\sqrt{nH}}\right) = \tilde{\Theta}\left(\frac{\epsilon^2}{C_{\text{conc}}^2 H^5 d^2}\right) \quad \text{Defined in Eq. (59)} \qquad (26)$$

$$\tilde{\eta} = \eta(10H^2 d_0/\alpha + 1) = \tilde{\mathcal{O}}\left(\frac{H^{3/2} d}{\sqrt{n}}\right) \qquad\qquad \text{Defined in Lemma 4.2} \qquad (27)$$

$$\check{\epsilon} = \tilde{\mathcal{O}}\left(d/\sqrt{n}\right) \qquad\qquad \text{Defined in Eq. (72)} \qquad (28)$$

$$\bar{\epsilon} = \tilde{\mathcal{O}}\left(\frac{C_{\text{conc}} H^{5/2} d^2}{\sqrt{n}}\right) \qquad\qquad \text{Defined in Eq. (51)} \qquad (29)$$

$$\tilde{\epsilon} = \tilde{\mathcal{O}}\left(\frac{C_{\text{conc}}^2 H^{5/2} d^2}{\sqrt{n}}\right) \qquad\qquad \text{Defined in Eq. (44)} \qquad (30)$$

$$\bar{\beta} = \tilde{\mathcal{O}}\left(H^{3/2} d\right) \qquad\qquad \text{Defined in Eq. (77)} \qquad (31)$$

$$\beta = \tilde{\mathcal{O}}\left(H^{3/2} d\right) \qquad\qquad \text{Defined in Eq. (58)} \qquad (32)$$

$$|C_\xi^{\mathbf{G}}| \leq (1 + 2L_2/\xi))^{dHd_0}, \xi > 0 \qquad\qquad \text{Defined in Lemma I.4} \qquad (33)$$

$$L_\xi = 12\sqrt{2d} H^2 L_1 \xi^{-1}\left(2\sqrt{n} L_1 \tilde{L}_2/(H^{3/2} d)\right)^H, \xi > 0 \qquad \text{Defined in Eq. (88)} \qquad (34)$$

$$\alpha = \frac{\epsilon}{12(H+1)} < 1 \qquad\qquad \text{Defined at the end of Section 5}$$

$$\tag{35}$$

# B  Trajectory Data vs. Non-Trajectory Data

To compare the two types of data, we first define "non-trajectory" data, which consist of individual transitions without any guarantees of them coming from complete MDP trajectories.

**Assumption 4** (Non-Trajectory Data). *Assume that for each $h \in [H]$ the learner is given a dataset of transition tuples and corresponding features of size $n \geq 1$:*

$$(s_h^j, a_h^j, r_h^j, \bar{s}_{h+1}^j)_{j \in [n]} \quad and \quad ((\phi(s_h^j, a))_{a \in \mathcal{A}}, (\phi(\bar{s}_{h+1}^j, a))_{a \in \mathcal{A}})_{j \in [n]},$$

*where $(s_h^j, a_h^j) \sim \mu_h', r_h^j \sim \mathcal{R}(s_h^j, a_h^j), \bar{s}_{h+1}^j \sim P(s_h^j, a_h^j),$ for all $j \in [n]$, and $\mu_h' \in \mathcal{M}_1(\mathcal{S}_h \times \mathcal{A})$ is a "data collection distribution" unknown to the learner.*

There are two differences between the above non-trajectory data and trajectory data as defined in Assumption 2. In particular, if $\mu_h' = \mathbb{P}_{\pi^0, s_1}^h$ and $\bar{s}_{h+1}^j = s_{h+1}^j$, then we get back the trajectory data assumption.

Foster et al. [2021] showed a negative result under Assumption 4. Our method addresses the hard instance from [Foster et al., 2021] if the data is given as complete trajectories. Below we explain why the lower bound constructions from [Foster et al., 2021] break down if they need to use trajectory data, and why our algorithm breaks down if it doesnt have trajectory data.

The lower bound constructions in Theorems 1.1 and 1.2 of [Foster et al., 2021] were both made hard because the data collection distributions of individual transition tuples $(s, a, r, \bar{s})$ were selected such that they reveal no (or almost no) information about the MDP instance. In both cases, receiving samples from the joint distribution of the entire trajectory $\mathbb{P}_{\pi^0, s_1}$ makes the problem easy. In the case of Theorem 1.1, one would simply observe which states are reachable from the start state (the planted states). For Theorem 1.2, some information on whether any next-state $\bar{s}$ is planted or not would be leaking in each trajectory, in the form of being able to observe the next-state transition from exactly $\bar{s}$.

A simpler example showing the root of the problem with non-trajectory data is as follows. Consider the toy problem of learning the value of some policy $\pi$ after taking action $a$ in state $s_1$ in a 2-stage MDP. The data is given as tuples of the form $(s_1, a, r_1^1, \bar{s}_2^1, \ldots, s_1, a, r_1^n, \bar{s}_2^n)$ for the first stage and $(s_2^1, a_2^1, r_2^1, \bar{s}_3^1, \ldots, s_2^n, a_2^n, r_2^n, \bar{s}_3^n)$ for the second stage. Notice there is no guarantee that $s_2^j \sim P(s_1, a)$ with $j \in [n]$. We cannot infer what the rewards from the second-stage states distributed as $P(s_1, a)$ might look like from the data, making this problem hopelessly hard. In the extreme, the MDP might have infinitely many second-stage states, with the probability of any $\bar{s}_2^j = s_2^k$ (for any $j$ and $k$) being 0, highlighting that one cannot just connect and importance weight matching next-states $\bar{s}_2^j$ of the first-stage transitions with matching start-states $s_2^k$ of the second-stage transitions. In contrast, if we assume the data is such that $\bar{s}_2^j = s_2^j$, this problem is immediately avoided as samples from $P(s_1, a)$, along with rewards from those states are directly handed to the learner. The learner can then simply use all of the rewards $r_2^j$ from tuples that contain the action $\pi(s_2^j)$ (which we have on average at least $1/C_{\text{conc}}$ of, due to concentrability) to estimate the value of policy $\pi$ after taking $s_1, a$ (solving the toy problem).

Now consider our algorithm if we do not have trajectory data. In this case we are no longer able to construct least squares targets of the form needed to make use of Lemma 4.2 (discussed in Section 4.2). This means that we would not be able to guarantee that our targets are linear, even under the true skipping mechanism $\bar{G}$, implying that $\bar{G}$ might not be a feasible solution to our optimization problem. Then our optimism argument that the output of the optimization problem has a value estimate at least as large as the value estimate based on $\bar{G}$ would no longer hold, causing our whole proof strategy to break down.

# C  Proof of Lemma 4.2

*Proof.* We follow a proof technique introduced in [Weisz et al., 2023]. We start by quoting their definition of admissible functions and their admissible realizability lemma.

**Definition 2** (Definition 4.6 in [Weisz et al., 2023]). *For any $h \in [H]$, $f : \mathcal{S}_h \to \mathbb{R}$ is $\alpha'$-admissible for some $\alpha' > 0$ if for all $s \in \mathcal{S}_h$, $|f(s)| \leq \text{range}(s)/\alpha'$.*

**Lemma C.1** (Admissible-realizability (Lemma 4.7 in [Weisz et al., 2023])). *If $f : \mathcal{S}_h \to \mathbb{R}$ is $\alpha'$-admissible then it is realizable, that is, for all $t \in [h-1]$ and $\pi \in \Pi$, there exists some $\tilde{\theta} \in \mathbb{R}^d$ with $\left\| \tilde{\theta} \right\|_2 \leq 4d_0 L_2/\alpha'$ such that for all $(s,a) \in \mathcal{S}_t \times \mathcal{A}$,*

$$\left| \mathbb{E}_{\text{Traj} \sim \mathbb{P}_{\pi,s,a}} f(S_h) - \left\langle \phi(s,a), \tilde{\theta} \right\rangle \right| \leq \eta_0 \qquad \text{where } \eta_0 = 5d_0\eta/\alpha'.$$

Next, we fix some $f : \mathcal{S} \to [0, H]$ with $f(s_\top) = 0$, and policy $\pi \in \Pi$. For $2 \leq h \leq H+1$, define $\tilde{g}_h : \mathcal{S}_h \to [-H, H]$ as $\tilde{g}_{H+1}(\cdot) = 0$, and

$$\tilde{g}_h(s) = \mathbb{E}_{\text{Traj} \sim \mathbb{P}_{\pi,s}} \mathbb{E}_{\tau \sim F_{\bar{G},\text{Traj},h}} [-R_{\tau:H} + f(S_\tau)]$$

$$= \mathbb{E}_{\text{Traj} \sim \mathbb{P}_{\pi,s}} \sum_{t=h}^{H} [-R_{t:H} + f(S_t)](1 - \omega_{\bar{G}}(S_t)) \prod_{u=h}^{t-1} \omega_{\bar{G}}(S_u).$$

Notice that for any $h \in [H]$, $(s,a) \in \mathcal{S}_h \times \mathcal{A}$, the function of $(s,a)$ we aim to linearly realize can be written as

$$\mathbb{E}_{\text{Traj} \sim \mathbb{P}_{\pi,s,a}} \mathbb{E}_{\tau \sim F_{\bar{G},\text{Traj},h}} [R_{h:\tau-1} + f(S_\tau)] = \mathbb{E}_{\text{Traj} \sim \mathbb{P}_{\pi,s,a}} \tilde{g}_{h+1}(S_{h+1}) + R_{h:H}$$

$$= \mathbb{E}_{S_{h+1} \sim P(s,a)} \tilde{g}_{h+1}(S_{h+1}) + q^\pi(s,a). \tag{36}$$

The second term of the sum, $q^\pi(s,a)$ is linearly realizable by Assumption 1 with parameter $\psi_h(\pi)$. The first term needs more work before Lemma C.1 can be applied. To this end, for $h \in [2 : H]$ we define $g_h : \mathcal{S}_h \to \mathbb{R}$ as

$$g_h(s) = (1 - \omega_{\bar{G}}(s)) \mathbb{E}_{\text{Traj} \sim \mathbb{P}_{\pi,s}} [-R_{h:H} + f(s) - \tilde{g}_{h+1}(S_{h+1})].$$

Notice that $\tilde{g}_h$ can be decomposed into a sum of $g_t$ functions as for all $h \in [2 : H]$, $s \in \mathcal{S}_h$,

$$\tilde{g}_h(s) = \mathbb{E}_{\text{Traj} \sim \mathbb{P}_{\pi,s}} \sum_{t=h}^{H} g_t(S_t). \tag{37}$$

The benefit of decomposing $\tilde{g}_h$ into $g_t$ functions is that $g_t$ are $\alpha'$-admissible under Definition 2 for $\alpha' = \alpha/(2H)$. To see this, note that for any trajectory and $s$, $-R_{h:H} + f(s) - \tilde{g}_{h+1}(S_{h+1}) \in [-2H, 2H]$. $g_t(s)$ multiplies this by $1 - \omega_{\bar{G}}(s)$ which by Eq. (6) is between 0 and 1, and satisfies $1 - \omega_{\bar{G}}(s) = 0$ if $\text{range}^{\bar{G}}(s) \leq \alpha/\sqrt{2d}$. By Lemma 4.1, $\text{range}(s) \leq \sqrt{2d} \cdot \text{range}^{\bar{G}}(s)$, so $g_t(s) = 0$ for any $s$ with $\text{range}(s) \leq \alpha$. On any other $s$, $|g_t(s)| \leq 2H$. Therefore $g_t$ is $\alpha'$-admissible. This allows us to use Lemma C.1 to get that for any $h \in [H-1]$, there exist $\tilde{\theta}_{h+1:H} \in \mathcal{B}(8Hd_0 L_2/\alpha)^{H-h}$, such that for any stage $t \in [h+1 : H]$, for all $(s,a) \in \mathcal{S}_h \times \mathcal{A}$,

$$\left| \mathbb{E}_{\text{Traj} \sim \mathbb{P}_{\pi,s,a}} g_t(S_t) - \left\langle \phi(s,a), \tilde{\theta}_t \right\rangle \right| \leq 10Hd_0\eta/\alpha.$$

By combining this with Eq. (37), for all $h \in [H]$ there exists $\tilde{\theta} = \sum_{t=h+1}^{H} \tilde{\theta}_t$ with $\tilde{\theta} \in \mathcal{B}(8H^2 d_0 L_2/\alpha)$ such that for all $(s,a) \in \mathcal{S}_h \times \mathcal{A}$,

$$\left| \mathbb{E}_{S_{h+1} \sim P(s,a)} \tilde{g}_{h+1}(S_{h+1}) - \left\langle \phi(s,a), \tilde{\theta} \right\rangle \right| \leq 10H^2 d_0\eta/\alpha.$$

Combined with Eq. (36) and the parameter $\psi_h(\pi)$ from Assumption 1, there exists $\theta = \tilde{\theta} + \psi_h(\pi)$ with $\theta \in \mathcal{B}(L_2(8H^2 d_0/\alpha + 1))$ such that for all $(s,a) \in \mathcal{S}_h \times \mathcal{A}$,

$$\left| \mathbb{E}_{\text{Traj} \sim \mathbb{P}_{\pi,s,a}} \mathbb{E}_{\tau \sim F_{\bar{G},\text{Traj},h}} [R_{h:\tau-1} + f(S_\tau)] - \langle \phi(s,a), \theta \rangle \right| \leq \eta(10H^2 d_0/\alpha + 1) = \tilde{\eta}.$$

To finish the proof, we define $\rho_h^\pi(f) = \theta$ for the arbitrary $h \in [H]$, $\pi$, and $f$ picked above. $\qquad \square$

# D  Results used in Section 5

## D.1  Bounding term (I)

We begin by defining an alternative policy $\pi_{\bar{G}}^{\dagger}$ as

$$\pi_{\bar{G}}^{\dagger}(a|s) = \pi^0(a|s)\omega_{\bar{G}}(s) + \pi^{\star}(a|s)(1 - \omega_{\bar{G}}(s)).$$

This policy can only be worse in value than $\pi_{\bar{G}}^{\star}$:

**Lemma D.1.** *For all $s \in \mathcal{S}$,*

$$v^{\pi_{\bar{G}}^{\star}}(s) \geq v^{\pi_{\bar{G}}^{\dagger}}(s).$$

*Proof.* We prove by induction for $h = H + 1, H, \ldots, 1$ that for all $s \in \mathcal{S}_h$, $v^{\pi_{\bar{G}}^{\star}}(s) \geq v^{\pi_{\bar{G}}^{\dagger}}(s)$. The base case of $h = H + 1$ is immediately true by definition as $v$-values are 0 on $s_{\top}$, regardless of the policy. Assuming the inductive hypothesis holds for $h + 1$, we continue by proving it for $h$. Let $(s, a) \in \mathcal{S}_h \times \mathcal{A}$ be arbitrary. Notice that

$$q^{\pi_{\bar{G}}^{\star}}(s, a) - q^{\pi_{\bar{G}}^{\dagger}}(s, a) = \mathop{\mathbb{E}}_{S' \sim P(s,a)} v^{\pi_{\bar{G}}^{\star}}(S') - v^{\pi_{\bar{G}}^{\dagger}}(S') \geq 0,$$

where the inequality is due to the inductive hypothesis. Next, for any $s \in \mathcal{S}_h$, by the above and the definition of the policies,

$$v^{\pi_{\bar{G}}^{\star}}(s) = \omega_{\bar{G}}(s) \mathop{\mathbb{E}}_{A \sim \pi^0(s)} q^{\pi_{\bar{G}}^{\star}}(s, A) + (1 - \omega_{\bar{G}}(s)) \max_{a \in \mathcal{A}} q^{\pi_{\bar{G}}^{\star}}(s, a)$$

$$\geq \omega_{\bar{G}}(s) \mathop{\mathbb{E}}_{A \sim \pi^0(s)} q^{\pi_{\bar{G}}^{\dagger}}(s, A) + (1 - \omega_{\bar{G}}(s)) \max_{a \in \mathcal{A}} q^{\pi_{\bar{G}}^{\dagger}}(s, a)$$

$$\geq \omega_{\bar{G}}(s) \mathop{\mathbb{E}}_{A \sim \pi^0(s)} q^{\pi_{\bar{G}}^{\dagger}}(s, A) + (1 - \omega_{\bar{G}}(s)) \mathop{\mathbb{E}}_{A \sim \pi_{\bar{G}}^{\star}(s)} q^{\pi_{\bar{G}}^{\dagger}}(s, A) = v^{\pi_{\bar{G}}^{\dagger}}(s),$$

finishing the induction. $\qquad\square$

Due to Lemma D.1, $v^{\star}(s_1) - v^{\pi_{\bar{G}}^{\star}}(s_1) \leq v^{\star}(s_1) - v^{\pi_{\bar{G}}^{\dagger}}(s_1)$. We continue by bounding $v^{\star}(s_1) - v^{\pi_{\bar{G}}^{\dagger}}(s_1)$. We first decompose it using the performance difference lemma (Lemma J.3) to get that

$$v^{\star}(s_1) - v^{\pi_{\bar{G}}^{\dagger}}(s_1) = \sum_{h=2}^{H} \mathop{\mathbb{E}}_{(S_h, A_h) \sim \mathbb{P}_{\pi^{\star}, s_1}^h} \left( q^{\pi_{\bar{G}}^{\dagger}}(S_h, A_h) - v^{\pi_{\bar{G}}^{\dagger}}(S) \right), \tag{38}$$

where we only have the sum from $h = 2$ to $H$, since for $h = 1$ and $h = H + 1$, for any $s \in \mathcal{S}_h$, $\omega_{\bar{G}}(s) = 0$ and therefore $\pi_{\bar{G}}^{\dagger}(s) = \pi^{\star}(s)$. By the definition of $\omega_{\bar{G}}$ (Eq. (6)), we can see that for $s \in \mathcal{S}, \omega_{\bar{G}}(s) \neq 0$ only when $\text{range}^{\bar{G}}(s) \leq 2\alpha/\sqrt{2d}$ Thus, the policies $\pi^{\star}(s)$ and $\pi_{\bar{G}}^{\dagger}(s)$ are equal for all $s \in \mathcal{S}$ that satisfy $\text{range}^{\bar{G}}(s) \geq 2\alpha/\sqrt{2d}$. Making use of this result in Eq. (38), we get that

$$\sum_{h=2}^{H} \mathop{\mathbb{E}}_{(S_h, A_h) \sim \mathbb{P}_{\pi^{\star}, s_1}^h} \left( q^{\pi_{\bar{G}}^{\dagger}}(S_h, A_h) - v^{\pi_{\bar{G}}^{\dagger}}(S) \right)$$

$$= \sum_{h=2}^{H} \mathop{\mathbb{E}}_{(S_h, A_h) \sim \mathbb{P}_{\pi^{\star}, s_1}^h} \left[ \mathbb{1}\left\{ \text{range}^{\bar{G}}(S_h) \leq 2\alpha/\sqrt{2d} \right\} \left( q^{\pi_{\bar{G}}^{\dagger}}(S_h, A_h) - v^{\pi_{\bar{G}}^{\dagger}}(S_h) \right) \right].$$

By Eq. (3), we know that

$$q^{\pi_{\bar{G}}^{\dagger}}(s, a) - v^{\pi_{\bar{G}}^{\dagger}}(s) \leq \text{range}(s) + 2\eta.$$

Then, we can use Lemma 4.1 to get that for all $h \in [2 : H]$ and $(s, a) \in \mathcal{S}_h \times \mathcal{A}$

$$q^{\pi_{\bar{G}}^{\dagger}}(s, a) - v^{\pi_{\bar{G}}^{\dagger}}(s) \leq \sqrt{2d} \cdot \text{range}^{\bar{G}}(s) + 2\eta.$$

Putting things together we get the following bound.

$$
\begin{aligned}
(\mathrm{I}) &\leq v^{\star}(s_1) - v^{\pi_{\bar{G}}^{\star}}(s_1) \\
&\leq v^{\star}(s_1) - v^{\pi_{\bar{G}}^{\dagger}}(s_1) \\
&= \sum_{h=2}^{H} \mathbb{E}_{(S_h, A_h) \sim \mathbb{P}_{\pi^{\star}, s_1}^{h}} \left[ \mathbb{1}\left\{ \mathrm{range}^{\bar{G}}(S) \leq 2\alpha/\sqrt{2d} \right\} \left( q^{\pi_{\bar{G}}^{\dagger}}(S, A) - v^{\pi_{\bar{G}}^{\dagger}}(S) \right) \right] \\
&\leq H(2\alpha + 2\eta).
\end{aligned}
$$

## D.2 Bounding term $(\mathrm{II})$

To bound $v^{\pi_{\bar{G}}^{\star}}(s_1) - \bar{v}_{\theta'_1}(s_1)$ we decompose it into the following error terms

$$
\begin{aligned}
v^{\pi_{\bar{G}}^{\star}}(s_1) - \bar{v}_{\theta'_1}(s_1) &= v^{\pi_{\bar{G}}^{\star}}(s_1) - \bar{v}_{\psi_1(\pi_{\bar{G}}^{\star})}(s_1) + \bar{v}_{\psi_1(\pi_{\bar{G}}^{\star})}(s_1) - \bar{v}_{\theta'_1}(s_1) \\
&\leq \bar{v}_{\psi_1(\pi_{\bar{G}}^{\star})}(s_1) - \bar{v}_{\theta'_1}(s_1) + \eta,
\end{aligned} \tag{39}
$$

where the inequality holds since we have approximate linear $q^{\pi}$-realizability (Assumption 1), which implies that

$$
v^{\pi_{\bar{G}}^{\star}}(s_1) - \bar{v}_{\psi_1(\pi_{\bar{G}}^{\star})}(s_1) \leq \max_{a \in \mathcal{A}} \left( q^{\pi_{\bar{G}}^{\star}}(s_1, a) - \bar{q}_{\psi_1(\pi_{\bar{G}}^{\star})}(s_1, a) \right) \leq \eta.
$$

To help us bound $\bar{v}_{\psi_1(\pi_{\bar{G}}^{\star})}(s_1) - \bar{v}_{\theta'_1}(s_1)$ in Eq. (39) we make use of two lemmas. The first is the following (proof in Appendix E).

**Lemma D.2.** *There is an event $\mathcal{E}_1$, which occurs with probability at least $1 - \delta/3$, such that under $\mathcal{E}_1$, for all $G \in \mathbf{G}$ and $h \in [H+1]$, it holds that $\psi_h(\pi_G^{\star}) \in \Theta_{G,h}$.*

Lemma D.2 tells us that under event $\mathcal{E}_1$, $\psi_1(\pi_{\bar{G}}^{\star}) \in \Theta_{\bar{G},1}$. The second lemma is the following (proof in Appendix G).

**Lemma D.3** (Feasibility). *There is an event $\mathcal{E}_1 \cap \mathcal{E}_2 \cap \mathcal{E}_3$, which occurs with probability at least $1 - \delta$, such that under event $\mathcal{E}_1 \cap \mathcal{E}_2 \cap \mathcal{E}_3$, the true guess $\bar{G}$ is a feasible solution to Optimization Problem 1.*

Notice that since $(G', \theta'_{1:H+1})$ is the solution to Optimization Problem 1, it holds that $\bar{v}_{\theta'_1}(s_1) \geq \bar{v}_{\theta}(s_1)$ for any $\theta \in \Theta_{G,1}$ where $G$ is a feasible solution to Optimization Problem 1. Thus, we get that under event $\mathcal{E}_1 \cap \mathcal{E}_2 \cap \mathcal{E}_3$, since $\psi_1(\pi_{\bar{G}}^{\star}) \in \Theta_{\bar{G},1}$ (by Lemma D.3), and $\bar{G}$ is a feasible solution to Optimization Problem 1 (by Lemma D.3), it holds that

$$
\bar{v}_{\psi_1(\pi_{\bar{G}}^{\star})}(s_1) - \bar{v}_{\theta'_1}(s_1) \leq \eta,
$$

which together with Eq. (39), implies that

$$
(\mathrm{II}) = v^{\pi_{\bar{G}}^{\star}}(s_1) - \bar{v}_{\theta'_1}(s_1) \leq \eta.
$$

## D.3 Proof of Lemma 5.2

*Proof.* Recall that $(G', \theta'_{1:H+1})$ is the solution to Optimization Problem 1. We aim to show that under event $\mathcal{E}_1 \cap \mathcal{E}_2$, for any $h \in [2 : H+1]$, it holds that

$$
\mathbb{E}_{\mathrm{Traj} \sim \mathbb{P}_{\pi', s_1}} \left[ v^{\pi_{G'}^{\star}}(S_h) - v^{\pi'}(S_h) \right] \leq 2(H - h + 2)(\eta + \tilde{\epsilon}). \tag{40}
$$

To prove Eq. (40) we will use induction. The base case is when $h = H + 1$, which trivially holds, since $v^{\pi}(s_{\top}) = r_{H+1}(s, a) = 0$ for all $\pi \in \Pi$, $s_{\top} \in \mathcal{S}_{H+1}$, $a \in \mathcal{A}$.

Now, we show the inductive step. Let $h \in [2 : H]$ be arbitrary. Assume that Eq. (40) holds for any $t \in [h + 1 : H + 1]$. We prove that Eq. (40) also holds for $h$. For any $(s, a) \in (\mathcal{S} \backslash \mathcal{S}_1) \times \mathcal{A}$, let

$$
\tilde{\pi}_h(a|s) = \begin{cases} \pi_{G'}^{\star}(a|s) & \text{if } \mathrm{stage}(s) = h \\ \pi'(a|s) & \text{if } \mathrm{stage}(s) \neq h. \end{cases}
$$

Then

$$\mathop{\mathbb{E}}_{\text{Traj}\sim\mathbb{P}_{\pi',s_1}}\left[v^{\pi^\star_{G'}}(S_h) - v^{\pi'}(S_h)\right]$$

$$= \mathop{\mathbb{E}}_{\text{Traj}\sim\mathbb{P}_{\bar\pi_h,s_1}} q^{\pi^\star_{G'}}(S_h, A_h) - \mathop{\mathbb{E}}_{\text{Traj}\sim\mathbb{P}_{\pi',s_1}} q^{\pi'}(S_h, A_h)$$

$$= \mathop{\mathbb{E}}_{\text{Traj}\sim\mathbb{P}_{\bar\pi_h,s_1}}\left[q^{\pi^\star_{G'}}(S_h, A_h) - \bar q_{\theta'_h}(S_h, A_h)\right] + \mathop{\mathbb{E}}_{\text{Traj}\sim\mathbb{P}_{\bar\pi_h,s_1}} \bar q_{\theta'_h}(S_h, A_h) - \mathop{\mathbb{E}}_{\text{Traj}\sim\mathbb{P}_{\pi',s_1}} \bar q_{\theta'_h}(S_h, A_h)$$

$$+ \mathop{\mathbb{E}}_{\text{Traj}\sim\mathbb{P}_{\pi',s_1}}\left[\bar q_{\theta'_h}(S_h, A_h) - q^{\pi^\star_{G'}}(S_h, A_h)\right] + \mathop{\mathbb{E}}_{\text{Traj}\sim\mathbb{P}_{\pi',s_1}}\left[q^{\pi^\star_{G'}}(S_h, A_h) - q^{\pi'}(S_h, A_h)\right]$$

$$\leq \mathop{\mathbb{E}}_{\text{Traj}\sim\mathbb{P}_{\bar\pi_h,s_1}}\left[\bar q_{\psi_h(\pi^\star_{G'})}(S_h, A_h) - \bar q_{\theta'_h}(S_h, A_h)\right] + \mathop{\mathbb{E}}_{\text{Traj}\sim\mathbb{P}_{\bar\pi_h,s_1}} \bar q_{\theta'_h}(S_h, A_h) - \mathop{\mathbb{E}}_{\text{Traj}\sim\mathbb{P}_{\pi',s_1}} \bar q_{\theta'_h}(S_h, A_h)$$

$$+ \mathop{\mathbb{E}}_{\text{Traj}\sim\mathbb{P}_{\pi',s_1}}\left[\bar q_{\theta'_h}(S_h, A_h) - \bar q_{\psi_h(\pi^\star_{G'})}(S_h, A_h)\right] + \mathop{\mathbb{E}}_{\text{Traj}\sim\mathbb{P}_{\pi',s_1}}\left[q^{\pi^\star_{G'}}(S_h, A_h) - q^{\pi'}(S_h, A_h)\right] + 2\eta\,,$$

$$\tag{41}$$

where the inequality holds since we have approximate linear $q^\pi$-realizability (Assumption 1). To bound the first and third error terms above notice that under event $\mathcal{E}_1$, by Lemma D.2, we know that $\psi_h(\pi^\star_{G'}) \in \Theta_{G',h}$. We also know that $\theta'_h \in \Theta_{G',h}$, by definition. Let $\tilde\nu_u(s_u, a_u) = \mathbb{P}^u_{\bar\pi_h,s_1}(s_u, a_u)$ and $\nu'_u(s_u, a_u) = \mathbb{P}^u_{\pi',s_1}(s_u, a_u)$ for all $u \in [H], (s_u, a_u) \in \mathcal{S}_u \times \mathcal{A}$. Clearly, $\tilde\nu = (\tilde\nu_u)_{u\in[H]}$ and $\nu' = (\nu'_u)_{u\in[H]}$ are admissible distributions, by Definition 1. Notice that we have satisfied all the conditions to make use of Lemma 5.1. Thus, under event $\mathcal{E}_1 \cap \mathcal{E}_2$ it holds that

$$\mathop{\mathbb{E}}_{\text{Traj}\sim\mathbb{P}_{\bar\pi_h,s_1}}\left[\bar q_{\psi_h(\pi^\star_{G'})}(S_h, A_h) - \bar q_{\theta'_h}(S_h, A_h)\right] = \mathop{\mathbb{E}}_{(S_h,A_h)\sim\tilde\nu_h}\left[\bar q_{\psi_h(\pi^\star_{G'})}(S_h, A_h) - \bar q_{\theta'_h}(S_h, A_h)\right] \leq \tilde\epsilon\,,$$

and

$$\mathop{\mathbb{E}}_{\text{Traj}\sim\mathbb{P}_{\pi',s_1}}\left[\bar q_{\theta'_h}(S_h, A_h) - \bar q_{\psi_h(\pi^\star_{G'})}(S_h, A_h)\right] = \mathop{\mathbb{E}}_{(S_h,A_h)\sim\nu'_h}\left[\bar q_{\theta'_h}(S_h, A_h) - \bar q_{\psi_h(\pi^\star_{G'})}(S_h, A_h)\right] \leq \tilde\epsilon\,.$$

The term $\mathbb{E}_{\text{Traj}\sim\mathbb{P}_{\bar\pi_h,s_1}} \bar q_{\theta'_h}(S_h, A_h) - \mathbb{E}_{\text{Traj}\sim\mathbb{P}_{\pi',s_1}} \bar q_{\theta'_h}(S_h, A_h)$ in Eq. (41) can be bounded by recalling the definition of $\pi'(s)$ (line 3), to get that

$$\mathop{\mathbb{E}}_{\text{Traj}\sim\mathbb{P}_{\bar\pi_h,s_1}} \bar q_{\theta'_h}(S_h, A_h) \leq \max_{a\in\mathcal{A}} \bar q_{\theta'_h}(S_h, a) = \mathop{\mathbb{E}}_{\text{Traj}\sim\mathbb{P}_{\pi',s_1}} \bar q_{\theta'_h}(S_h, A_h)\,.$$

Under event $\mathcal{E}_1 \cap \mathcal{E}_2$, after plugging the above bounds into Eq. (41), we have that

$$\mathop{\mathbb{E}}_{\text{Traj}\sim\mathbb{P}_{\pi',s_1}}\left[v^{\pi^\star_{G'}}(S_h) - v^{\pi'}(S_h)\right] \leq \mathop{\mathbb{E}}_{\text{Traj}\sim\mathbb{P}_{\pi',s_1}}\left[q^{\pi^\star_{G'}}(S_h, A_h) - q^{\pi'}(S_h, A_h)\right] + 2\eta + 2\tilde\epsilon$$

$$= \mathop{\mathbb{E}}_{\text{Traj}\sim\mathbb{P}_{\pi',s_1}}\left[v^{\pi^\star_{G'}}(S_{h+1}) - v^{\pi'}(S_{h+1})\right] + 2\eta + 2\tilde\epsilon$$

$$\leq 2(H - h + 2)(\eta + \tilde\epsilon)\,,$$

where the last inequality holds by the inductive hypothesis for $h+1$ (Eq. (40)), completing the proof of the claim. $\qquad\square$

# E  Proof of Lemma D.2

*Proof.* We will prove the claim using induction. The base case is when $h = H + 1$, for which $\psi_{H+1}(\pi) = \vec{0}$ for all $\pi \in \Pi$ by definition. Thus, for all $G \in \mathbf{G}$, it holds that $\psi_{H+1}(\pi_G^\star) \in \Theta_{G,H+1} = \{\vec{0}\}$.

Now, we show the inductive step. Let $h \in [H]$ be arbitrary. Assume Lemma D.2 holds for any $t \in [h+1 : H+1]$. We prove that it also holds for $h$. Define

$$\hat{\psi}_h(\pi_G^\star) = X_h^{-1} \sum_{j \in [n]} \phi_h^j \underset{\tau \sim F_{G,h+1}^j}{\mathbb{E}} \left[ r_{h:\tau-1}^j + \bar{v}_{\psi_{h+1:H+1}(\pi_G^\star)}\big(s_\tau^j\big) \right].$$

By the inductive hypothesis we know that for any $G \in \mathbf{G}$

$$\psi_{h+1:H+1}(\pi_G^\star) \in \Theta_{G,h+1} \times \cdots \times \Theta_{G,H+1}$$

Thus, $\hat{\psi}_h(\pi_G^\star) \in \hat{\Theta}_{G,h}$. It is left to show that $\left\| \hat{\psi}_h(\pi_G^\star) - \psi_h(\pi_G^\star) \right\|_{X_h} \leq \beta$, which together with the fact that $\psi_h(\pi_G^\star) \in \mathcal{B}(L_2)$, implies the desired result, $\psi_h(\pi_G^\star) \in \Theta_{G,h}$.

We would like to make use of Lemma H.1 to bound $\left\| \hat{\psi}_h(\pi_G^\star) - \psi_h(\pi_G^\star) \right\|_{X_h}$. To do so, we map the terms used in the Lemma H.1 to our terms as follows

$$n = n, \lambda = \lambda, \theta_\star = \psi_h(\pi_G^\star), V = X_h, \hat{\theta} = \hat{\psi}_h(\pi_G^\star), A = \left( \phi_h^j \right)_{j \in [n]},$$

$$Y = \tilde{Y} + \Delta = \left( \left\langle \phi_h^j, \psi_h(\pi_G^\star) \right\rangle \right)_{j \in [n]} + \gamma + \Delta = \left( \underset{\tau \sim F_{G,h+1}^j}{\mathbb{E}} \left[ r_{h:\tau-1}^j + \bar{v}_{\psi_{h+1:H+1}(\pi_G^\star)}\big(s_\tau^j\big) \right] \right)_{j \in [n]},$$

$$\gamma = \left( \underset{\tau \sim F_{G,h+1}^j}{\mathbb{E}} \left[ r_{h:\tau-1}^j + \bar{v}_{\psi_{h+1:H+1}(\pi_G^\star)}\big(s_\tau^j\big) \right] - \underset{\mathrm{Traj} \sim \mathbb{P}_{\pi^0, s_h^j, a_h^j}}{\mathbb{E}} \underset{\tau \sim F_{G,\mathrm{Traj},h+1}}{\mathbb{E}} \left[ R_{h:\tau-1} + \bar{v}_{\psi_{h+1:H+1}(\pi_G^\star)}(S_\tau) \right] \right)_{j \in [n]},$$

$$\Delta = \left( \underset{\mathrm{Traj} \sim \mathbb{P}_{\pi^0, s_h^j, a_h^j}}{\mathbb{E}} \underset{\tau \sim F_{G,\mathrm{Traj},h+1}}{\mathbb{E}} \left[ R_{h:\tau-1} + \bar{v}_{\psi_{h+1:H+1}(\pi_G^\star)}(S_\tau) \right] - \left\langle \phi_h^j, \psi_h(\pi_G^\star) \right\rangle \right)_{j \in [n]},$$

$$\iota = \sum_{j \in [n]} \phi_h^j \left( \underset{\tau \sim F_{G,h+1}^j}{\mathbb{E}} \left[ r_{h:\tau-1}^j + \bar{v}_{\psi_{h+1:H+1}(\pi_G^\star)}\big(s_\tau^j\big) \right] - \underset{\mathrm{Traj} \sim \mathbb{P}_{\pi^0, s_h^j, a_h^j}}{\mathbb{E}} \underset{\tau \sim F_{G,\mathrm{Traj},h+1}}{\mathbb{E}} \left[ R_{h:\tau-1} + \bar{v}_{\psi_{h+1:H+1}(\pi_G^\star)}(S_\tau) \right] \right).$$

With the definitions as above, applying Lemma H.1 we get

$$\left\| \hat{\psi}_h(\pi_G^\star) - \psi_h(\pi_G^\star) \right\|_{X_h} \leq \sqrt{\lambda} \|\psi_h(\pi_G^\star)\|_2 + \|\Delta\|_\infty \sqrt{n} + \|\iota\|_{X_h^{-1}}. \tag{42}$$

The first term in Eq. (42) can be bounded by $H^{3/2}d$ by recalling that $\sqrt{\lambda} = H^{3/2}d/\tilde{L}_2$ (Eq. (25)) and $\|\psi_h(\pi_G^\star)\|_2 \leq L_2 \leq \tilde{L}_2$.

The $\|\Delta\|_\infty$ in the second term in Eq. (42) can be bounded by first decomposing the error, and using a triangle inequality as follows.

$$\|\Delta\|_\infty \leq \left\| \left( \underset{\mathrm{Traj} \sim \mathbb{P}_{\pi^0, s_h^j, a_h^j}}{\mathbb{E}} \underset{\tau \sim F_{G,\mathrm{Traj},h+1}}{\mathbb{E}} \left[ \bar{v}_{\psi_{h+1:H+1}(\pi_G^\star)}(S_\tau) - \max_{a' \in \mathcal{A}} q^{\pi_G^\star}(S_\tau, a') \right] \right)_{j \in [n]} \right\|_\infty$$

$$+ \left\| \left( \underset{\mathrm{Traj} \sim \mathbb{P}_{\pi^0, s_h^j, a_h^j}}{\mathbb{E}} \underset{\tau \sim F_{G,\mathrm{Traj},h+1}}{\mathbb{E}} \left[ R_{h:\tau-1} + \max_{a' \in \mathcal{A}} q^{\pi_G^\star}(S_\tau, a') \right] - \left\langle \phi_h^j, \psi_h(\pi_G^\star) \right\rangle \right)_{j \in [n]} \right\|_\infty. \tag{43}$$

For the first term in Eq. (43), notice that

$$\bar{v}_{\psi_{h+1:H+1}(\pi_G^\star)}(S_\tau) - \max_{a' \in \mathcal{A}} q^{\pi_G^\star}(S_\tau, a') \leq \max_{a' \in \mathcal{A}} \left( \left\langle \phi(S_\tau, a'), \psi_{\mathrm{stage}(S_\tau)}(\pi_G^\star) \right\rangle - q^{\pi_G^\star}(S_\tau, a') \right) \leq \eta,$$

where the last inequality holds since we have approximate linear $q^\pi$-realizability (Assumption 1). To bound the second term in Eq. (43) notice that by the definition of $\pi_G^\star$ (Eq. (15)), $\arg\max_{a'\in\mathcal{A}} q^{\pi_G^\star}(S_\tau, a')$ is exactly the action $\pi_G^\star$ would take at the stopping stage $\tau$, and that the distribution of Traj under policy $\pi^0$ until stopping stage $\tau$ is same as the distribution of Traj under policy $\pi_G^\star$ until stopping stage $\tau$. This implies that

$$
\left\| \left( \underset{\text{Traj}\sim\mathbb{P}_{\pi^0,s_h^j,a_h^j}}{\mathbb{E}} \underset{\tau\sim F_{G,\text{Traj},h+1}}{\mathbb{E}} \left[ R_{h:\tau-1} + \max_{a'\in\mathcal{A}} q^{\pi_G^\star}(S_\tau, a') \right] - \left\langle \phi_h^j, \psi_h(\pi_G^\star) \right\rangle \right)_{j\in[n]} \right\|_\infty
$$
$$
= \left\| \left( q^{\pi_G^\star}(s_h^j, a_h^j) - \left\langle \phi_h^j, \psi_h(\pi_G^\star) \right\rangle \right)_{j\in[n]} \right\|_\infty \leq \eta \,,
$$

where the last inequality holds since we have approximate linear $q^\pi$-realizability (Assumption 1). Plugging the above bounds into Eq. (43), we get that $\|\Delta\|_\infty \leq 2\eta$.

To bound the third term in Eq. (42), let the event $\mathcal{E}_1$ be as defined in the proof of Lemma H.2, which occurs with probability at least $1 - \delta/3$. Then, under event $\mathcal{E}_1$, the third term in Eq. (42) can be bounded by $\bar{\beta}$ (Eq. (31)), by applying Lemma H.2 since $\psi_{h+1:H+1}(\pi_G^\star) \in \mathcal{B}(L_2)^{H-h+1} \subset \mathcal{B}(\tilde{L}_2)^{H-h+1}$.

Plugging the three bounds above back into Eq. (42) we get that, under event $\mathcal{E}_1$, it holds that

$$
\left\| \hat{\psi}_h(\pi_G^\star) - \psi_h(\pi_G^\star) \right\|_{X_h} \leq H^{3/2}d + 2\eta\sqrt{n} + \bar{\beta} \leq \beta \,,
$$

where $\beta$ is defined in Eq. (32), and the last inequality can be seen to hold by plugging in parameter values according to Appendix A. Thus, under event $\mathcal{E}_1$, which occurs with probability at least $1-\delta/3$, for any $G \in \mathbf{G}$ and $h \in [H]$ it holds that $\psi_h(\pi_G^\star) \in \Theta_{G,h}$, which completes the proof. $\qquad\square$

# F  Proof of Lemma 5.1

*Proof.* Let $G \in \mathbf{G}$ be a feasible solution to Optimization Problem 1. By Lemma I.1, there is an event $\mathcal{E}_2$, that occurs with probability at least $1 - \delta/3$, such that under event $\mathcal{E}_2$, for all $G \in \mathbf{G}$, and for all $h \in [H]$, it holds that

$$
\left| \mathop{\mathbb{E}}_{(S,A)\sim\mu_h} \left[ \max_{\theta\in\Theta_{G,h}} \bar{q}_\theta(S,A) - \min_{\theta\in\Theta_{G,h}} \bar{q}_\theta(S,A) \right] - \frac{1}{n} \sum_{i\in[n]} \left( \max_{\theta\in\Theta_{G,h}} \bar{q}_\theta(s_h^i, a_h^i) - \min_{\theta\in\Theta_{G,h}} \bar{q}_\theta(s_h^i, a_h^i) \right) \right|
$$

$$
\leq \frac{H}{\sqrt{n}} \sqrt{\log\left(\frac{6H|C_\xi^{\mathbf{G}}|}{\delta}\right)} + 2L_\xi \, .
$$

where $|C_\xi^{\mathbf{G}}|$, $L_\xi$, are defined in Eqs. (33) and (34). Let $h \in [H]$. Recall that since $G$ is a feasible solution to Optimization Problem 1 we know that Eq. (14) passed for $h$. Thus,

$$
\frac{1}{n} \sum_{i\in[n]} \left( \max_{\theta\in\Theta_{G,h}} \bar{q}_\theta(s_h^i, a_h^i) - \min_{\theta\in\Theta_{G,h}} \bar{q}_\theta(s_h^i, a_h^i) \right) \leq \bar{\epsilon} \, .
$$

Combining the above two results, we have that under event $\mathcal{E}_2$, for any $h \in [H]$, it holds that

$$
\mathop{\mathbb{E}}_{(S,A)\sim\mu_h} \left[ \max_{\theta\in\Theta_{G,h}} \bar{q}_\theta(S,A) - \min_{\theta\in\Theta_{G,h}} \bar{q}_\theta(S,A) \right] \leq \frac{H}{\sqrt{n}} \sqrt{\log\left(\frac{6H|C_\xi^{\mathbf{G}}|}{\delta}\right)} + 2L_\xi + \bar{\epsilon} \, .
$$

Now we will relate the data collecting distribution $\mu$ to any admissible distribution $\nu$. Notice that by the definition of max and min, for any $(s,a) \in \mathcal{S} \times \mathcal{A}$, it holds that

$$
\max_{\theta\in\Theta_{G,h}} \bar{q}_\theta(s,a) - \min_{\theta\in\Theta_{G,h}} \bar{q}_\theta(s,a) \geq 0 \, .
$$

Thus, we can apply Lemma G.4 to get that under event $\mathcal{E}_2$, for all $h \in [H]$, and admissible distribution $\nu = (\nu_t)_{t\in[H]}$, it holds that

$$
\mathop{\mathbb{E}}_{(S,A)\sim\nu_h} \left[ \max_{\theta\in\Theta_{G,h}} \bar{q}_\theta(S,A) - \min_{\theta\in\Theta_{G,h}} \bar{q}_\theta(S,A) \right] \leq C_{\text{conc}} \left( \frac{H}{\sqrt{n}} \sqrt{\log\left(\frac{6H|C_\xi^{\mathbf{G}}|}{\delta}\right)} + 2L_\xi + \bar{\epsilon} \right) .
$$

To conclude, we have that under event $\mathcal{E}_2$, for any $G \in \mathbf{G}$ that is a feasible solution to Optimization Problem 1, for any $h \in [H]$, for any $(\theta_{s,a})_{(s,a)\in\mathcal{S}_h\times\mathcal{A}}$ and $(\check{\theta}_{s,a})_{(s,a)\in\mathcal{S}_h\times\mathcal{A}}$ with $\theta_{s,a}, \check{\theta}_{s,a} \in \Theta_{G,h}$ for all $(s,a) \in \mathcal{S}_h \times \mathcal{A}$, and for any admissible distribution $\nu = (\nu_t)_{t\in[H]}$, it holds that

$$
\mathop{\mathbb{E}}_{(S,A)\sim\nu_h} \left[ \bar{q}_{\theta_{S,A}}(S,A) - \bar{q}_{\check{\theta}_{S,A}}(S,A) \right]
$$

$$
\leq \mathop{\mathbb{E}}_{(S,A)\sim\nu_h} \left[ \max_{\theta\in\Theta_{G,h}} \bar{q}_\theta(S,A) - \min_{\theta\in\Theta_{G,h}} \bar{q}_\theta(S,A) \right]
$$

$$
\leq C_{\text{conc}} \left( \frac{H}{\sqrt{n}} \sqrt{\log\left(\frac{6H|C_\xi^{\mathbf{G}}|}{\delta}\right)} + 2L_\xi + \bar{\epsilon} \right)
$$

$$
\leq C_{\text{conc}} \left( \frac{H}{\sqrt{n}} \sqrt{\log\left(\frac{6H(1 + 2L_2/\xi))^{dHd_0}}{\delta}\right)} + 24\sqrt{2d}H^2 L_1 \xi \alpha^{-1} \left(2\sqrt{n}L_1\tilde{L}_2/(H^{3/2}d)\right)^H + \bar{\epsilon} \right)
$$

$$
\leq C_{\text{conc}} \left( \frac{H}{\sqrt{n}} \sqrt{dH^2 d_0 \log\left(1 + 96\sqrt{2d}H^2 L_1 L_2 \alpha^{-1}\sqrt{n}L_1\tilde{L}_2/(H^{3/2}d)\right) + \log\left(\frac{6H}{\delta}\right)} + \frac{1}{\sqrt{n}} + \bar{\epsilon} \right)
$$

$$
= \tilde{\epsilon} = \tilde{\mathcal{O}}\left( \frac{C_{\text{conc}}H^2 d}{\sqrt{n}} + \frac{C_{\text{conc}}}{\sqrt{n}} + \frac{C_{\text{conc}}^2 H^{5/2}d^2}{\sqrt{n}} \right) = \tilde{\mathcal{O}}\left( \frac{C_{\text{conc}}^2 H^{5/2}d^2}{\sqrt{n}} \right) . \tag{44}
$$

The third inequality holds by plugging in the values of $|C_\xi^{\mathbf{G}}|$, $L_\xi$, as defined in Eqs. (33) and (34). The last inequality holds by setting $\xi^{-1} = 24\sqrt{2d}\sqrt{n}H^2 L_1 \alpha^{-1}\left(2\sqrt{n}L_1\tilde{L}_2/(H^{3/2}d)\right)^H$. The last two equalities hold by plugging in parameter values according to Appendix A. $\qquad\square$

# G    Proof of Lemma D.3

*Proof.* To show that $\bar{G}$ is a feasible solution we need to show that Eq. (14) is satisfied for all $h \in [H]$.

By Lemma I.1, there is an event $\mathcal{E}_2$, that occurs with probability at least $1 - \delta/3$, such that under event $\mathcal{E}_2$, for all $h \in [H]$, it holds that

$$\frac{1}{n} \sum_{i \in [n]} \left( \max_{\theta \in \Theta_{\bar{G},h}} \bar{q}_\theta(s_h^i, a_h^i) - \min_{\theta \in \Theta_{\bar{G},h}} \bar{q}_\theta(s_h^i, a_h^i) \right) - \mathbb{E}_{(S,A) \sim \mu_h} \left[ \max_{\theta \in \Theta_{\bar{G},h}} \bar{q}_\theta(S, A) - \min_{\theta \in \Theta_{\bar{G},h}} \bar{q}_\theta(S, A) \right]$$

$$\leq \frac{H}{\sqrt{n}} \sqrt{\log \left( \frac{6H|C_\xi^{\mathbf{G}}|}{\delta} \right)} + 2L_\xi. \tag{45}$$

where $|C_\xi^{\mathbf{G}}|, L_\xi$, are defined in Eqs. (33) and (34). Let $h \in [H]$ be arbitrary for the remainder of the proof. We focus on bounding $\mathbb{E}_{(S,A) \sim \mu_h} \left[ \max_{\theta \in \Theta_{\bar{G},h}} \bar{q}_\theta(S, A) - \min_{\theta \in \Theta_{\bar{G},h}} \bar{q}_\theta(S, A) \right]$ for the remainder of the proof. The following lemma will be helpful (proof in Appendix G.1).

**Lemma G.1.** *There is an event $\mathcal{E}_1$, that occurs with probability at least $1 - \delta/3$, such that under event $\mathcal{E}_1$, for any $h \in [H], (s,a) \in \mathcal{S}_h \times \mathcal{A}, \theta_h \in \Theta_{\bar{G},h}$, it holds that*

$$\left| \bar{q}_{\theta_h}(s,a) - q^{\pi_{\bar{G}}^\star}(s,a) \right| \leq 2\beta \mathbb{E}_{\mathrm{Traj} \sim \mathbb{P}_{\bar{\pi},s,a}} \sum_{t=h}^{H} \min \left\{ 1, \|\phi(S_t, A_t)\|_{X_t^{-1}} \right\} + (H - h + 1)\tilde{\eta}, \tag{46}$$

*where, for any $(s', a') \in \mathcal{S} \times \mathcal{A}$,*

$$\bar{\pi}(a'|s') = \pi^0(a'|s')\omega_{\bar{G}}(s') + \mathbb{1}\left\{ \arg\max_{a'' \in \mathcal{A}} g^{\bar{\pi}}(s', a'') = a' \right\} (1 - \omega_{\bar{G}}(s')), \tag{47}$$

*and $g^{\bar{\pi}}$ is a state-action value function of policy $\bar{\pi}$ (similar to $q^{\bar{\pi}}$), except in the alternative MDP that has the same state and action spaces, and transition distributions as the original MDP under consideration, but with a reward function modified as follows. For all $(s', a') \in \mathcal{S} \times \mathcal{A}$, the reward in this alternative MDP is deterministically $\min \left\{ 1, \|\phi(s', a')\|_{X_h^{-1}} \right\}$ $\left( i.e. \; \mathcal{R}(s', a') = \mathbb{1}\left\{ \min \left\{ 1, \|\phi(s', a')\|_{X_h^{-1}} \right\} \right\} \right)$. In particular for any $h' \in [H], (s', a') \in \mathcal{S}_{h'} \times \mathcal{A}$*

$$g^{\bar{\pi}}(s', a') = \mathbb{E}_{\mathrm{Traj} \sim \mathbb{P}_{\bar{\pi}, s', a'}} \sum_{t=h'}^{H} \min \left\{ 1, \|\phi(S_t, A_t)\|_{X_t^{-1}} \right\}. \tag{48}$$

*The recursive definition of $\bar{\pi}$ can be interpreted in the same way as described below Eq. (15).*

Let $\bar{\pi}$ be as defined in Lemma G.1. Then, by Lemma G.1, under event $\mathcal{E}_1$, it holds that

$$\mathbb{E}_{(S_h, A_h) \sim \mu_h} \left[ \max_{\theta \in \Theta_{\bar{G},h}} \bar{q}_\theta(S_h, A_h) - \min_{\theta \in \Theta_{\bar{G},h}} \bar{q}_\theta(S_h, A_h) \right]$$

$$= \mathbb{E}_{(S_h, A_h) \sim \mu_h} \left[ \max_{\theta \in \Theta_{\bar{G},h}} \bar{q}_\theta(S_h, A_h) - v^{\pi_{\bar{G}}^\star}(S_h) + v^{\pi_{\bar{G}}^\star}(S_h) - \min_{\theta \in \Theta_{\bar{G},h}} \bar{q}_\theta(S_h, A_h) \right]$$

$$\leq \mathbb{E}_{(S_h, A_h) \sim \mu_h} \left[ 4\beta \mathbb{E}_{\mathrm{Traj} \sim \mathbb{P}_{\bar{\pi}, S_h, A_h}} \sum_{t=h}^{H} \min \left\{ 1, \|\phi(S_t, A_t)\|_{X_t^{-1}} \right\} + 2(H - h + 1)\tilde{\eta} \right]$$

$$= 4\beta \mathbb{E}_{(S_h, A_h) \sim \mu_h} \min \left\{ 1, \|\phi(S_h, A_h)\|_{X_h^{-1}} \right\} + 4\beta \sum_{t=h+1}^{H} \mathbb{E}_{(S_h, A_h) \sim \mu_h} \mathbb{E}_{(S_t, A_t) \sim \mathbb{P}_{\bar{\pi}, S_h, A_h}^t} \min \left\{ 1, \|\phi(S_t, A_t)\|_{X_t^{-1}} \right\}$$

$$+ (H - h + 1)\tilde{\eta}. \tag{49}$$

The inequality used Lemma G.1. As we will show shortly, the first term can be bounded by Lemma G.3, since its expectation is taken w.r.t. the data collecting distribution $\mu$. Thus, the approach we take to bounding the second term is to relate its nested expectations to just be a single

expectation taken w.r.t. the distribution $\mu$ (similar to the first term, which we claim we know how to bound). To this end, for any $(s', a') \in \mathcal{S} \times \mathcal{A}$, define the policy $\check{\pi}_h$ as

$$\check{\pi}_h(a'|s') = \begin{cases} \pi^0(a'|s') & \text{if stage}(s') \leq h \\ \bar{\pi}(a'|s') & \text{if stage}(s') > h . \end{cases}$$

Notice that for any $t \in [H], u \in [t+1:H+1], \mathbb{E}_{(S_t,A_t)\sim\mu_t} \mathbb{E}_{(S_u,A_u)\sim\mathbb{P}^u_{\bar{\pi},S_t,A_t}} = \mathbb{E}_{(S_u,A_u)\sim\mathbb{P}^u_{\check{\pi}_t,s}}$. Let $\check{\nu}_u(s_u, a_u) = \mathbb{P}^u_{\check{\pi}_h,s_1}(s_u, a_u)$ for all $u \in [H], (s_u, a_u) \in \mathcal{S}_u \times \mathcal{A}$. Clearly, $\check{\nu} = (\check{\nu}_u)_{u\in[H]}$ is an admissible distribution by Definition 1. Thus, with the definition of $\check{\nu}$ and using Lemma G.4, we get that Eq. (49) is

$$= 4\beta \underset{(S_h,A_h)\sim\mu_h}{\mathbb{E}} \min\left\{1, \|\phi(S_h, A_h)\|_{X_h^{-1}}\right\} + 4\beta \sum_{t=h+1}^{H} \underset{(S_t,A_t)\sim\check{\nu}_h}{\mathbb{E}} \min\left\{1, \|\phi(S_t, A_t)\|_{X_t^{-1}}\right\} + (H - h + 1)\tilde{\eta}$$

$$\leq 4\beta \underset{(S_h,A_h)\sim\mu_h}{\mathbb{E}} \min\left\{1, \|\phi(S_h, A_h)\|_{X_h^{-1}}\right\} + 4C_{\text{conc}}\beta \sum_{t=h+1}^{H} \underset{(S_t,A_t)\sim\mu_t}{\mathbb{E}} \min\left\{1, \|\phi(S_t, A_t)\|_{X_t^{-1}}\right\} + (H - h + 1)\tilde{\eta}$$

$$\leq 4C_{\text{conc}}\beta \sum_{t=h}^{H} \underset{(S_t,A_t)\sim\mu_t}{\mathbb{E}} \min\left\{1, \|\phi(S_t, A_t)\|_{X_t^{-1}}\right\} + (H - h + 1)\tilde{\eta} .$$

The last inequality used that $C_{\text{conc}} \geq 1$. Finally, we can apply Lemma G.3 to bound $\mathbb{E}_{(S_t,A_t)\sim\mu_t} \min\left\{1, \|\phi(S_t, A_t)\|_{X_t^{-1}}\right\}$. In particular, let $\mathcal{E}_3$ be as defined in the proof of Lemma G.3. Then, by Lemma G.3, under event $\mathcal{E}_3$, it holds that

$$4C_{\text{conc}}\beta \sum_{t=h}^{H} \underset{(S_t,A_t)\sim\mu_t}{\mathbb{E}} \min\left\{1, \|\phi(S_t, A_t)\|_{X_t^{-1}}\right\} \leq 4C_{\text{conc}}\beta \sum_{t=h}^{H} \check{\epsilon} \leq 4HC_{\text{conc}}\check{\epsilon}\beta .$$

Putting all of the bound after Eq. (49) together and plugging them into Eq. (49), we have that, under event $\mathcal{E}_1 \cap \mathcal{E}_2$, it holds that

$$\underset{(S_h,A_h)\sim\mu_h}{\mathbb{E}} \left[\max_{\theta\in\Theta_{\bar{G},h}} \bar{q}_\theta(S_h, A_h) - \min_{\theta\in\Theta_{\bar{G},h}} \bar{q}_\theta(S_h, A_h)\right] \leq (H - h + 1)\tilde{\eta} + 4HC_{\text{conc}}\check{\epsilon}\beta . \quad (50)$$

We are now ready to state the final result. Noting that $h \in [H]$ was arbitrary, by combining Eqs. (45) and (50), we have that, under event $\mathcal{E}_1 \cap \mathcal{E}_2 \cap \mathcal{E}_3$, for all $h \in [H]$, it holds that

$$\frac{1}{n} \sum_{i\in[n]} \left(\max_{\theta\in\Theta_{\bar{G},h}} \bar{q}_\theta(s_h^i, a_h^i) - \min_{\theta\in\Theta_{\bar{G},h}} \bar{q}_\theta(s_h^i, a_h^i)\right)$$

$$\leq \frac{H}{\sqrt{n}} \sqrt{\log\left(\frac{6H|C_\xi^{\mathbf{G}}|}{\delta}\right)} + 2L_\xi + (H - h + 1)\tilde{\eta} + 4HC_{\text{conc}}\check{\epsilon}\beta$$

$$= \frac{H}{\sqrt{n}} \sqrt{\log\left(\frac{6H(1 + 2L_2/\xi)^{dHd_0}}{\delta}\right)} + 24\sqrt{2d}H^2 L_1\xi\alpha^{-1}\left(2\sqrt{n}L_1\tilde{L}_2/(H^{3/2}d)\right)^H$$

$$\qquad + (H - h + 1)\tilde{\eta} + 4HC_{\text{conc}}\check{\epsilon}\beta$$

$$\leq \frac{H}{\sqrt{n}} \sqrt{dH^2 d_0 \log\left(1 + 96\sqrt{n}\sqrt{2d}H^2 L_1 L_2\alpha^{-1}\sqrt{n}L_1\tilde{L}_2/(H^{3/2}d)\right) + \log\left(\frac{6H}{\delta}\right)} + \frac{1}{\sqrt{n}}$$

$$\qquad + (H - h + 1)\tilde{\eta} + 4HC_{\text{conc}}\check{\epsilon}\beta$$

$$= \bar{\epsilon} = \tilde{\mathcal{O}}\left(\frac{H^2 d}{\sqrt{n}} + \frac{1}{\sqrt{n}} + \frac{H^{5/2}d}{\sqrt{n}} + \frac{C_{\text{conc}}H^{5/2}d^2}{\sqrt{n}}\right) = \tilde{\mathcal{O}}\left(\frac{C_{\text{conc}}H^{5/2}d^2}{\sqrt{n}}\right) . \quad (51)$$

The first equality holds by plugging in the values of $|C_\xi^{\mathbf{G}}|, L_\xi$, as defined in Eqs. (33) and (34). The second equality holds by setting $\xi^{-1} = 24\sqrt{n}\sqrt{2d}H^2 L_1\alpha^{-1}\left(2\sqrt{n}L_1\tilde{L}_2/(H^{3/2}d)\right)^H$. The last two equalities hold by plugging in parameter values according to Appendix A.

Noticing that this is exactly the condition (Eq. (14)) in Optimization Problem 1 that needs to be satisfied by any feasible solution, we conclude that, under event $\mathcal{E}_1 \cap \mathcal{E}_2 \cap \mathcal{E}_3$, the true guess $\bar{G}$ is a feasible solution to Optimization Problem 1.

$\square$

## G.1 Proof of Lemma G.1

*Proof.* To prove Eq. (46) we will use induction. The base case is when $h = H + 1$, for which $\bar{v}_{\theta_{H+1}}(s) = v^{\pi_{\bar{G}}^\star}(s)$ for all $s \in \mathcal{S}_{H+1}$. This holds, since for all $(s, a) \in \mathcal{S}_{H+1} \times \mathcal{A}$, $\bar{q}_{\theta_{H+1}}(s, a) = 0$ for all $\theta_{H+1} \in \Theta_{\bar{G}, H+1}$, by the definition of $\Theta_{\bar{G}, H+1}$ (Eq. (13)), and $v^{\pi_{\bar{G}}^\star}(s) = 0$, by Eq. (1), since $\mathcal{S}_{H+1} = \{s_\top\}$.

Now, we show the inductive step. Let $h \in [H]$ be arbitrary. Assume that Eq. (46) holds for any $t \in [h+1 : H+1]$. We prove that Eq. (46) holds for $h$. Let $(s, a) \in \mathcal{S}_h \times \mathcal{A}, \theta_h \in \Theta_{\bar{G}, h}$. Then,

$$\left| \bar{q}_{\theta_h}(s, a) - q^{\pi_{\bar{G}}^\star}(s, a) \right| \leq \left| \min \left\{ H, \langle \phi(s, a), \theta_h \rangle - q^{\pi_{\bar{G}}^\star}(s, a) \right\} \right|$$
$$= \min \left\{ H, \left| \langle \phi(s, a), \theta_h \rangle - q^{\pi_{\bar{G}}^\star}(s, a) \right| \right\}, \qquad (52)$$

where the last equality holds since $\left( \langle \phi(s, a), \theta_h \rangle - q^{\pi_{\bar{G}}^\star}(s, a) \right) \in [-H, H]$. We will focus on bounding $\left| \langle \phi(s, a), \theta_h \rangle - q^{\pi_{\bar{G}}^\star}(s, a) \right|$. By the definition of the set $\Theta_{\bar{G}, h}$ (Eq. (13)) we know that there exists a $\hat{\theta}_h \in \hat{\Theta}_{\bar{G}, h}$ such that $\left\| \theta_h - \hat{\theta}_h \right\|_{X_h} \leq \beta$. Thus, by the Cauchy-Schwarz inequality, we have that

$$\left| \left\langle \phi(s, a), \theta_h - \hat{\theta}_h \right\rangle \right| \leq \|\phi(s, a)\|_{X_h^{-1}} \left\| \theta_h - \hat{\theta}_h \right\|_{X_h} \leq \beta \|\phi(s, a)\|_{X_h^{-1}}.$$

This implies that

$$\left| \langle \phi(s, a), \theta_h \rangle - q^{\pi_{\bar{G}}^\star}(s, a) \right| \leq \left| \left\langle \phi(s, a), \hat{\theta}_h \right\rangle - q^{\pi_{\bar{G}}^\star}(s, a) \right| + \beta \|\phi(s, a)\|_{X_h^{-1}}. \qquad (53)$$

Since we know $\hat{\theta}_h \in \hat{\Theta}_{\bar{G}, h}$, by the definition of $\hat{\Theta}_{\bar{G}, h}$ (Eq. (12)), there exists a $\theta_{h+1:H+1} \in \Theta_{\bar{G}, h+1} \times \cdots \times \Theta_{\bar{G}, H+1}$, such that

$$\hat{\theta}_h = X_h^{-1} \sum_{j \in [n]} \phi_h^j \mathop{\mathbb{E}}_{\tau \sim F_{\bar{G}, h+1}^j} \left[ r_{h:\tau-1}^j + \bar{v}_{\theta_{h+1:H+1}} \left( s_\tau^j \right) \right].$$

By Lemma 4.2, we know there exists a parameter $\rho_h^{\pi^0}(f) \in \mathcal{B}(\tilde{L}_2)$, such that for all $(s, a) \in \mathcal{S}_h \times \mathcal{A}$,

$$\left| \mathop{\mathbb{E}}_{\mathrm{Traj} \sim \mathbb{P}_{\pi^0, s, a}} \mathop{\mathbb{E}}_{\tau \sim F_{\bar{G}, \mathrm{Traj}, h+1}} \left[ R_{h:\tau-1} + \bar{v}_{\theta_{h+1:H+1}}(S_\tau) \right] - \left\langle \phi(s, a), \rho_h^{\pi^0} \left( \bar{v}_{\theta_{h+1:H+1}} \right) \right\rangle \right| \leq \tilde{\eta}. \qquad (54)$$

Let $\rho_h^{\pi^0}$ be as defined above. Next, we will show a bound on $\left| \left\langle \phi(s, a), \hat{\theta}_h - \rho_h^{\pi^0} \left( \bar{v}_{\theta_{h+1:H+1}} \right) \right\rangle \right|$ and $\left| \left\langle \phi(s, a), \rho_h^{\pi^0} \left( \bar{v}_{\theta_{h+1:H+1}} \right) \right\rangle - q^{\pi_{\bar{G}}^\star}(s, a) \right|$, which together will give us a bound on $\left| \left\langle \phi(s, a), \hat{\theta}_h \right\rangle - q^{\pi_{\bar{G}}^\star}(s, a) \right|$, as desired. The following result gives us a bound on $\left\| \hat{\theta}_h - \rho_h^{\pi^0} \left( \bar{v}_{\theta_{h+1:H+1}} \right) \right\|_{X_h}$.

**Lemma G.2.** *There is an event $\mathcal{E}_1$, which occurs with probability at least $1 - \delta/3$, such that under event $\mathcal{E}_1$, for all $G \in \mathbf{G}$ for all $h \in [H]$, and for all $\hat{\theta}_h \in \hat{\Theta}_{G, h}$, it holds that*

$$\left\| \hat{\theta}_h - \theta_h^\star \right\|_{X_h} \leq \beta,$$

*where*

$$\hat{\theta}_h = X_h^{-1} \sum_{j \in [n]} \phi_h^j \mathop{\mathbb{E}}_{\tau \sim F_{G, h+1}^j} \left[ r_{h:\tau-1}^j + \bar{v}_{\theta_{h+1:H+1}} \left( s_\tau^j \right) \right] \quad \text{for some } \theta_{h+1:H+1} \in \Theta_{G, h+1} \times \cdots \times \Theta_{G, H+1},$$

*and $\theta_h^\star \in \mathcal{B}(\tilde{L}_2)$ is such that, for all $(s, a) \in \mathcal{S}_h \times \mathcal{A}$, it satisfies*

$$\left| \mathop{\mathbb{E}}_{\mathrm{Traj} \sim \mathbb{P}_{\pi^0, s, a}} \mathop{\mathbb{E}}_{\tau \sim F_{G, \mathrm{Traj}, h+1}} \left[ R_{h:\tau-1} + \bar{v}_{\theta_{h+1:H+1}}(S_\tau) \right] - \langle \phi(s, a), \theta_h^\star \rangle \right| \leq \tilde{\eta}.$$

*Proof.* Fix $G \in \mathbf{G}, h \in [H]$, and $\hat{\theta}_h \in \hat{\Theta}_{G,h}$, such that

$$\hat{\theta}_h = X_h^{-1} \sum_{j \in [n]} \phi_h^j \mathop{\mathbb{E}}_{\tau \sim F_{G,h+1}^j} \left[ r_{h:\tau-1}^j + \bar{v}_{\theta_{h+1:H+1}} \left( s_\tau^j \right) \right] \quad \text{for some } \theta_{h+1:H+1} \in \Theta_{G,h+1} \times \cdots \times \Theta_{G,H+1} \,.$$

Fix $\theta_h^\star \in \mathcal{B}(\tilde{L}_2)$, such that, for all $(s,a) \in \mathcal{S}_h \times \mathcal{A}$, it satisfies

$$\left| \mathop{\mathbb{E}}_{\text{Traj} \sim \mathbb{P}_{\pi^0,s,a}} \mathop{\mathbb{E}}_{\tau \sim F_{G,\text{Traj},h+1}} \left[ R_{h:\tau-1} + \bar{v}_{\theta_{h+1:H+1}}(S_\tau) \right] - \langle \phi(s,a), \theta_h^\star \rangle \right| \le \tilde{\eta} \,. \tag{55}$$

We would like to make use of Lemma H.1 to bound $\left\| \hat{\theta}_h - \theta_h^\star \right\|_{X_h}$. To do so, we map the terms used in the Lemma H.1 to our terms as follows

$$n = n, \lambda = \lambda, \theta_\star = \theta_h^\star, V = X_h, \hat{\theta} = \hat{\theta}_h, A = \left( \phi_h^j \right)_{j \in [n]},$$

$$Y = \tilde{Y} + \Delta = \left( \left\langle \phi_h^j, \theta_h^\star \right\rangle \right)_{j \in [n]} + \gamma + \Delta = \left( \mathop{\mathbb{E}}_{\tau \sim F_{G,h+1}^j} \left[ r_{h:\tau-1}^j + \bar{v}_{\theta_{h+1:H+1}} \left( s_\tau^j \right) \right] \right)_{j \in [n]},$$

$$\gamma = \left( \mathop{\mathbb{E}}_{\tau \sim F_{G,h+1}^j} \left[ r_{h:\tau-1}^j + \bar{v}_{\theta_{h+1:H+1}} \left( s_\tau^j \right) \right] - \mathop{\mathbb{E}}_{\text{Traj} \sim \mathbb{P}_{\pi^0,s_h^j,a_h^j}} \mathop{\mathbb{E}}_{\tau \sim F_{G,\text{Traj},h+1}} \left[ R_{h:\tau-1} + \bar{v}_{\theta_{h+1:H+1}}(S_\tau) \right] \right)_{j \in [n]},$$

$$\Delta = \left( \mathop{\mathbb{E}}_{\text{Traj} \sim \mathbb{P}_{\pi^0,s_h^j,a_h^j}} \mathop{\mathbb{E}}_{\tau \sim F_{G,\text{Traj},h+1}} \left[ R_{h:\tau-1} + \bar{v}_{\theta_{h+1:H+1}}(S_\tau) \right] - \left\langle \phi_h^j, \theta_h^\star \right\rangle \right)_{j \in [n]},$$

$$\iota = \sum_{j \in [n]} \phi_h^j \left( \mathop{\mathbb{E}}_{\tau \sim F_{G,h+1}^j} \left[ r_{h:\tau-1}^j + \bar{v}_{\theta_{h+1:H+1}} \left( s_\tau^j \right) \right] - \mathop{\mathbb{E}}_{\text{Traj} \sim \mathbb{P}_{\pi^0,s_h^j,a_h^j}} \mathop{\mathbb{E}}_{\tau \sim F_{G,\text{Traj},h+1}} \left[ R_{h:\tau-1} + \bar{v}_{\theta_{h+1:H+1}}(S_\tau) \right] \right).$$

With the definitions as above, applying Lemma H.1 we get

$$\left\| \hat{\theta}_h - \theta_h^\star \right\|_{X_h} \le \sqrt{\lambda} \| \theta_h^\star \|_2 + \| \Delta \|_\infty \sqrt{n} + \| \iota \|_{X_h^{-1}} \,. \tag{56}$$

The first term can be bounded by $H^{3/2} d$, by noting that $\| \theta_h^\star \|_2 \le \tilde{L}_2$, and setting

$$\sqrt{\lambda} = H^{3/2} d / \tilde{L}_2 \,. \tag{57}$$

The second term can be bounded by $\tilde{\eta} \sqrt{n}$, by using Eq. (55).

To bound the third term let the event $\mathcal{E}_1$ be as defined in the proof of Lemma H.2, which occurs with probability at least $1 - \delta/3$. Then, under event $\mathcal{E}_1$, the third term in Eq. (56) can be bounded by $\bar{\beta}$ (Eq. (31)), by applying Lemma H.2, since $\theta_{h+1:H+1} \in \Theta_{G,h+1} \times \cdots \times \Theta_{G,H+1} \subset \mathcal{B}(\tilde{L}_2)^{H-h+1}$.

Plugging the three bounds above back into Eq. (56) we get that under event $\mathcal{E}_1$, it holds that

$$\left\| \hat{\theta}_h - \theta_h^\star \right\|_{X_h} \le H^{3/2} d + \tilde{\eta} \sqrt{n} + \bar{\beta}$$

$$= \beta = \tilde{\mathcal{O}} \left( H^{3/2} d \right). \tag{58}$$

The last equality holds by setting

$$\eta \le \frac{H^{3/2} d}{\sqrt{n}(10 H^2 d_0 / \alpha + 1)} \implies \tilde{\eta} \le H^{3/2} d / \sqrt{n} \,, \tag{59}$$

and the values of $n, \bar{\beta}$ are set according to Appendix A. $\qquad\square$

We return back to the proof of Lemma G.1. Let $\mathcal{E}_1$ be as defined in the proof of Lemma G.2. For the remainder of the proof, operate under event $\mathcal{E}_1$. By Lemma G.2 (with $\rho_h^{\pi^0} \left( \bar{v}_{\theta_{h+1:H+1}} \right) = \theta_h^\star$), we have that

$$\left\| \hat{\theta}_h - \rho_h^{\pi^0} \left( \bar{v}_{\theta_{h+1:H+1}} \right) \right\|_{X_h} \le \beta \,. \tag{60}$$

Returning to $\left|\left\langle\phi(s,a),\hat{\theta}_h-\rho_h^{\pi^0}\big(\bar{v}_{\theta_{h+1:H+1}}\big)\right\rangle\right|$, we can now bound it as follows.

$$\left|\left\langle\phi(s,a),\hat{\theta}_h-\rho_h^{\pi^0}\big(\bar{v}_{\theta_{h+1:H+1}}\big)\right\rangle\right|\leq\|\phi(s,a)\|_{X_h^{-1}}\left\|\hat{\theta}_h-\rho_h^{\pi^0}\big(\bar{v}_{\theta_{h+1:H+1}}\big)\right\|_{X_h}\leq\beta\|\phi(s,a)\|_{X_h^{-1}}\,.$$
(61)

The first inequality used the Cauchy-Schwarz inequality. The second inequality used Eq. (60). Now, we bound $\left|\left\langle\phi(s,a),\rho_h^{\pi^0}\big(\bar{v}_{\theta_{h+1:H+1}}\big)\right\rangle-q^{\pi_{\bar{G}}^{\star}}(s,a)\right|$, by making use of Eq. (54), to get that

$$\left|\left\langle\phi(s,a),\rho_h^{\pi^0}\big(\bar{v}_{\theta_{h+1:H+1}}\big)\right\rangle-q^{\pi_{\bar{G}}^{\star}}(s,a)\right|$$

$$\leq\left|\underset{\mathrm{Traj}\sim\mathbb{P}_{\pi^0,s,a}}{\mathbb{E}}\,\underset{\tau\sim F_{\bar{G},\mathrm{Traj},h+1}}{\mathbb{E}}\big[R_{h:\tau-1}+\bar{v}_{\theta_{h+1:H+1}}(S_\tau)\big]-q^{\pi_{\bar{G}}^{\star}}(s,a)\right|+\tilde{\eta}$$

$$=\left|\underset{\mathrm{Traj}\sim\mathbb{P}_{\pi^0,s,a}}{\mathbb{E}}\,\underset{\tau\sim F_{\bar{G},\mathrm{Traj},h+1}}{\mathbb{E}}\Big[R_{h:\tau-1}+\max_{a'\in\mathcal{A}}\bar{q}_{\theta_{\mathrm{stage}(S_\tau)}}(S_\tau,a')-\max_{a'\in\mathcal{A}}q^{\pi_{\bar{G}}^{\star}}(S_\tau,a')+\max_{a'\in\mathcal{A}}q^{\pi_{\bar{G}}^{\star}}(S_\tau,a')\Big]-q^{\pi_{\bar{G}}^{\star}}(s,a)\right|+\tilde{\eta}$$

$$\leq\left|\underset{\mathrm{Traj}\sim\mathbb{P}_{\pi^0,s,a}}{\mathbb{E}}\,\underset{\tau\sim F_{\bar{G},\mathrm{Traj},h+1}}{\mathbb{E}}\Big[R_{h:\tau-1}+\max_{a'\in\mathcal{A}}q^{\pi_{\bar{G}}^{\star}}(S_\tau,a')\Big]-q^{\pi_{\bar{G}}^{\star}}(s,a)\right|$$

$$+\left|\underset{\mathrm{Traj}\sim\mathbb{P}_{\pi^0,s,a}}{\mathbb{E}}\,\underset{\tau\sim F_{\bar{G},\mathrm{Traj},h+1}}{\mathbb{E}}\max_{a'\in\mathcal{A}}\Big(\bar{q}_{\theta_{\mathrm{stage}(S_\tau)}}(S_\tau,a')-q^{\pi_{\bar{G}}^{\star}}(S_\tau,a')\Big)\right|+\tilde{\eta}\,.$$
(62)

The equality used the definition of $\bar{v}$ (Eq. (10)). To bound the first term in Eq. (62) notice that by the definition of $\pi_{\bar{G}}^{\star}$ (Eq. (15)), $\arg\max_{a'\in\mathcal{A}}q^{\pi_{\bar{G}}^{\star}}(S_\tau,a')$ is exactly the action $\pi_{\bar{G}}^{\star}$ would take at the stopping stage $\tau$, and that the distribution of Traj under policy $\pi^0$ until stopping stage $\tau$ is same as the distribution of Traj under policy $\pi_{\bar{G}}^{\star}$ until stopping stage $\tau$. This gives that

$$\left|\underset{\mathrm{Traj}\sim\mathbb{P}_{\pi^0,s,a}}{\mathbb{E}}\,\underset{\tau\sim F_{\bar{G},\mathrm{Traj},h+1}}{\mathbb{E}}\Big[R_{h:\tau-1}+\max_{a'\in\mathcal{A}}q^{\pi_{\bar{G}}^{\star}}(S_\tau,a')\Big]-q^{\pi_{\bar{G}}^{\star}}(s,a)\right|=\left|q^{\pi_{\bar{G}}^{\star}}(s,a)-q^{\pi_{\bar{G}}^{\star}}(s,a)\right|=0\,.$$

To bound the second term in Eq. (62) we can use the inductive hypothesis (Eq. (46)). Defining notation that will be needed for the below display, for any $(s',a')\in\mathcal{S}\times\mathcal{A}$ we will write $\mathrm{Traj}'\sim\mathbb{P}_{\tilde{\pi},s,a}$ to have the usual definition $\mathrm{Traj}'=(s',a',R'_h,\ldots,S'_{H+1},A'_{H+1},R'_{H+1})$, except with a superscript $(\cdot)'$ added to all of the random elements. Then, by letting $a_\tau=\arg\max_{a'\in\mathcal{A}}\big(\bar{q}_{\theta_{\mathrm{stage}(S_\tau)}}(S_\tau,a')-q^{\pi_{\bar{G}}^{\star}}(S_\tau,a')\big)$, applying the inductive hypothesis, and a triangle inequality, we have that

$$\left|\underset{\mathrm{Traj}\sim\mathbb{P}_{\pi^0,s,a}}{\mathbb{E}}\,\underset{\tau\sim F_{\bar{G},\mathrm{Traj},h+1}}{\mathbb{E}}\max_{a'\in\mathcal{A}}\Big(\bar{q}_{\theta_{\mathrm{stage}(S_\tau)}}(S_\tau,a')-q^{\pi_{\bar{G}}^{\star}}(S_\tau,a')\Big)\right|$$

$$\leq\left|\underset{\mathrm{Traj}\sim\mathbb{P}_{\pi^0,s,a}}{\mathbb{E}}\,\underset{\tau\sim F_{\bar{G},\mathrm{Traj},h+1}}{\mathbb{E}}\Big[2\beta\underset{\mathrm{Traj}'\sim\mathbb{P}_{\tilde{\pi},S_\tau,a_\tau}}{\mathbb{E}}\sum_{t=\tau}^{H}\min\Big\{1,\|\phi(S'_t,A'_t)\|_{X_t^{-1}}\Big\}\right|+\left|\underset{\mathrm{Traj}\sim\mathbb{P}_{\pi^0,s,a}}{\mathbb{E}}\,\underset{\tau\sim F_{\bar{G},\mathrm{Traj},h+1}}{\mathbb{E}}(H-\tau+1)\tilde{\eta}\Big]\right|\,.$$

We bound each of the terms in the expectation separately. For the first term, we first recall the definition of $g^{\bar{\pi}}$ (Eq. (48)) and upper bound the term inside the expectation as follows, which will help us relate things to $\bar{\pi}$ as we shall see soon.

$$\underset{\mathrm{Traj}'\sim\mathbb{P}_{\bar{\pi},S_\tau,a_\tau}}{\mathbb{E}}\sum_{t=\tau}^{H}\min\Big\{1,\|\phi(S'_t,A'_t)\|_{X_t^{-1}}\Big\}=g^{\bar{\pi}}(S_\tau,a_\tau)\leq\max_{a'\in\mathcal{A}}g^{\bar{\pi}}(S_\tau,a')\,.$$

Along with the above result, notice that by the definition of $\bar{\pi}$ (Eq. (47)), $\arg\max_{a'\in\mathcal{A}}g^{\bar{\pi}}(S_\tau,a')$ is exactly the action $\bar{\pi}$ would take at the stopping stage $\tau$, and that the distribution of Traj under policy $\pi^0$ until stopping stage $\tau$ is same as the distribution of Traj under policy $\bar{\pi}$ until stopping stage $\tau$. Thus,

$$2\beta\left|\underset{\mathrm{Traj}\sim\mathbb{P}_{\pi^0,s,a}}{\mathbb{E}}\,\underset{\tau\sim F_{\bar{G},\mathrm{Traj},h+1}}{\mathbb{E}}\max_{a'\in\mathcal{A}}g^{\bar{\pi}}(S_\tau,a')\right|$$

$$\leq2\beta\underset{\mathrm{Traj}\sim\mathbb{P}_{\pi^0,s,a}}{\mathbb{E}}\,\underset{\tau\sim F_{\bar{G},\mathrm{Traj},h+1}}{\mathbb{E}}\Big[\sum_{t=h+1}^{\tau-1}\min\Big\{1,\|\phi(S_t,A_t)\|_{X_t^{-1}}\Big\}+\max_{a'\in\mathcal{A}}g^{\bar{\pi}}(S_\tau,a')\Big]$$

$$=2\beta g^{\bar{\pi}}(s,a)=2\beta\underset{\mathrm{Traj}\sim\mathbb{P}_{\bar{\pi},s,a}}{\mathbb{E}}\sum_{t=h+1}^{H}\min\Big\{1,\|\phi(S_t,A_t)\|_{X_t^{-1}}\Big\}\,.$$

For the second term, since $\tau \geq h+1$ where $\tau \sim F_{\bar{G}, \text{Traj}, h+1}$, it holds that

$$\left| \mathbb{E}_{\text{Traj} \sim \mathbb{P}_{\pi^0, s, a}} \mathbb{E}_{\tau \sim F_{\bar{G}, \text{Traj}, h+1}} (H - \tau + 1)\tilde{\eta} \right| \leq (H - h)\tilde{\eta}.$$

Plugging the above two bounds into Eq. (62), we get that

$$\left| \left\langle \phi(s, a), \rho_h^{\pi^0}\left(\bar{v}_{\theta_{h+1:H+1}}\right) \right\rangle - q^{\pi_{\bar{G}}^\star}(s, a) \right| \leq 2\beta \mathbb{E}_{\text{Traj} \sim \mathbb{P}_{\bar{\pi}, s, a}} \sum_{t=h+1}^{H} \min\left\{ 1, \|\phi(S_t, A_t)\|_{X_t^{-1}} \right\} + (H - h + 1)\tilde{\eta}. \tag{63}$$

Combining Eqs. (52), (53), (61) and (63), we get that

$$\left| \bar{q}_{\theta_h}(s, a) - q^{\pi_{\bar{G}}^\star}(s, a) \right|$$

$$\leq \min\left\{ H, 2\beta \|\phi(s, a)\|_{X_h^{-1}} + 2\beta \mathbb{E}_{\text{Traj} \sim \mathbb{P}_{\bar{\pi}, s, a}} \sum_{t=h+1}^{H} \min\left\{ 1, \|\phi(S_t, A_t)\|_{X_t^{-1}} \right\} + (H - h + 1)\tilde{\eta} \right\}$$

$$\leq 2\beta \min\left\{ 1, \|\phi(s, a)\|_{X_h^{-1}} \right\} + 2\beta \mathbb{E}_{\text{Traj} \sim \mathbb{P}_{\bar{\pi}, s, a}} \sum_{t=h+1}^{H} \min\left\{ 1, \|\phi(S_t, A_t)\|_{X_t^{-1}} \right\} + (H - h + 1)\tilde{\eta}$$

$$\leq 2\beta \mathbb{E}_{\text{Traj} \sim \mathbb{P}_{\bar{\pi}, s, a}} \sum_{t=h}^{H} \min\left\{ 1, \|\phi(S_t, A_t)\|_{X_t^{-1}} \right\} + (H - h + 1)\tilde{\eta}.$$

The second inequality used that $\beta \geq H$ and that $\min(a, b + c) \leq \min(a, b) + c$ for $a, b, c \geq 0$. $\quad\square$

**Lemma G.3.** *There is an event $\mathcal{E}_3$, that occurs with probability at least $1 - \delta/3$, such that under event $\mathcal{E}_3$, for all $h \in [H]$, it holds that*

$$\mathbb{E}_{(S,A) \sim \mu_h} \min\left\{ 1, \|\phi(S, A)\|_{X_h^{-1}} \right\} \leq \check{\epsilon}.$$

*where $\check{\epsilon}$ is defined in Eq. (28).*

*Proof.* First we will show two useful results (namely Eq. (64) and Eq. (66)) that are needed in the proof. Let

$$\mathbb{X} = \left\{ M \in \mathbb{R}^{d \times d} : M \text{ is positive semi-definite, and } \lambda_{\max}(M) \leq 1/\lambda \right\}.$$

Notice that any $X \in \mathbb{X}$ can be written as $X = \sum_{i=1}^{d} x_i x_i^\top$ with $x_i \in \mathcal{B}(1/\lambda)$ for all $i \in [d]$. By Lemma J.2, we know there exists a set $C_\xi \subset \mathcal{B}(a), a, \xi > 0$ with $|C_\xi| = (1 + 2a/\xi)^d$ such that for any $x \in \mathcal{B}(a)$ there exists a $y \in C_\xi$ such that $\|x - y\|_2 \leq \xi$. Define the set

$$\mathbb{Y} = \left\{ \sum_{i=1}^{d} y_i y_i^\top : y_i \in C_\xi \text{ for all } i \in [d] \right\},$$

with $|\mathbb{Y}| = (1 + 2/(\lambda\xi))^{d^2}$. Then, for any $X = \sum_{i=1}^{d} x_i x_i^\top \in \mathbb{X}$ there exists a $Y = \sum_{i=1}^{d} y_i y_i^\top \in \mathbb{Y}$, such that $\|x_i - y_i\|_2 \leq \xi$ for all $i \in [d]$. Let $X, Y$ be as we just defined them. Then, writing $\|\cdot\|_{\text{op}}$

for the operator norm,

$$
\begin{aligned}
\|X - Y\|_{\mathrm{op}} &= \left\| \sum_{i=1}^{d} x_i x_i^\top - y_i y_i^\top \right\|_{\mathrm{op}} \\
&= \left\| \sum_{i=1}^{d} (x_i - y_i)(x_i - y_i)^\top + y_i(x_i - y_i)^\top + (x_i - y_i)y_i^\top \right\|_{\mathrm{op}} \\
&\leq \sum_{i=1}^{d} \left\| (x_i - y_i)(x_i - y_i)^\top \right\|_{\mathrm{op}} + \left\| y_i(x_i - y_i)^\top \right\|_{\mathrm{op}} + \left\| (x_i - y_i)y_i^\top \right\|_{\mathrm{op}} \\
&\leq \sum_{i=1}^{d} \|(x_i - y_i)\|_2 \|(x_i - y_i)\|_2 + \|y_i\|_2 \|(x_i - y_i)\|_2 + \|(x_i - y_i)\|_2 \|y_i\|_2 \\
&\leq \sum_{i=1}^{d} \xi^2 + \frac{2\xi}{\sqrt{\lambda}} = d\xi^2 + \frac{2d\xi}{\sqrt{\lambda}} \, .
\end{aligned}
$$

Then, for any $u \in \mathcal{B}(L_1)$

$$
\left| \|u\|_X^2 - \|u\|_Y^2 \right| = \left| u^\top (X - Y)u \right| \leq \|u\|_2^2 \|X - Y\|_{\mathrm{op}} \leq L_1^2 \left( d\xi^2 + \frac{2d\xi}{\sqrt{\lambda}} \right),
$$

which implies that (since for non-negative $a, b$, $\sqrt{a+b} \leq \sqrt{a} + \sqrt{b}$)

$$
\|u\|_X \leq \sqrt{\|u\|_Y^2 + L_1^2 \left( d\xi^2 + \frac{2d\xi}{\sqrt{\lambda}} \right)} \leq \|u\|_Y + \sqrt{L_1^2 \left( d\xi^2 + \frac{2d\xi}{\sqrt{\lambda}} \right)},
$$

$$
\|u\|_Y \leq \sqrt{\|u\|_X^2 + L_1^2 \left( d\xi^2 + \frac{2d\xi}{\sqrt{\lambda}} \right)} \leq \|u\|_X + \sqrt{L_1^2 \left( d\xi^2 + \frac{2d\xi}{\sqrt{\lambda}} \right)} \, .
$$

Thus, for any $u \in \mathcal{B}(L_1)$

$$
\left| \|u\|_X - \|u\|_Y \right| \leq L_1 \sqrt{d\xi^2 + \frac{2d\xi}{\sqrt{\lambda}}} \, . \tag{64}
$$

Eq. (64) is the first useful result that we alluded to at the beginning of the proof.

Now, we will show the second useful result. For any $Y \in \mathbb{Y}$ and $h \in [H]$, define the event

$$
\mathcal{E}_3^{Y,h} = \left\{ \left| \mathbb{E}_{(S,A) \sim \mu_h} \min\{1, \|\phi(S,A)\|_Y\} - \frac{1}{n} \sum_{j \in [n]} \min\left\{1, \left\|\phi_h^j\right\|_Y\right\} \right| \leq \frac{1}{\sqrt{n}} \sqrt{\log\left( \frac{6H|\mathbb{Y}|}{\delta} \right)} \right\} .
$$

Since $\min\{1, \|u\|_Y\} \in [0, 1]$ for all $u \in \mathcal{B}(L_1), Y \in \mathbb{Y}$, we can use Hoeffding's inequality (Lemma J.1) to get that, for any $Y \in \mathbb{Y}, h \in [H]$, event $\mathcal{E}_3^{Y,h}$ occurs with probability at least $1 - \delta/(3H|\mathbb{Y}|)$. Let

$$
\mathcal{E}_3 = \bigcap_{Y \in \mathbb{Y}, h \in [H]} \mathcal{E}_3^{Y,h} . \tag{65}
$$

Then, by applying a union bound over $Y, h$ we have that the event $\mathcal{E}_3$ occurs with probability at least $1 - \delta/3$, and under event $\mathcal{E}_3$, for all $Y \in \mathbb{Y}, h \in [H]$, it holds that

$$
\mathbb{E}_{(S,A) \sim \mu_h} \min\{1, \|\phi(S,A)\|_Y\} \leq \frac{1}{n} \sum_{j \in [n]} \min\left\{1, \left\|\phi_h^j\right\|_Y\right\} + \frac{1}{\sqrt{n}} \sqrt{\log\left( \frac{6H|\mathbb{Y}|}{\delta} \right)} . \tag{66}
$$

Eq. (66) is the second useful result that we alluded to at the beginning of the proof.

Now, we turn to proving Lemma G.3. Let $h \in [H]$. Let $X \in \mathbb{X}$, and select $Y \in \mathbb{Y}$ such that, for any $u \in \mathcal{B}(L_1)$

$$\left| \|u\|_X - \|u\|_Y \right| \leq L_1 \sqrt{d\xi^2 + \frac{2d\xi}{\sqrt{\lambda}}} \,, \tag{67}$$

which we know exists by Eq. (64). By using Eq. (67) we get that

$$\mathop{\mathbb{E}}_{(S,A)\sim\mu_h} \min\{1, \|\phi(S,A)\|_X\} \leq \mathop{\mathbb{E}}_{(S,A)\sim\mu_h} \min\{1, \|\phi(S,A)\|_Y\} + L_1\sqrt{d\xi^2 + \frac{2d\xi}{\sqrt{\lambda}}} \,. \tag{68}$$

To bound the first term on the RHS in Eq. (68) we can use Eq. (66), to get that under event $\mathcal{E}_3$

$$\mathop{\mathbb{E}}_{(S,A)\sim\mu_h} \min\{1, \|\phi(S,A)\|_Y\} \leq \frac{1}{n}\sum_{j\in[n]} \min\left\{1, \left\|\phi_h^j\right\|_Y\right\} + \frac{1}{\sqrt{n}}\sqrt{\log\left(\frac{6H|\mathbb{Y}|}{\delta}\right)} \,. \tag{69}$$

We can bound the first term on the RHS of Eq. (69), by again using Eq. (67), to get that

$$\frac{1}{n}\sum_{j\in[n]} \min\left\{1, \left\|\phi_h^j\right\|_Y\right\} \leq \frac{1}{n}\sum_{j\in[n]} \min\left\{1, \left\|\phi_h^j\right\|_X\right\} + L_1\sqrt{d\xi^2 + \frac{2d\xi}{\sqrt{\lambda}}} \,. \tag{70}$$

Then, by Jensen's inequality we have that

$$\frac{1}{n}\sum_{j\in[n]} \min\left\{1, \left\|\phi_h^j\right\|_X\right\} = \sqrt{\left(\frac{1}{n}\sum_{j\in[n]} \min\left\{1, \left\|\phi_h^j\right\|_X\right\}\right)^2} \leq \sqrt{\frac{1}{n}\sum_{j\in[n]} \min\left\{1, \left\|\phi_h^j\right\|_X^2\right\}} \,. \tag{71}$$

Putting Eqs. (68) to (71) together and noting that $X, h$ were arbitrary, we get that, under event $\mathcal{E}_3$, for any $X \in \mathbb{X}, h \in [H]$, it holds that

$$\mathop{\mathbb{E}}_{(S,A)\sim\mu_h} \min\{1, \|\phi(S,A)\|_X\} \leq \sqrt{\frac{1}{n}\sum_{j\in[n]} \min\left\{1, \left\|\phi_h^j\right\|_X^2\right\}} + 2L_1\sqrt{d\xi^2 + \frac{2d\xi}{\sqrt{\lambda}}} + \frac{1}{\sqrt{n}}\sqrt{\log\left(\frac{6H|\mathbb{Y}|}{\delta}\right)} \,.$$

We can now introduce $X_h$ and make use of the above result. Notice that for any $h \in [H], X_h^{-1} = (\lambda I + \sum_{j\in[n]} \phi_h^j(\phi_h^j)^\top)^{-1}$ is such that $\lambda_{\max}(X_h^{-1}) \leq 1/\lambda$, since $\lambda_{\min}(X_h) \geq \lambda$. Thus, $X_h^{-1} \in \mathbb{X}$. For any $t \in [n], h \in [H]$, define $X_{t,h} = \lambda I + \sum_{j\in[t]} \phi_h^j(\phi_h^j)^\top$, and notice that $X_{n,h}^{-1} = X_h^{-1}$, and that $X_{t,h}^{-1} - X_{n,h}^{-1}$ is positive semidefinite. This implies that, for all $h \in [H]$

$$\frac{1}{n}\sum_{j\in[n]} \min\left\{1, \left\|\phi_h^j\right\|_{X_h^{-1}}^2\right\} \leq \frac{1}{n}\sum_{j\in[n]} \min\left\{1, \left\|\phi_h^j\right\|_{X_{j-1,h}^{-1}}^2\right\} \,.$$

Now, we can use the elliptical potential lemma (Lemma J.4), to conclude that, for all $h \in [H]$

$$\frac{1}{n}\sum_{j\in[n]} \min\left\{1, \left\|\phi_h^j\right\|_{X_{j-1,h}^{-1}}^2\right\} \leq \frac{2d}{n}\log\left(\frac{d\lambda + nL_1^2}{d\lambda}\right) \,.$$

Putting everything together, we get that, under event $\mathcal{E}_3$, for all $h \in [H]$, it holds that

$$\mathop{\mathbb{E}}_{(S,A)\sim\mu_h} \min\left\{1, \|\phi(S,A)\|_{X_h^{-1}}\right\}$$

$$\leq 2L_1\sqrt{d\xi^2 + \frac{2d\xi}{\sqrt{\lambda}}} + \frac{1}{\sqrt{n}}\sqrt{\log\left(\frac{6H|\mathbb{Y}|}{\delta}\right)} + \sqrt{\frac{2d}{n}\log\left(\frac{d\lambda + nL_1^2}{d\lambda}\right)}$$

$$\leq 2L_1\sqrt{d\xi^2 + \frac{2\xi\tilde{L}_2}{H^{3/2}}} + \frac{1}{\sqrt{n}}\sqrt{\log\left(\frac{3H(1 + 2\tilde{L}_2^2/\xi)^{d^2}}{\delta}\right)} + \sqrt{\frac{2d}{n}\log\left(\frac{d\lambda + nL_1^2}{d\lambda}\right)}$$

$$= \frac{\sqrt{d}}{\sqrt{n}} + \frac{1}{\sqrt{n}}\sqrt{d^2\log\left(1 + 16nL_1^2\tilde{L}_2^3\right) + \log\left(\frac{3H}{\delta}\right)} + \sqrt{\frac{2d}{n}\log\left(\frac{d\lambda + nL_1^2}{d\lambda}\right)}$$

$$= \check{\epsilon} = \tilde{\mathcal{O}}\left(d/\sqrt{n}\right) \,. \tag{72}$$

The second inequality used that $|\mathbb{Y}| = (1 + 2/(\lambda\xi))^{d^2}$. The first equality holds by setting $\xi^{-1} = 8\tilde{L}_2 L_1^2 n$. The last equality holds by plugging in parameter values according to Appendix A. $\qquad \square$

**Lemma G.4.** *If Assumption 3 holds, then for any non-negative function $f : \mathcal{S} \times \mathcal{A} \to [0, \infty)$, for any admissible distribution $\nu = (\nu_t)_{t \in [H]}$, and for any $h \in [H]$, it holds that*

$$\underset{(S,A) \sim \nu_h}{\mathbb{E}} f(S, A) \leq C_{\text{conc}} \underset{(S,A) \sim \mu_h}{\mathbb{E}} f(S, A) .$$

*Proof.* Let $f : \mathcal{S} \times \mathcal{A} \to [0, \infty)$ be any non-negative function. Let $h \in [H]$ be any stage. Then,

$$\begin{aligned}
\underset{(S,A) \sim \nu_h}{\mathbb{E}} f(S, A) &= \int_{z \in \mathcal{S}_h \times \mathcal{A}} f(z) \nu_h(z) dz \\
&= \int_{z \in \mathcal{S}_h \times \mathcal{A}} f(z) \frac{\nu_h(z)}{\mu_h(z)} \mu_h(z) dz \\
&\leq \int_{z \in \mathcal{S}_h \times \mathcal{A}} f(z) C_{\text{conc}} \mu_h(z) dz \\
&= C_{\text{conc}} \underset{(S,A) \sim \mu_h}{\mathbb{E}} f(S, A) ,
\end{aligned}$$

where the inequality holds by applying Assumption 3, and noting that $f$ is non-negative. This implies the desired result, since $f$ and $h$ were arbitrary. $\qquad \square$

# H    Lemmas Related to Least-squares

**Lemma H.1** (Least-squares Error Decomposition). *Let $\lambda > 0, \theta_\star \in \mathbb{R}^d$ and $n \in \mathbb{N}^+$, For all $k \in [n]$, let*

$$A_k \in \mathbb{R}^d, \; \gamma_k \in \mathbb{R}, \; \tilde{Y}_k = \langle A_k, \theta_\star \rangle + \gamma_k, \; Y_k = \tilde{Y}_k + \Delta_k,$$

$$V = \lambda I + \sum_{t=1}^n A_t A_t^\top, \; \hat{\theta} = V^{-1} \sum_{t=1}^n A_t Y_t, \; \iota = \sum_{t=1}^n A_t \gamma_t, \; \Delta = (\Delta_t)_{t \in [n]}.$$

*Then,*

$$\left\| \hat{\theta} - \theta_\star \right\|_V \leq \sqrt{\lambda} \|\theta_\star\|_2 + \|\Delta\|_\infty \sqrt{n} + \|\iota\|_{V^{-1}}.$$

*Proof.* We begin by decomposing the targets used in $\hat{\theta}$ as follows

$$\hat{\theta} = V^{-1} \sum_{t=1}^n A_t Y_t$$

$$= V^{-1} \sum_{t=1}^n A_t (\langle A_t, \theta_\star \rangle + \gamma_t + \Delta_t)$$

$$= \left( \lambda I + \sum_{t=1}^n A_t A_t^\top \right)^{-1} \left( \sum_{t=1}^n A_t A_t^\top \theta_\star + \lambda I \theta_\star - \lambda I \theta_\star \right) + V^{-1} \sum_{t=1}^n A_t (\gamma_t + \Delta_t)$$

$$= \theta_\star + \lambda V^{-1} \theta_\star + V^{-1} \sum_{t=1}^n A_t \gamma_t + V^{-1} \sum_{t=1}^n A_t \Delta_t.$$

Then, subtracting $\theta_\star$ from both sides and taking the matrix $V$ weighted norm of both sides gives us that

$$\left\| \hat{\theta} - \theta_\star \right\|_V = \left\| \lambda V^{-1} \theta_\star + V^{-1} \sum_{t=1}^n A_t \gamma_t + V^{-1} \sum_{t=1}^n A_t \Delta_t \right\|_V$$

$$\leq \lambda \|\theta_\star\|_{V^{-1}} + \left\| \sum_{t=1}^n A_t \gamma_t \right\|_{V^{-1}} + \left\| \sum_{t=1}^n A_t \Delta_t \right\|_{V^{-1}}$$

$$\leq \frac{\lambda}{\lambda_{\min}(V)} \|\theta_\star\|_2 + \|\iota\|_{V^{-1}} + \|\Delta\|_\infty \sqrt{n}$$

$$\leq \sqrt{\lambda} \|\theta_\star\|_2 + \|\iota\|_{V^{-1}} + \|\Delta\|_\infty \sqrt{n},$$

where the second inequality used that $\left\| V^{-1} \right\| \leq \lambda_{\max}(V^{-1}) = 1/\lambda_{\min}(V)$ and Lemma J.5 to bound $\|\sum_{t=1}^n A_t \Delta_t\|_{V^{-1}}$. $\qquad\square$

**Lemma H.2** (Least-squares Noise Bound). *There is an event $\mathcal{E}_1$, which occurs with probability at least $1 - \delta/3$, such that under event $\mathcal{E}_1$, for all $h \in [H]$, for all $G \in \mathbf{G}$, and $\theta_{h+1:H+1} \in \mathcal{B}(\tilde{L}_2)^{H-h+1}$, it holds that*

$$\left\| \sum_{j \in [n]} \phi_h^j \left( \underset{\tau \sim F_{G,h+1}^j}{\mathbb{E}} \left[ r_{h:\tau-1}^j + \bar{v}_{\theta_{h+1:H+1}} (s_\tau^j) \right] - \underset{\mathrm{Traj} \sim \mathbb{P}_{\pi^0, s_h^j, a_h^j}}{\mathbb{E}} \underset{\tau \sim F_{G,\mathrm{Traj},h+1}}{\mathbb{E}} \left[ R_{h:\tau-1} + \bar{v}_{\theta_{h+1:H+1}} (S_\tau) \right] \right) \right\|_{X_h^{-1}} \leq \bar{\beta},$$

*where $\bar{\beta}$ is defined in Eq. (31).*

*Proof.* We begin the proof by showing two useful results (namely Eq. (74) and Eq. (75)), which will be needed later in the proof.

By Lemma J.2, we know there exists a set $C_\xi \subset \mathcal{B}(a), a, \xi > 0$ with $|C_\xi| = (1 + 2a/\xi)^d$ such that for any $x \in \mathcal{B}(a)$ there exists a $y \in C_\xi$ such that $\|x - y\|_2 \leq \xi$. Define the set $C_\xi^\Theta =$

$\bigtimes_{h\in[2:H+1]} C_\xi \subset \mathcal{B}(\tilde{L}_2)^H$ with $|C_\xi^\Theta| = (1 + 2\tilde{L}_2/\xi))^{dH}$. Then, for any $\theta_{2:H+1} \in \mathcal{B}(\tilde{L}_2)^H$, there exists a $\tilde{\theta}_{2:H+1} \in C_\xi^\Theta$ such that

$$\left\|\theta_h - \tilde{\theta}_h\right\|_2 \leq \xi \quad \text{for all } h \in [2:H+1].$$

Which implies that for any $h \in [H], t \in [h+1:H+1], s \in \mathcal{S}_t$ and $\theta_{h+1:H+1}, \theta_{h+1:H+1}^\sim$ as defined above

$$
\begin{aligned}
\left|\bar{v}_{\theta_t}(s) - \bar{v}_{\theta_t^\sim}(s)\right| &\leq \left|\text{clip}_{[0,H]} \max_{a\in\mathcal{A}}\langle\phi(s,a),\theta_t\rangle - \text{clip}_{[0,H]} \max_{a\in\mathcal{A}}\langle\phi(s,a),\theta_t^\sim\rangle\right| \\
&\leq \left|\max_{a\in\mathcal{A}}\langle\phi(s,a),\theta_t - \theta_t^\sim\rangle\right| \\
&\leq \max_{a\in\mathcal{A}}\|\phi(s,a)\|_2\|\theta_t - \theta_t^\sim\|_2 \\
&\leq L_1\xi.
\end{aligned}
\tag{73}
$$

Let $\xi > 0$. Combining Eq. (73) with Lemma I.4, we get that there exists a set $C_\xi^{\mathbf{G}} \times C_\xi^\Theta \subset \mathbf{G} \times \mathcal{B}(\tilde{L}_2)^H$ with $|C_\xi^{\mathbf{G}} \times C_\xi^\Theta| \leq (1+2L_2\tilde{L}_2/\xi))^{dH(d_0+1)}$ such that, for any $(G, \theta_{2:H+1}) \in \mathbf{G}\times\mathcal{B}(\tilde{L}_2)^H$, there exists a $(\tilde{G}, \theta_{2:H+1}^\sim) \in C_\xi^{\mathbf{G}} \times C_\xi^\Theta$ such that, for any $h \in [H]$, for any $u \in [h]$ and trajectory $\text{traj} = (s_t, a_t, r_t)_{t\in[u,H+1]}$ it holds that

$$
\begin{aligned}
&\left|\mathop{\mathbb{E}}_{\tau\sim F_{G,\text{traj},h+1}}\left[r_{h:\tau-1} + \bar{v}_{\theta_{h+1:H+1}}(s_\tau)\right] - \mathop{\mathbb{E}}_{\tau\sim F_{\tilde{G},\text{traj},h+1}}\left[r_{h:\tau-1} + \bar{v}_{\theta_{h+1:H+1}^\sim}(s_\tau)\right]\right| \\
&\leq (H-h+1)6\sqrt{2d}HL_1\xi/\alpha + \sum_{t=h}^H L_1\xi \\
&= (H-h+1)7\sqrt{2d}HL_1\xi/\alpha.
\end{aligned}
\tag{74}
$$

Eq. (74) is the first useful result we alluded to at the beginning of the proof.

Now, we show the second useful result, which is a bound under a high probability event. For any $(\tilde{G}, \theta_{2:H+1}^\sim) \in C_\xi^{\mathbf{G}} \times C_\xi^\Theta$ and $h \in [H]$ define the event

$$
\mathcal{E}_1^{\tilde{G},\theta^\sim,h} = \left\{\left\|\iota_{\tilde{G},\theta^\sim,h}\right\|_{X_h^{-1}} \leq \sqrt{2H^2\log\left(\frac{3H|C_\xi^{\mathbf{G}} \times C_\xi^\Theta|}{\delta}\right) + \log\left(\sqrt{\frac{\det(X_h)}{\det(\lambda I)}}\right)}\right\}
$$

where $\iota_{\tilde{G},\theta^\sim,h} = \sum_{j\in[n]} \phi_h^j\left(\mathop{\mathbb{E}}_{\tau\sim F_{\tilde{G},h+1}^j}\left[r_{h:\tau-1}^j + \bar{v}_{\theta_{h+1:H+1}^\sim}(s_\tau^j)\right] - \mathop{\mathbb{E}}_{\text{Traj}\sim\mathbb{P}_{\pi^0,s_h^j,a_h^j}}\mathop{\mathbb{E}}_{\tau\sim F_{\tilde{G},\text{Traj},h+1}}\left[R_{h:\tau-1} + \bar{v}_{\theta_{h+1:H+1}^\sim}(S_\tau)\right]\right)$.

Notice that $\iota_{\tilde{G},\theta^\sim,h}$ is $H$-subgaussian. Thus, we can use Theorem 1 from [Abbasi-Yadkori et al., 2011] to get that the event $\mathcal{E}_1^{\tilde{G},\theta^\sim,h}$ occurs with probability at least $1 - \delta/(3H|C_\xi^{\mathbf{G}} \times C_\xi^\Theta|)$. Define

$$
\mathcal{E}_1 = \bigcap_{(\tilde{G},\theta^\sim)\in C_\xi^{\mathbf{G}}\times C_\xi^\Theta, h\in[H]} \mathcal{E}_1^{\tilde{G},\theta^\sim,h}.
$$

Then, by applying a union bound over $\tilde{G}, \theta^\sim, h$ we have that the event $\mathcal{E}_1$ occurs with probability at least $1 - \delta/3$, and under event $\mathcal{E}_1$, for all $(\tilde{G}, \theta^\sim) \in C_\xi^{\mathbf{G}} \times C_\xi^\Theta, h \in [H]$, it holds that

$$
\begin{aligned}
&\left\|\sum_{j\in[n]} \phi_h^j\left(\mathop{\mathbb{E}}_{\tau\sim F_{\tilde{G},h+1}^j}\left[r_{h:\tau-1}^j + \bar{v}_{\theta_{h+1:H+1}^\sim}(s_\tau^j)\right] - \mathop{\mathbb{E}}_{\text{Traj}\sim\mathbb{P}_{\pi^0,s_h^j,a_h^j}}\mathop{\mathbb{E}}_{\tau\sim F_{\tilde{G},\text{Traj},h+1}}\left[R_{h:\tau-1} + \bar{v}_{\theta_{h+1:H+1}^\sim}(S_\tau)\right]\right)\right\|_{X_h^{-1}} \\
&\leq \sqrt{2H^2\log\left(\frac{3H(1 + 2L_2\tilde{L}_2/\xi))^{dH(d_0+1)}}{\delta}\right) + \log\left(\sqrt{\frac{\det(X_h)}{\det(\lambda I)}}\right)}.
\end{aligned}
\tag{75}
$$

Now with all of the above results in hand, we turn to finally proving Lemma H.2. Let $G \in \mathbf{G}$ as in the lemma statement. Then, by Eq. (74) we know that there exists a $(\tilde{G}, \theta^{\sim}_{2:H+1}) \in C^{\mathbf{G}}_{\xi} \times C^{\Theta}_{\xi}$ such that, for any $h \in [H]$, for any $u \in [h]$ and trajectory $\text{traj} = (s_t, a_t, r_t)_{t \in [u:H+1]}$, it holds that

$$\left| \mathop{\mathbb{E}}_{\tau \sim F_{G,\text{traj},h+1}} \left[ r_{h:\tau-1} + \bar{v}_{\theta_{h+1:H+1}}(s_\tau) \right] - \mathop{\mathbb{E}}_{\tau \sim F_{\tilde{G},\text{traj},h+1}} \left[ r_{h:\tau-1} + \bar{v}_{\theta^{\sim}_{h+1:H+1}}(s_\tau) \right] \right| \le 7\sqrt{2d}H^2 L_1 \xi/\alpha \,.$$
(76)

Let $\tilde{G}, \theta^{\sim}_{2:H+1}$ be as defined above. Then, for any $h \in [H]$, by using the triangle inequality we can write

$$\left\| \sum_{j \in [n]} \phi_h^j \left( \mathop{\mathbb{E}}_{\tau \sim F^j_{G,h+1}} \left[ r^j_{h:\tau-1} + \bar{v}_{\theta_{h+1:H+1}}\left(s^j_\tau\right) \right] - \mathop{\mathbb{E}}_{\text{Traj} \sim \mathbb{P}_{\pi^0, s^j_h, a^j_h}} \mathop{\mathbb{E}}_{\tau \sim F_{G,\text{Traj},h+1}} \left[ R_{h:\tau-1} + \bar{v}_{\theta_{h+1:H+1}}(S_\tau) \right] \right) \right\|_{X_h^{-1}}$$

$$\le \left\| \sum_{j \in [n]} \phi_h^j \left( \mathop{\mathbb{E}}_{\tau \sim F^j_{\tilde{G},h+1}} \left[ r^j_{h:\tau-1} + \bar{v}_{\theta^{\sim}_{h+1:H+1}}\left(s^j_\tau\right) \right] - \mathop{\mathbb{E}}_{\text{Traj} \sim \mathbb{P}_{\pi^0, s^j_h, a^j_h}} \mathop{\mathbb{E}}_{\tau \sim F_{\tilde{G},\text{Traj},h+1}} \left[ R_{h:\tau-1} + \bar{v}_{\theta^{\sim}_{h+1:H+1}}(S_\tau) \right] \right) \right\|_{X_h^{-1}}$$

$$+ \left\| \sum_{j \in [n]} \phi_h^j \left( \mathop{\mathbb{E}}_{\tau \sim F^j_{G,h+1}} \left[ r^j_{h:\tau-1} + \bar{v}_{\theta_{h+1:H+1}}\left(s^j_\tau\right) \right] - \mathop{\mathbb{E}}_{\tau \sim F^j_{\tilde{G},h+1}} \left[ r^j_{h:\tau-1} + \bar{v}_{\theta^{\sim}_{h+1:H+1}}\left(s^j_\tau\right) \right] \right) \right\|_{X_h^{-1}}$$

$$+ \left\| \sum_{j \in [n]} \phi_h^j \left( \mathop{\mathbb{E}}_{\text{Traj} \sim \mathbb{P}_{\pi^0, s^j_h, a^j_h}} \left[ \mathop{\mathbb{E}}_{\tau \sim F_{\tilde{G},\text{Traj},h+1}} \left[ R_{h:\tau-1} + \bar{v}_{\theta^{\sim}_{h+1:H+1}}(S_\tau) \right] - \mathop{\mathbb{E}}_{\tau \sim F_{G,\text{Traj},h+1}} \left[ R_{h:\tau-1} + \bar{v}_{\theta_{h+1:H+1}}(S_\tau) \right] \right] \right) \right\|_{X_h^{-1}} \,.$$

The first term can be bounded by Eq. (75), if we are under event $\mathcal{E}_1$. For the second and third term we make use of Lemma J.5, which ensures that for any sequence $(b_j)_{j \in [n]}$ such that $|b_j| \le c \in \mathbb{R}$ the following holds

$$\left\| \sum_{j \in [n]} \phi_h^j b_j \right\|_{X_h^{-1}} \le c\sqrt{n} \,.$$

For the second term and third term the respective $b_j$ terms can be bounded by using Eq. (76), giving us $c = 7\sqrt{2d}H^2 L_1 \xi/\alpha$.

Putting the above three bounds together we have that under event $\mathcal{E}_1$, which occurs with probability at least $1 - \delta/3$, for all $h \in [H]$, for all $G \in \mathbf{G}$, and $\theta_{h+1:H+1} \in \mathcal{B}(\tilde{L}_2)^{H-h+1}$, it holds that

$$\left\| \sum_{j \in [n]} \phi_h^j \left( \mathop{\mathbb{E}}_{\tau \sim F^j_{G,h+1}} \left[ r^j_{h:\tau-1} + \bar{v}_{\theta_{h+1:H+1}}\left(s^j_\tau\right) \right] - \mathop{\mathbb{E}}_{\text{Traj} \sim \mathbb{P}_{\pi^0, s^j_h, a^j_h}} \mathop{\mathbb{E}}_{\tau \sim F_{G,\text{Traj},h+1}} \left[ R_{h:\tau-1} + \bar{v}_{\theta_{h+1:H+1}}(S_\tau) \right] \right) \right\|_{X_h^{-1}}$$

$$\le \sqrt{2H^2 \log\left( \frac{3H(1 + 2L_2\tilde{L}_2/\xi)^{dH(d_0+1)}}{\delta} \right) + \log\left( \sqrt{\frac{\det(X_h)}{\det(\lambda I)}} \right)} + 14\sqrt{n}\sqrt{2d}H^2 L_1 \xi/\alpha$$

$$= H\sqrt{2dH(d_0 + 1)\log\left(1 + 2L_2\tilde{L}_2/\xi\right) + \log(\det(X_h)) - d\log(\lambda) + \log\left(\frac{3H}{\delta}\right)} + 14\sqrt{n}\sqrt{2d}H^2 L_1 \xi/\alpha$$

$$\le H\sqrt{2dH(d_0 + 1)\log\left(1 + 2L_2\tilde{L}_2/\xi\right) + d\log(\lambda + nL_1^2/d) - d\log(\lambda) + \log\left(\frac{3H}{\delta}\right)} + 14\sqrt{n}\sqrt{2d}H^2 L_1 \xi/\alpha \,,$$

where the last inequality used the Determinant-Trace Inequality (see Lemma 10 in [Abbasi-Yadkori et al., 2011]). Setting $\xi^{-1} = 14\sqrt{n}\sqrt{2d}H^2 L_1 \alpha^{-1}$, we get that the above display is

$$\le H\sqrt{2dH(d_0 + 1)\log\left(1 + 28\sqrt{2d}H^2 L_2 \tilde{L}_2 L_1 \alpha^{-1}\right) + d\log(\lambda + nL_1^2/d) - d\log(\lambda) + \log\left(\frac{3H}{\delta}\right)} + 1$$

$$= \bar{\beta} = \tilde{\mathcal{O}}\left(H^{3/2}d\right).$$
(77)

The last equality holds by plugging in parameter values according to Appendix A. $\qquad \square$

# I Lemmas Related to Covering G

**Lemma I.1.** *There is an event $\mathcal{E}_2$, that occurs with probability at least $1 - \delta/3$, such that under event $\mathcal{E}_2$, for all $G \in \mathbf{G}$, and for all $h \in [H]$, it holds that*

$$
\left| \mathop{\mathbb{E}}_{(S,A)\sim\mu_h} \left[ \max_{\theta\in\Theta_{G,h}} \bar{q}_\theta(S,A) - \min_{\theta\in\Theta_{G,h}} \bar{q}_\theta(S,A) \right] - \frac{1}{n} \sum_{i\in[n]} \left( \max_{\theta\in\Theta_{G,h}} \bar{q}_\theta(s_h^i, a_h^i) - \min_{\theta\in\Theta_{G,h}} \bar{q}_\theta(s_h^i, a_h^i) \right) \right|
$$

$$
\leq \frac{H}{\sqrt{n}} \sqrt{\log\left( \frac{6H|C_\xi^{\mathbf{G}}|}{\delta} \right)} + 2L_\xi \, ,
$$

*where $|C_\xi^{\mathbf{G}}|$, $L_\xi$, are defined in Eqs. (33) and (34).*

*Proof.* Let $G \in \mathbf{G}$ be a feasible solution to Optimization Problem 1. By the first result in Lemma I.2, there exists a set $C_\xi^{\mathbf{G}} \subset \mathbf{G}$ such that, there exists a $\tilde{G} \in C_\xi^{\mathbf{G}}$ such that, for any $h \in [H], (s,a) \in \mathcal{S}_h \times \mathcal{A}$, it holds that

$$
\left| \left( \max_{\theta\in\Theta_{G,h}} \bar{q}_\theta(s,a) - \min_{\theta\in\Theta_{G,h}} \bar{q}_\theta(s,a) \right) - \left( \max_{\theta^\sim\in\Theta_{\tilde{G},h}} \bar{q}_{\theta^\sim}(s,a) - \min_{\theta^\sim\in\Theta_{\tilde{G},h}} \bar{q}_{\theta^\sim}(s,a) \right) \right| \leq L_\xi \, . \quad (78)
$$

Select $\tilde{G}$ as defined above. Let $h \in [H]$. Using Eq. (78), we know that

$$
\left| \mathop{\mathbb{E}}_{(S,A)\sim\mu_h} \left[ \max_{\theta\in\Theta_{G,h}} \bar{q}_\theta(S,A) - \min_{\theta\in\Theta_{G,h}} \bar{q}_\theta(S,A) \right] - \mathop{\mathbb{E}}_{(S,A)\sim\mu_h} \left[ \max_{\theta^\sim\in\Theta_{\tilde{G},h}} \bar{q}_{\theta^\sim}(S,A) - \min_{\theta\in\Theta_{\tilde{G},h}} \bar{q}_{\theta^\sim}(S,A) \right] \right| \leq L_\xi \, .
$$
$$
(79)
$$

To bound the second term in the absolute value of Eq. (79) to its empirical mean, we can use the second result in Lemma I.2, which gives us that under event $\mathcal{E}_2$

$$
\left| \mathop{\mathbb{E}}_{(S,A)\sim\mu_h} \left[ \max_{\theta^\sim\in\Theta_{\tilde{G},h}} \bar{q}_{\theta^\sim}(S,A) - \min_{\theta^\sim\in\Theta_{\tilde{G},h}} \bar{q}_{\theta^\sim}(S,A) \right] - \frac{1}{n} \sum_{i\in[n]} \left( \max_{\theta^\sim\in\Theta_{\tilde{G},h}} \bar{q}_{\theta^\sim}(s_h^i, a_h^i) - \min_{\theta^\sim\in\Theta_{\tilde{G},h}} \bar{q}_{\theta^\sim}(s_h^i, a_h^i) \right) \right|
$$

$$
\leq \frac{H}{\sqrt{n}} \sqrt{\log\left( \frac{6H|C_\xi^{\mathbf{G}}|}{\delta} \right)} \, . \quad\quad\quad (80)
$$

We can relate the $\tilde{G}$ in the second term of the absolute value in Eq. (80) back to $G$, by once again using Eq. (78), to get that

$$
\left| \frac{1}{n} \sum_{i\in[n]} \left( \max_{\theta^\sim\in\Theta_{\tilde{G},h}} \bar{q}_{\theta^\sim}(s_h^i, a_h^i) - \min_{\theta^\sim\in\Theta_{\tilde{G},h}} \bar{q}_{\theta^\sim}(s_h^i, a_h^i) \right) - \frac{1}{n} \sum_{i\in[n]} \left( \max_{\theta\in\Theta_{G,h}} \bar{q}_\theta(s_h^i, a_h^i) - \min_{\theta\in\Theta_{G,h}} \bar{q}_\theta(s_h^i, a_h^i) \right) \right|
$$

$$
\leq L_\xi \, . \quad\quad\quad (81)
$$

Putting together Eqs. (79) to (81), and noting that $h$ was arbitrary, gives the desired result. $\square$

**Lemma I.2.** *Let $\xi > 0$. There exists a set $C_\xi^{\mathbf{G}} \subset \mathbf{G}$ such that, for any $G \in \mathbf{G}$, there exists a $\tilde{G} \in C_\xi^{\mathbf{G}}$ such that, for any $h \in [H], (s,a) \in \mathcal{S}_h \times \mathcal{A}$, it holds that*

$$
\left| \left( \max_{\theta\in\Theta_{G,h}} \bar{q}_\theta(s,a) - \min_{\theta\in\Theta_{G,h}} \bar{q}_\theta(s,a) \right) - \left( \max_{\theta^\sim\in\Theta_{\tilde{G},h}} \bar{q}_{\theta^\sim}(s,a) - \min_{\theta^\sim\in\Theta_{\tilde{G},h}} \bar{q}_{\theta^\sim}(s,a) \right) \right| \leq L_\xi \, ,
$$

*where $|C_\xi^{\mathbf{G}}|$, $L_\xi$, are defined in Eqs. (33) and (34). Furthermore, there is an event $\mathcal{E}_2$, which occurs with probability at least $1 - \delta/3$, such that under event $\mathcal{E}_2$, for any $\tilde{G} \in C_\xi^{\mathbf{G}}$, and $h \in [H]$, it holds that*

$$
\left| \mathop{\mathbb{E}}_{(S,A)\sim\mu_h} \left[ \max_{\theta^\sim\in\Theta_{\tilde{G},h}} \bar{q}_{\theta^\sim}(S,A) - \min_{\theta^\sim\in\Theta_{\tilde{G},h}} \bar{q}_{\theta^\sim}(S,A) \right] - \frac{1}{n} \sum_{i\in[n]} \left( \max_{\theta^\sim\in\Theta_{\tilde{G},h}} \bar{q}_{\theta^\sim}(s_h^i, a_h^i) - \min_{\theta^\sim\in\Theta_{\tilde{G},h}} \bar{q}_{\theta^\sim}(s_h^i, a_h^i) \right) \right|
$$

$$
\leq \frac{H}{\sqrt{n}} \sqrt{\log\left( \frac{6H|C_\xi^{\mathbf{G}}|}{\delta} \right)} \, .
$$

*Proof.* Let $\xi > 0, \kappa_t \geq 0, \forall t \in [2 : H + 1]$. By Lemma I.4, there exists a set $C_\xi^{\mathbf{G}} \subset \mathbf{G}$ with $|C_\xi^{\mathbf{G}}| \leq (1 + 2L_2/\xi))^{dHd_0}$ such that, for any $G \in \mathbf{G}$, there exists a $\tilde{G} \in C_\xi^{\mathbf{G}}$ such that, for any $h \in [H]$, for any $\theta_{h+1:H+1}, \theta_{h+1:H+1}^{\sim} \in \mathcal{B}(\tilde{L}_2)^{H-h+1}$, such that for all $t \in [h + 1 : H + 1], s \in \mathcal{S}_t, |\bar{v}_{\theta_t}(s) - \bar{v}_{\theta_t^{\sim}}(s)| \leq \kappa_t$, and for any $j \in [n]$, it holds that

$$\left| \mathop{\mathbb{E}}_{\tau \sim F_{G,h+1}^j} \left[ r_{h:\tau-1} + \bar{v}_{\theta_{h+1:H+1}}(s_\tau) \right] - \mathop{\mathbb{E}}_{\tau \sim F_{\tilde{G},h+1}^j} \left[ r_{h:\tau-1} + \bar{v}_{\theta_{h+1:H+1}^{\sim}}(s_\tau) \right] \right|$$

$$\leq (H - h + 1)6\sqrt{2d}HL_1\xi/\alpha + \sum_{t=h}^{H} \kappa_{t+1} . \tag{82}$$

For the remainder of the proof let $G, \tilde{G}$ be as described above.

We first show the following intermediate result.

**Lemma I.3.** *For all $h \in [H]$, it holds that:*

1. *For any $\hat{\theta}_h \in \hat{\Theta}_{G,h}, \theta_h \in \Theta_{G,h}$, there exists $\hat{\theta}_h^{\sim} \in \hat{\Theta}_{\tilde{G},h}, \theta_h^{\sim} \in \Theta_{\tilde{G},h}$ such that*

$$\left\| \hat{\theta}_h - \hat{\theta}_h^{\sim} \right\|_{X_h} \leq c_h^\xi, \quad \|\theta_h - \theta_h^{\sim}\|_{X_h} \leq c_h^\xi . \tag{83}$$

2. *For any $\hat{\theta}_h^{\sim} \in \hat{\Theta}_{\tilde{G},h}, \theta_h^{\sim} \in \Theta_{\tilde{G},h}$, there exists $\hat{\theta}_h \in \hat{\Theta}_{G,h}, \theta_h \in \Theta_{G,h}$ such that*

$$\left\| \hat{\theta}_h - \hat{\theta}_h^{\sim} \right\|_{X_h} \leq c_h^\xi, \quad \|\theta_h - \theta_h^{\sim}\|_{X_h} \leq c_h^\xi , \tag{84}$$

*where,*

$$c_h^\xi = 6\sqrt{n}\sqrt{2d}H^2L_1\xi\alpha^{-1}\left(1 + \sqrt{n}L_1\tilde{L}_2/(H^{3/2}d)\right)^{H-h} . \tag{85}$$

*Proof.* **Proof of result** 1.**:** To show Eq. (83) we will use induction. The base case is when $h = H$, for which

$$\hat{\Theta}_{G,H} = \hat{\Theta}_{\tilde{G},H} = \left\{ X_H^{-1} \sum_{j \in [n]} \phi_H^j r_H^j \right\}, \implies \Theta_{G,H} = \Theta_{\tilde{G},H} .$$

Thus, for any $\hat{\theta}_H \in \hat{\Theta}_{G,H}, \theta_H \in \Theta_{G,H}$, select $\hat{\theta}_H^{\sim} = \hat{\theta}_H \in \hat{\Theta}_{\tilde{G},H}, \theta_H^{\sim} = \theta_H \in \Theta_{\tilde{G},H}$. Then

$$\left\| \hat{\theta}_H - \hat{\theta}_H^{\sim} \right\|_{X_H} \leq 0, \quad \|\theta_H - \theta_H^{\sim}\|_{X_H} \leq 0 .$$

Now, we show the inductive step. Let $h \in [H - 1]$ be arbitrary. Assume Eq. (83) holds for any $t \in [h + 1 : H]$. We prove that Eq. (83) also holds for $h$. Let $\hat{\theta}_h \in \hat{\Theta}_{G,h}$ be arbitrary. Notice that $\hat{\theta}_h$ must have the following form.

$$\hat{\theta}_h = X_h^{-1} \sum_{j \in [n]} \phi_h^j \mathop{\mathbb{E}}_{\tau \sim F_{G,h+1}^j} \left[ r_{h:\tau-1}^j + \bar{v}_{\theta_{h+1:H+1}}(s_\tau^j) \right] \in \hat{\Theta}_{G,h}, \text{ for some } \theta_{h+1:H+1} \in \Theta_{G,h+1} \times \cdots \times \Theta_{G,H+1} .$$

Select $\theta_{h+1:H+1}^{\sim} \in \Theta_{\tilde{G},h+1} \times \cdots \times \Theta_{\tilde{G},H+1}$ such that for all $t \in [h + 1 : H + 1]$

$$\|\theta_t - \theta_t^{\sim}\|_{X_t} \leq c_t^\xi , \tag{86}$$

which exists by the inductive hypothesis (Eq. (83)) and since $\Theta_{G,H+1} = \Theta_{\tilde{G},H+1} = \{\vec{0}\}$. Define

$$\hat{\theta}_h^{\sim} = X_h^{-1} \sum_{j \in [n]} \phi_h^j \mathop{\mathbb{E}}_{\tau \sim F_{\tilde{G},h+1}^j} \left[ r_{h:\tau-1}^j + \bar{v}_{\theta_{h+1:H+1}^{\sim}}(s_\tau^j) \right] \in \hat{\Theta}_{\tilde{G},h} .$$

Recall that we aim to bound $\left\|\hat{\theta}_h - \hat{\theta}_h^{\sim}\right\|_{X_h}$. Plugging in the expressions for $\hat{\theta}_h, \hat{\theta}_h^{\sim}$, as defined above, we get that

$$\left\|\hat{\theta}_h - \hat{\theta}_h^{\sim}\right\|_{X_h} = \left\|X_h^{-1} \sum_{j\in[n]} \phi_h^j \left( \mathop{\mathbb{E}}_{\tau\sim F_{G,h+1}^j}\left[r_{h:\tau-1}^j + \bar{v}_{\theta_{h+1:H+1}}(s_\tau^j)\right] - \mathop{\mathbb{E}}_{\tau\sim F_{\tilde{G},h+1}^j}\left[r_{h:\tau-1}^j + \bar{v}_{\theta_{h+1:H+1}^{\sim}}(s_\tau^j)\right] \right)\right\|_{X_h}$$

$$= \left\|X_h^{-1} \sum_{j\in[n]} \phi_h^j b_j\right\|_{X_h} = \left\|\sum_{j\in[n]} \phi_h^j b_j\right\|_{X_h^{-1}},$$

where in the second equality we let $b_j = \mathop{\mathbb{E}}_{\tau\sim F_{G,h+1}^j}\left[r_{h:\tau-1}^j + \bar{v}_{\theta_{h+1:H+1}}(s_\tau^j)\right] - \mathop{\mathbb{E}}_{\tau\sim F_{\tilde{G},h+1}^j}\left[r_{h:\tau-1}^j + \bar{v}_{\theta_{h+1:H+1}^{\sim}}(s_\tau^j)\right]$. To bound the above term we can make use of Lemma J.5, which ensures that for any sequence $(b_j)_{j\in[n]}$ such that $|b_j| \le a \in \mathbb{R}$ the following holds

$$\left\|\sum_{j\in[n]} \phi_h^j b_j\right\|_{X_h^{-1}} \le a\sqrt{n}\,.$$

Thus, we are left to bound $|b_j|$. To do so, we can make use of Eq. (82), which requires us to bound $\left|\bar{v}_{\theta_t}(s) - \bar{v}_{\theta_t^{\sim}}(s)\right|$ for all $t \in [h+1:H], s \in \mathcal{S}_t$, which can be done as follows.

$$\left|\bar{v}_{\theta_t}(s) - \bar{v}_{\theta_t^{\sim}}(s)\right| = \left|\mathrm{clip}_{[0,H]} \max_{a\in\mathcal{A}}\langle\phi(s,a),\theta_t\rangle - \mathrm{clip}_{[0,H]} \max_{a\in\mathcal{A}}\langle\phi(s,a),\theta_t^{\sim}\rangle\right|$$

$$\le \left|\max_{a\in\mathcal{A}}\langle\phi(s,a),\theta_t - \theta_t^{\sim}\rangle\right|$$

$$\le \max_{a\in\mathcal{A}}\|\phi(s,a)\|_{X_t^{-1}}\|\theta_t - \theta_t^{\sim}\|_{X_t}$$

$$\le \frac{L_1}{\sqrt{\lambda}}c_t^\xi = L_1\tilde{L}_2 c_t^\xi/(H^{3/2}d)\,. \tag{87}$$

The second inequality uses the Cauchy-Schwarz inequality. The third inequality used Eq. (22), that $\lambda_{\max}(X_h^{-1}) \le 1/\lambda$ (by definition of $X_h$ Eq. (11)), and Eq. (86). The last equality used that $\sqrt{\lambda} = H^{3/2}d/\tilde{L}_2$ (Eq. (25)). Plugging Eq. (87) into Eq. (82) we get that for any $j \in [n]$

$$|b_j| = \left|\mathop{\mathbb{E}}_{\tau\sim F_{G,h+1}^j}\left[r_{h:\tau-1} + \bar{v}_{\theta_{h+1:H+1}}(s_\tau)\right] - \mathop{\mathbb{E}}_{\tau\sim F_{\tilde{G},h+1}^j}\left[r_{h:\tau-1} + \bar{v}_{\theta_{h+1:H+1}^{\sim}}(s_\tau)\right]\right|$$

$$\le (H - h + 1)6\sqrt{2d}HL_1\xi/\alpha + \sum_{t=h}^{H} L_1\tilde{L}_2 c_{t+1}^\xi/(H^{3/2}d)\,.$$

Thus,

$$\left\|\hat{\theta}_h - \hat{\theta}_h^{\sim}\right\|_{X_h} = \left\|\sum_{j\in[n]} \phi_h^j b_j\right\|_{X_h^{-1}} \le \left(6\sqrt{2d}H^2L_1\xi\alpha^{-1} + \frac{L_1\tilde{L}_2}{H^{3/2}d}\sum_{t=h}^{H} c_{t+1}^\xi\right)\sqrt{n}$$

$$= 6\sqrt{n}\sqrt{2d}H^2L_1\xi\alpha^{-1} + \sqrt{n}\frac{L_1\tilde{L}_2}{H^{3/2}d}\sum_{t=h}^{H} c_{t+1}^\xi$$

$$= 6\sqrt{n}\sqrt{2d}H^2L_1\xi\alpha^{-1}\left(1 + \sqrt{n}L_1\tilde{L}_2/(H^{3/2}d)\right)^{H-h}$$

$$= c_h^\xi\,.$$

To see why the second last equality is true, let $x = 6\sqrt{n}\sqrt{2d}H^2L_1\xi\alpha^{-1}$ and $y = \sqrt{n}L_1\tilde{L}_2/(H^{3/2}d)$. Then, for any $t \in [h:H], c_t^\xi = x(1+y)^{H-t}$ (by Eq. (85)) and

$$6\sqrt{n}\sqrt{2d}H^2L_1\xi\alpha^{-1} + \sqrt{n}\frac{L_1\tilde{L}_2}{H^{3/2}d}\sum_{t=h}^{H} c_{t+1}^\xi = x + y\sum_{t=h+1}^{H} x(1+y)^{H-t}\,.$$

Notice that the sum can be rewritten as a finite geometric series.

$$\sum_{t=h+1}^{H} x(1+y)^{H-t} = x \sum_{k=0}^{H-h-1} (1+y)^k = x\frac{(1+y)^{H-h}-1}{y}\,.$$

Thus,

$$x + y\sum_{t=h+1}^{H} x(1+y)^{H-t} = x + y\cdot x\frac{(1+y)^{H-h}-1}{y} = x(1+y)^{H-h} = c_h^\xi\,.$$

This proves the first result in Eq. (83).

Next, we show the second result in Eq. (83). Let $\theta_h \in \Theta_{G,h}$. By the definition of the set $\Theta_{G,h}$, there exists a $\hat{\theta}_h \in \hat{\Theta}_{G,h}$ such that $\left\|\theta_h - \hat{\theta}_h\right\|_{X_h} \le \beta$. Then, by the first result in Eq. (83) (which we have shown holds for $h$ above), there exists a $\hat{\theta}_{\tilde{h}} \in \hat{\Theta}_{\tilde{G},h}$, such that $\left\|\hat{\theta}_h - \hat{\theta}_{\tilde{h}}\right\|_{X_h} \le c_h^\xi$. Let $\hat{\theta}_h, \hat{\theta}_{\tilde{h}}$ be as defined above, and select $\theta_{\tilde{h}} = \theta_h - \hat{\theta}_h + \hat{\theta}_{\tilde{h}}$, which is an element of $\Theta_{\tilde{G},h}$ since

$$\left\|\theta_{\tilde{h}} - \hat{\theta}_{\tilde{h}}\right\|_{X_h} = \left\|\theta_h - \hat{\theta}_h + \hat{\theta}_{\tilde{h}} - \hat{\theta}_{\tilde{h}}\right\|_{X_h} = \left\|\theta_h - \hat{\theta}_h\right\|_{X_h} \le \beta\,.$$

Then,

$$\|\theta_h - \theta_{\tilde{h}}\|_{X_h} = \left\|\theta_h - \theta_h - \hat{\theta}_h + \hat{\theta}_{\tilde{h}}\right\|_{X_h} = \left\|\hat{\theta}_{\tilde{h}} - \hat{\theta}_h\right\|_{X_h} \le c_h^\xi\,.$$

This completes the proof of the second result in Eq. (83).

**Proof of result** 2.**:** The proof is identical to that of the proof of result 1., except swapping the roles of $\hat{\theta}_h, \theta_h, G$ and $\hat{\theta}_{\tilde{h}}, \theta_{\tilde{h}}, \tilde{G}$. $\qquad\square$

Now, with the results of Eqs. (83) and (84) in hand, we return to proving Lemma I.2. For any $h \in [H]$ let $\theta_h = \arg\max_{\theta \in \Theta_{G,h}} \langle \phi(s,a), \theta\rangle$ then, by Eq. (83), there exists a $\theta_{\tilde{h}} \in \Theta_{\tilde{G},h}$ such that $\|\theta_h - \theta_{\tilde{h}}\|_{X_t} \le c_h^\xi$. This gives that, for all $h \in [H], (s,a) \in \mathcal{S}_h \times \mathcal{A}$

$$\max_{\theta \in \Theta_{G,h}} \langle \phi(s,a), \theta\rangle - \max_{\theta^\sim \in \Theta_{\tilde{G},h}} \langle \phi(s,a), \theta^\sim\rangle = \langle \phi(s,a), \theta_h\rangle - \langle \phi(s,a), \theta_{\tilde{h}}\rangle + \langle \phi(s,a), \theta_{\tilde{h}}\rangle - \max_{\theta^\sim \in \Theta_{\tilde{G},h}} \langle \phi(s,a), \theta^\sim\rangle$$

$$\le \|\phi(s,a)\|_{X_h^{-1}}\|\theta_h - \theta_{\tilde{h}}\|_{X_h}$$

$$\le \frac{L_1}{\sqrt{\lambda}}c_h^\xi = L_1\tilde{L}_2 c_h^\xi/(H^{3/2}d)\,.$$

The second inequality used Eq. (22), that $\lambda_{\max}(X_h^{-1}) \le 1/\lambda$ (by definition of $X_h$ Eq. (11)), and Eq. (86). The last equality used that $\sqrt{\lambda} = H^{3/2}d/\tilde{L}_2$ (Eq. (25)). Now, for the other direction, using similar steps as above, for any $h \in [H]$, let $\theta_{\tilde{h}} = \arg\max_{\theta^\sim \in \Theta_{\tilde{G},h}} \langle \phi(s,a), \theta^\sim\rangle$ then, by Eq. (84), there exists a $\theta_h \in \Theta_{G,h}$ such that $\|\theta_h - \theta_{\tilde{h}}\|_{X_h} \le c_h^\xi$. This gives that, for all $h \in [H], (s,a) \in \mathcal{S}_h \times \mathcal{A}$

$$\max_{\theta^\sim \in \Theta_{\tilde{G},h}} \langle \phi(s,a), \theta^\sim\rangle - \max_{\theta \in \Theta_{G,h}} \langle \phi(s,a), \theta\rangle = \langle \phi(s,a), \theta_{\tilde{h}}\rangle - \langle \phi(s,a), \theta_h\rangle + \langle \phi(s,a), \theta_h\rangle - \max_{\theta \in \Theta_{G,h}} \langle \phi(s,a), \theta\rangle$$

$$\le \|\phi(s,a)\|_{X_h^{-1}}\|\theta_{\tilde{h}} - \theta_h\|_{X_h}$$

$$\le \frac{L_1}{\sqrt{\lambda}}c_h^\xi = L_1\tilde{L}_2 c_h^\xi/(H^{3/2}d)\,.$$

The above two results together imply that, for all $h \in [H], (s,a) \in \mathcal{S}_h \times \mathcal{A}$

$$\left|\max_{\theta \in \Theta_{G,h}} \langle \phi(s,a), \theta\rangle - \max_{\theta^\sim \in \Theta_{\tilde{G},h}} \langle \phi(s,a), \theta^\sim\rangle\right| \le L_1\tilde{L}_2 c_h^\xi/(H^{3/2}d)\,.$$

Following the same steps as above for $\min$ we can get that, for all $h \in [H], (s,a) \in \mathcal{S}_h \times \mathcal{A}$

$$\left|\min_{\theta \in \Theta_{G,h}} \langle \phi(s,a), \theta\rangle - \min_{\theta^\sim \in \Theta_{\tilde{G},h}} \langle \phi(s,a), \theta^\sim\rangle\right| \le L_1\tilde{L}_2 c_h^\xi/(H^{3/2}d)\,.$$

The above two results together imply that, for all $h \in [H], (s,a) \in \mathcal{S}_h \times \mathcal{A}$,

$$\left| \left( \max_{\theta \in \Theta_{G,h}} \bar{q}_\theta(s,a) - \min_{\theta \in \Theta_{G,h}} \bar{q}_\theta(s,a) \right) - \left( \max_{\theta^\sim \in \Theta_{\tilde{G},h}} \bar{q}_{\theta^\sim}(s,a) - \min_{\theta^\sim \in \Theta_{\tilde{G},h}} \bar{q}_{\theta^\sim}(s,a) \right) \right| = L_\xi \,,$$

where (by recalling Eq. (85)),

$$L_\xi = 12\sqrt{2d}H^2 L_1 \xi \alpha^{-1} \left( 2\sqrt{n} L_1 \tilde{L}_2 / (H^{3/2}d) \right)^H \geq 2L_1 \tilde{L}_2 c_h^\xi / (H^{3/2}d) \,. \tag{88}$$

This concludes the proof of the first result in Lemma I.2.

Now we prove the second result in Lemma I.2. For any $\tilde{G} \in C_\xi^{\mathbf{G}}, h \in [H]$, define the event

$$\mathcal{E}_2^{\tilde{G},h} = \left\{ \left| \underset{(S,A) \sim \mu_h}{\mathbb{E}} \left[ \max_{\theta^\sim \in \Theta_{\tilde{G},h}} \bar{q}_{\theta^\sim}(S,A) - \min_{\theta^\sim \in \Theta_{\tilde{G},h}} \bar{q}_{\theta^\sim}(S,A) \right] \right. \right.$$

$$\left. \left. - \frac{1}{n} \sum_{i \in [n]} \left( \max_{\theta^\sim \in \Theta_{\tilde{G},h}} \bar{q}_{\theta^\sim}(s_h^i, a_h^i) - \min_{\theta^\sim \in \Theta_{\tilde{G},h}} \bar{q}_{\theta^\sim}(s_h^i, a_h^i) \right) \right| \leq \frac{H}{\sqrt{n}} \sqrt{\log\left( \frac{6H|C_\xi^{\mathbf{G}}|}{\delta} \right)} \right\} \,.$$

Then, since $\max_{\theta^\sim \in \Theta_{\tilde{G},h}} \bar{q}_{\theta^\sim}(s,a) - \min_{\theta^\sim \in \Theta_{\tilde{G},h}} \bar{q}_{\theta^\sim}(s,a) \in [0, H]$ for all $(s,a) \in \mathcal{S}_h \times \mathcal{A}$, by Hoeffding's inequality (Lemma J.1), we have that, for any $\tilde{G} \in C_\xi^{\mathbf{G}}, h \in [H]$, event $\mathcal{E}_2^{\tilde{G},h}$ occurs with probability at least $1 - \delta/(3H|C_\xi^{\mathbf{G}}|)$. Let

$$\mathcal{E}_2 = \bigcap_{\tilde{G} \in C_\xi^{\mathbf{G}}, h \in [H]} \mathcal{E}_2^{\tilde{G},h} \,. \tag{89}$$

Then, by applying a union bound over $\tilde{G}, h$ we have that the event $\mathcal{E}_2$ occurs with probability at least $1 - \delta/3$. $\qquad\square$

**Lemma I.4.** *Let $\xi > 0, \kappa_t \geq 0, \forall t \in [2 : H+1]$. Then, there exists a set $C_\xi^{\mathbf{G}} \subset \mathbf{G}$ with $|C_\xi^{\mathbf{G}}| \leq (1 + 2L_2/\xi))^{dHd_0}$ such that, for any $G \in \mathbf{G}$, there exists a $\tilde{G} \in C_\xi^{\mathbf{G}}$ such that, for any $h \in [H]$, for any $u \in [h]$ and trajectory* $\mathrm{traj} = (s_t, a_t, r_t)_{t \in [u, H+1]}$, *and for any $\theta_{h+1:H+1}, \theta^\sim_{h+1:H+1} \in \mathcal{B}(\tilde{L}_2)^{H-h+1}$ that are close in predictions, that is, such that for all $t \in [h+1 : H+1], s \in \mathcal{S}_t, \left| \bar{v}_{\theta_t}(s) - \bar{v}_{\theta^\sim_t}(s) \right| \leq \kappa_t$, it holds that*

$$\left| \underset{\tau \sim F_{G, \mathrm{traj}, h+1}}{\mathbb{E}} \left[ r_{h:\tau-1} + \bar{v}_{\theta_{h+1:H+1}}(s_\tau) \right] - \underset{\tau \sim F_{\tilde{G}, \mathrm{traj}, h+1}}{\mathbb{E}} \left[ r_{h:\tau-1} + \bar{v}_{\theta^\sim_{h+1:H+1}}(s_\tau) \right] \right|$$

$$\leq (H - h + 1)6\sqrt{2d}HL_1 \xi / \alpha + \sum_{t=h}^{H} \kappa_{t+1} \,.$$

*Proof.* Recall that

$$\mathbf{G} = \left\{ (G_h)_{h \in [2:H]} = (\vartheta_h^i)_{h \in [2:H], i \in [d_0]} : \text{ for all } h \in [2:H], i \in [d_0], \vartheta_h^i \in \mathcal{B}(L_2) \right\} \,.$$

By Lemma J.2, we know there exists a set $C_\xi \subset \mathcal{B}(a), a, \xi > 0$ with $|C_\xi| = (1 + 2a/\xi)^d$ such that for any $x \in \mathcal{B}(a)$ there exists a $y \in C_\xi$ such that $\|x - y\|_2 \leq \xi$. Define the set $C_\xi^{\mathbf{G}} = \times_{h \in [2:H], i \in [d_0]} C_\xi \subset \mathbf{G}$ with $|C_\xi^{\mathbf{G}}| \leq (1 + 2L_2/\xi))^{dHd_0}$. Then, for any $G = (\vartheta_h^i)_{h \in [2:H], i \in [d_0]} \in \mathbf{G}$, there exists a $\tilde{G} = (\tilde{\vartheta}_h^i)_{h \in [2:H], i \in [d_0]} \in C_\xi^{\mathbf{G}}$ such that

$$\left\| \vartheta_h^i - \tilde{\vartheta}_h^i \right\|_2 \leq \xi \quad \text{for all } h \in [2:H], i \in [d_0] \,.$$

Let $G, \tilde{G}$ be as defined above. Then, for all $s \in \mathcal{S} \setminus \mathcal{S}_1$

$$|\mathrm{range}^G(s) - \mathrm{range}^{\tilde{G}}(s)| = \left| \max_{k \in [d_0]} \max_{a,a' \in \mathcal{A}} \left\langle \phi(s,a,a'), \vartheta_{\mathrm{stage}(s)}^k \right\rangle - \max_{k \in [d_0]} \max_{a,a' \in \mathcal{A}} \left\langle \phi(s,a,a'), \tilde{\vartheta}_{\mathrm{stage}(s)}^k \right\rangle \right|$$

$$\leq \left| \max_{k \in [d_0]} \max_{a,a' \in \mathcal{A}} \left\langle \phi(s,a,a'), \vartheta_{\mathrm{stage}(s)}^k - \tilde{\vartheta}_{\mathrm{stage}(s)}^k \right\rangle \right|$$

$$\leq \max_{k \in [d_0]} \max_{a,a' \in \mathcal{A}} \|\phi(s,a,a')\|_2 \left\| \vartheta_{\mathrm{stage}(s)}^k - \tilde{\vartheta}_{\mathrm{stage}(s)}^k \right\|_2$$

$$\leq 2L_1 \xi \,,$$

and, since we have by definition (Eq. (6)) that $\omega_G$ is a smooth function in terms of range$^G$, we get that

$$|\omega_G(s) - \omega_{\tilde{G}}(s)| \leq \left| 2 - \frac{\sqrt{2d} \cdot \text{range}^G(s)}{\alpha} - \left( 2 - \frac{\sqrt{2d} \cdot \text{range}^{\tilde{G}}(s)}{\alpha} \right) \right| \leq \frac{2\sqrt{2d}L_1\xi}{\alpha} . \quad (90)$$

For all $h \in [H]$, let $\theta_{h+1:H+1}, \theta^{\sim}_{h+1:H+1} \in \mathcal{B}(\tilde{L}_2)^{H-h+1}$, such that for all $t \in [h+1 : H+1], s \in \mathcal{S}_t, |\bar{v}_{\theta_t}(s) - \bar{v}_{\theta^{\sim}_t}(s)| \leq \kappa_t$. Then, for any $h \in [H], u \in [h]$, and trajectory traj $= (s_t, a_t, r_t)_{t \in [u:H+1]}$, it holds that

$$\underset{\tau \sim F_{G,\text{traj},h+1}}{\mathbb{E}} \left[ r_{h:\tau-1} + \bar{v}_{\theta_{h+1:H+1}}(s_\tau) \right]$$
$$= r_h + (1 - \omega_G(s_{h+1}))\bar{v}_{\theta_{h+1}}(s_{h+1}) + \omega_G(s_{h+1})\big( r_{h+1} + (1 - \omega_G(s_{h+2}))v_{\theta_{h+2}}(s_{h+2}) \big) + \ldots$$
$$= r_h + (1 - \omega_G(s_{h+1}))\bar{v}_{\theta_{h+1}}(s_{h+1}) + \omega_G(s_{h+1}) \underset{\tau \sim F_{G,\text{traj},h+2}}{\mathbb{E}} \left[ r_{h+1:\tau-1} + \bar{v}_{\theta_{h+2:H+1}}(s_\tau) \right] . \quad (91)$$

Using similar steps to above it can be shown that

$$\underset{\tau \sim F_{\tilde{G},\text{traj},h+1}}{\mathbb{E}} \left[ r_{h:\tau-1} + \bar{v}_{\theta^{\sim}_{h+1:H+1}}(s_\tau) \right]$$
$$= r_h + (1 - \omega_{\tilde{G}}(s_{h+1}))\bar{v}_{\theta_{h+1}}(s_{h+1}) + \omega_{\tilde{G}}(s_{h+1}) \underset{\tau \sim F_{\tilde{G},\text{traj},h+2}}{\mathbb{E}} \left[ r_{h+1:\tau-1} + \bar{v}_{\theta^{\sim}_{h+2:H+1}}(s_\tau) \right] . \quad (92)$$

We claim that for any $h \in [H], u \in [h]$, and trajectory traj $= (s_t, a_t, r_t)_{t \in [u:H+1]}$

$$\left| \underset{\tau \sim F_{G,\text{traj},h+1}}{\mathbb{E}} \left[ r_{h:\tau-1} + \bar{v}_{\theta_{h+1:H+1}}(s_\tau) \right] - \underset{\tau \sim F_{\tilde{G},\text{traj},h+1}}{\mathbb{E}} \left[ r_{h:\tau-1} + \bar{v}_{\theta^{\sim}_{h+1:H+1}}(s_\tau) \right] \right|$$
$$\leq (H - h + 1)6\sqrt{2d}HL_1\xi/\alpha + \sum_{t=h}^{H} \kappa_{t+1} . \quad (93)$$

To show Eq. (93) we will use induction on $h$. The base case is when $h = H$, for which

$$\left| \underset{\tau \sim F_{G,\text{traj},H+1}}{\mathbb{E}} \left[ r_{H:\tau-1} + \bar{v}_{\theta_{H+1:H+1}}(s_\tau) \right] - \underset{\tau \sim F_{\tilde{G},\text{traj},H+1}}{\mathbb{E}} \left[ r_{H:\tau-1} + \bar{v}_{\theta^{\sim}_{H+1:H+1}}(s_\tau) \right] \right|$$
$$= \left| \underset{\tau \sim F_{G,\text{traj},H+1}}{\mathbb{E}} \left[ r_{H:\tau-1} \right] - \underset{\tau \sim F_{\tilde{G},\text{traj},H+1}}{\mathbb{E}} \left[ r_{H:\tau-1} \right] \right| = 0 ,$$

where the first equality holds since $\Theta_{G,H+1} = \Theta_{\tilde{G},H+1} = \{\vec{0}\}$ (defined in Eq. (13)). The second equality holds since for any $u \in [H+1]$, and trajectory traj $= (s_t, a_t, r_t)_{t \in [u:H+1]}, F_{G,\text{traj},H+1}(\tau = H+1) = F_{\tilde{G},\text{traj},H+1}(\tau = H+1) = 1$.

Now, we show the inductive step. Let $h \in [H-1]$ be arbitrary. Assume Eq. (93) holds for any $t \in [h+1, H]$. We prove that Eq. (93) also holds for $h$. Fix any $u \in [H+1]$, and trajectory traj $= (s_t, a_t, r_t)_{t \in [u:H+1]}$. To shorten notation let $E_{G,\theta,h} = \mathbb{E}_{\tau \sim F_{G,\text{traj},h+1}} \left[ r_{h:\tau-1} + \bar{v}_{\theta_{h+1:H+1}}(s_\tau) \right]$ (and similar for $E_{\tilde{G},\theta^{\sim},h}$). Then using the assumption that for all $u \in [2 : H+1], s \in \mathcal{S}_u, |\bar{v}_{\theta_u}(s) - \bar{v}_{\theta^{\sim}_u}(s)| \leq \kappa_u$ along with Eqs. (90) to (92) and noting that for any $h \in$

$[H], E_{G,\theta,h}, E_{\tilde{G},\theta\sim,h} \in [0, 2H], \bar{v}_{\theta_{h+1:H+1}}, \bar{v}_{\theta_{\widetilde{h+1}:H+1}} \in [0, H], \omega_G, \omega_{\tilde{G}} \in [0, 1]$, we have that

$$\left| \underset{\tau \sim F_{G,\text{traj},h+1}}{\mathbb{E}} \left[ r_{h:\tau-1} + \bar{v}_{\theta_{h+1:H+1}}(s_\tau) \right] - \underset{\tau \sim F_{\tilde{G},\text{traj},h+1}}{\mathbb{E}} \left[ r_{h:\tau-1} + \bar{v}_{\theta_{\widetilde{h+1}:H+1}}(s_\tau) \right] \right|$$

$$= \left| (1 - \omega_G(s_{h+1}))\bar{v}_{\theta_{h+1}}(s_{h+1}) - (1 - \omega_{\tilde{G}}(s_{h+1}))\bar{v}_{\theta_{\widetilde{h+1}}}(s_{h+1}) + \omega_G(s_{h+1})E_{G,\theta,h+1} - \omega_{\tilde{G}}(s_{h+1})E_{\tilde{G},\theta\sim,h+1} \right|$$

$$\leq \left| (1 - \omega_G(s_{h+1})) - (1 - \omega_{\tilde{G}}(s_{h+1})) \right| \left| \bar{v}_{\theta_{h+1}}(s_{h+1}) \right| + \left| 1 - \omega_{\tilde{G}}(s_{h+1}) \right| \left| \bar{v}_{\theta_{h+1}}(s_{h+1}) - \bar{v}_{\theta_{\widetilde{h+1}}}(s_{h+1}) \right|$$

$$\qquad + \left| \omega_G(s_{h+1}) - \omega_{\tilde{G}}(s_{h+1}) \right| \left| E_{G,\theta,h} \right| + \left| \omega_{\tilde{G}}(s_{h+1}) \right| \left| E_{G,\theta,h+1} - E_{\tilde{G},\theta\sim,h+1} \right|$$

$$\leq 2\sqrt{2d}HL_1\xi/\alpha + \kappa_{h+1} + 4\sqrt{2d}HL_1\xi/\alpha + \left| E_{G,\theta,h+1} - E_{\tilde{G},\theta\sim,h+1} \right|$$

$$\leq 6\sqrt{2d}HL_1\xi/\alpha + \kappa_{h+1} + (H-h)6\sqrt{2d}HL_1\xi/\alpha + \sum_{t=h+1}^{H+1} \kappa_{t+1}$$

$$\leq (H-h+1)6\sqrt{2d}HL_1\xi/\alpha + \sum_{t=h}^{H} \kappa_{t+1},$$

where the second last inequality holds by the inductive hypothesis. $\qquad \square$

# J Other Useful Results and Definitions

**Definition 3.** *A finite set $G \subset \mathbb{R}^d$ is the basis of a near-optimal design for a set $\Theta \subseteq \mathbb{R}^d$, if there exists a probability distribution $\rho$ over elements of $G$, such that for any $\theta \in \Theta$,*

$$\langle v, \theta \rangle = 0 \quad \text{for all } v \in \text{Ker}(V(G, \rho)), \text{ and} \tag{94}$$

$$\|\theta\|_{V(G,\rho)^\dagger}^2 \leq 2d, \tag{95}$$

$$\text{where } V(G, \rho) = \sum_{x \in G} \rho(x) x x^\top, \tag{96}$$

*where for some matrix $X$, $X^\dagger$ denotes the Moore-Penrose inverse of some, and $\text{Ker}(X)$ its kernel (or null space).*

**Lemma J.1** (Hoeffding's Inequality (Theorem 2 in [Hoeffding, 1994])). *Let $(X_i)_{i \in \mathbb{N}}$ be independent random variables such that $X_i \in [a, b]$ for some $a, b \in \mathbb{R}$, and let $S_n = \frac{1}{n} \sum_{i=1}^n X_i$. Then, with probability at least $1 - \zeta$ it holds that*

$$|\mathbb{E} S_n - S_n| \leq \frac{(b-a)}{\sqrt{n}} \sqrt{\log\left(\frac{2}{\zeta}\right)}.$$

**Lemma J.2** (Covering number of the Euclidean ball). *Let $a > 0, \epsilon > 0, d \geq 1$, and $\mathcal{B}_d(a) = \{x \in \mathbb{R}^d : \|x\|_2 \leq a\}$ denote the $d$-dimensional Euclidean ball of radius $a$ centered at the origin. The covering number of $\mathcal{B}_d(a)$ is upper bounded by $\left(1 + \frac{2a}{\epsilon}\right)^d$.*

*Proof.* Same as the proof of Corollary 4.2.13 in [Vershynin, 2018] with $\mathcal{B}(1)$ replaced with $\mathcal{B}(a)$. $\square$

**Lemma J.3** (Performance Difference Lemma (Lemma 3.2 in [Cai et al., 2020])). *For any policies $\pi, \bar{\pi}$, it holds that*

$$v^\pi(s_1) - v^{\bar{\pi}}(s_1) = \sum_{h=1}^H \mathop{\mathbb{E}}_{(S_h, A_h) \sim \mathbb{P}_{\pi, s_1}^h} \left( q^{\bar{\pi}}(S_h, A_h) - v^{\bar{\pi}}(S) \right).$$

**Lemma J.4** (Elliptical Potential Lemma (Lemma 19.4 in [Lattimore and Szepesvári, 2020])). *Let $V_0 \in \mathbb{R}^{d \times d}$ be positive definite and $a_1, \ldots, a_n \in \mathbb{R}^d$ be a sequence of vectors with $\|a_t\|_2 \leq L \leq \infty$ for all $t \in [n], V_t = V_0 + \sum_{s \leq t} a_s a_s^\top$. then,*

$$\sum_{t=1}^n \min\left\{1, \|a_t\|_{V_{t-1}^{-1}}^2\right\} \leq 2 \log\left(\frac{\det V_n}{\det V_0}\right) \leq 2d \log\left(\frac{\text{Tr } V_0 + nL^2}{d \det(V_0)^{1/d}}\right) \leq 2d \log\left(\frac{d\lambda + nL^2}{d\lambda}\right).$$

**Lemma J.5** (Projection Bound (Lemma 8 in [Zanette et al., 2020])). *Let $(a_i)_{i \in [n]}$ be any sequence of vectors in $\mathbb{R}^d$ and $(b_i)_{i \ in[n]}$ be any sequence of scalars such that $|b_i| \leq c$. For any $\lambda \geq 0$ and $k \in \mathbb{N}$ we have*

$$\left\|\sum_{i=1}^n a_i b_i\right\|_{(\sum_{i=1}^n a_i a_i^\top + \lambda I)^{-1}}^2 \leq nc^2.$$

