# OpenReview forum: "Trajectory Data Suffices for Statistically Efficient Learning in Offline RL with Linear $q^\pi$-Realizability and Concentrability"
_NeurIPS.cc/2024/Conference — NeurIPS 2024 poster_

### Official Review · Reviewer_sDBe · 2024-07-05

**Soundness:** 3
**Presentation:** 3
**Contribution:** 4
**Rating:** 7
**Confidence:** 3

**Summary:**

This paper proves that finite horizon offline RL under linear q^\pi realizability assumption can be solved efficiently (in terms of sample complexity) if the data are trajectories collected by a policy with bounded concentrability coefficient (that is, the density ratio between the state-action distribution induced by any policy and the data distribution is upper bounded). The result highlights the effect of trajectory data by sharply contrasting with the existing impossibility result where the data can be arbitrary state-action distribution with bounded concentrability coefficient.

**Strengths:**

To the best of my knowledge, this paper solves a long-standing open question of offline RL with linear q^\pi realizability. The observation about trajectory data versus general offline data is solid and interesting.

**Weaknesses:**

I don’t think I completely follows the proof sketch shown in Section 4 & 5 given the complexity of the proofs, and some of the proof intuitions could be made more explicit instead of referring to some lemmas in the appendix. For example:

-	What is the reason that $G=\bar{G}$ passes the condition (14)? Is it because the concentrability assumption plus Lemma 4.2 results in a tight confidence interval for the q-value?
-	One thing I do not follow is how the algorithm eliminates incorrect guesses $G$. If I understand correctly, Lemma 4.2 only shows that linear q^\pi-realizable MDPs with skips can be approximated by linear MDPs under the true guess. Then for an incorrect guess $G’$, is the modified MDPs still linear? If not, does optimization problem 1 still give some meaningful guarantees so that the algorithm can eliminate these cases?
In addition, since the optimization problem 1 is mostly built upon prior except for condition (14), if could be better to emphasize this new condition.

Minor issue:

-	In the statement of problem 1, n does not depend on \delta.

**Questions:**

Please see above.

**Limitations:**

The authors adequately addressed the limitations and potential negative societal impact.

---

> ### Author Rebuttal · Authors · 2024-08-07
>
> We thank the reviewer for their helpful feedback.
>
> > What is the reason that $G = \bar G$ passes the condition (14)? Is it because the concentrability assumption plus Lemma 4.2 results in a tight confidence interval for the q-value?
>
> Yes, it is precisely as you say “because the concentrability assumption plus Lemma 4.2 results in a tight confidence interval for the q-value”. Lemma 4.2 establishes that the targets of the least-squares regressions in optimization problem 1 are linearly realizable with the features (because skipping with $\bar G$ results in an approximately linear MDP). Concentrability is then used to bound the average confidence of the least-squares predictor.
>
> > One thing I do not follow is how the algorithm eliminates incorrect guesses $G$. If I understand correctly, Lemma 4.2 only shows that linear $q^\pi$-realizable MDPs with skips can be approximated by linear MDPs under the true guess. Then for an incorrect guess $\bar G$, is the modified MDPs still linear? If not, does optimization problem 1 still give some meaningful guarantees so that the algorithm can eliminate these cases? In addition, since the optimization problem 1 is mostly built upon prior except for condition (14), if could be better to emphasize this new condition.
>
> Regarding “how the algorithm eliminates incorrect guesses”: it doesn't (necessarily). For an incorrect guess the modified MDP may not be linear, as you point out. In a nutshell, there are two key guarantees for optimization problem 1. (1) that specifically $G=\bar G$ is not eliminated, and (2) that any $G$ that passes condition (14) leads to an accurate value estimation (even if the associated modified MDP isn't linear).
> To show guarantee (2), we use $q^\pi$-realizability only (no linear MDP properties) to prove that the true parameter $\psi_h(\pi^\star_G)$ realizing $q^{\pi^\star_G}$ for stage $h$ is included in $\Theta_{G, h}$ (Lemma C.2). Then the left hand side of condition (14) is exactly the confidence range for our estimator of $q^{\pi^\star_G}$. Condition (14) thus directly constrains these confidence ranges to be tight, leading to accurate estimators.
> Guarantee (1) gives legitimacy to including condition (14) in the optimization problem, by arguing that at least one choice ($G=\bar G$) leading to a near-optimal policy value is considered by optimization problem 1. Therefore, whatever $G$ the optimization problem ends up choosing can only have a larger or equal value estimation than this near-optimal value. This value estimation is accurate by guarantee (2), finishing the proof.
>
> We will follow your suggestion to better highlight condition (14), and we will make sure that the proof intuition above is clearly communicated in the paper.
>
> > In the statement of problem 1, n does not depend on \delta.
>
> Thanks for spotting the error in problem 1. We will revise it to $n = \text{poly}(1/\epsilon, H, d, C_\text{conc}, \log(1/\delta), \log(1/L_1), \log(1/L_2))$.

---

### Official Review · Reviewer_eyWj · 2024-07-12

**Soundness:** 3
**Presentation:** 2
**Contribution:** 3
**Rating:** 7
**Confidence:** 2

**Summary:**

This paper presents an important theoretical result in offline reinforcement learning (RL) with linear function approximation. The authors show that under the assumptions of linear q-realizability, concentrability, and access to full trajectory data, it is possible to efficiently learn an ε-optimal policy with a sample complexity that scales polynomially in the relevant problem parameters (horizon H, feature dimension d, concentrability coefficient Cconc) and inversely in the desired accuracy ε. This is in contrast to previous negative results showing exponential sample complexity lower bounds when only having access to individual transitions.

**Strengths:**

This is a strong theoretical contribution that significantly advances our understanding of the statistical complexity of offline RL under linear function approximation. The results are of high interest to the RL theory community.

The theoretical analysis is rigorous and the proofs seem correct. The assumptions are clearly stated and discussed. The authors also provide a thoughtful discussion of the limitations of their work, including the open question of computational efficiency and the restrictive nature of the linear q-realizability assumption.

**Weaknesses:**

It would be beneficial if the authors could add the concrete definition of the previous non-trajectory data to make a direct comparison with full length trajectory data (Assumtpion 2). Also, the authors could add some comments on the hardness results without the trajectory data after the added definition.

In Assumption 2, the notation $\phi(s_h^1, \cdot)$ is not clearly defined. It may lead to the meaning of $\phi(s_h^1, a)$ for all $a$.

Given the lengthy proof with complex notations and limited review time, the proof details are challenging to follow. Adding more discussion on algorithm design, including a pseudocode, and explaining the intuitions behind the proof would be helpful.

One potential limitation is the absence of experimental results, which means the practical relevance of the theoretical findings is not directly demonstrated. However, considering the focus on fundamental statistical limits, the lack of experiments is understandable.

**Questions:**

na

**Limitations:**

Limitations are discussed in Section 6.

---

> ### Author Rebuttal · Authors · 2024-08-07
>
> We thank the reviewer for their helpful feedback.
>
> > It would be beneficial if the authors could add the concrete definition of the previous non-trajectory data to make a direct comparison with full length trajectory data (Assumtpion 2). Also, the authors could add some comments on the hardness results without the trajectory data after the added definition.
>
> We will add the definition of non-trajectory data and better highlight the corresponding negative result and how it contrasts with our positive one.
>
> > In Assumption 2, the notation $\phi(s_{h}^{1}, \cdot)$ is not clearly defined. It may lead to the meaning of $\phi(s_{h}^{1}, a)$ for all $a$.
>
> The learner is actually given access to $\phi(s_h^1, a)$ for all $a \in \mathcal{A}$. Note that features corresponding to all actions are required for the optimization problem to be able to evaluate any choice of action. We will clarify the notation of Assumption 2 to reflect this.
>
> > Given the lengthy proof with complex notations and limited review time, the proof details are challenging to follow. Adding more discussion on algorithm design, including a pseudocode, and explaining the intuitions behind the proof would be helpful.
>
> Although the algorithm itself is conceptually simple (solving the optimization problem and outputting the policy defined in Eq. (15)), we agree that its presentation was fragmented. Currently, the algorithm definition is only mentioned in one sentence directly above Eq. (15), making it easy to miss. To enhance clarity, we will add an algorithm pseudocode block to subsection 4.4 ("Learner") that explicitly outlines the algorithm steps. We will also improve the presentation of the intuition; see our response to reviewer sDBe for more details on this.

---

> > ### Comment · Reviewer_eyWj · 2024-08-13
> >
> > Thank you for your response. I still believe this paper makes a valuable contribution to an important problem. Therefore, I have decide to maintain my positive score.

---

### Official Review · Reviewer_kP5e · 2024-08-02

**Soundness:** 3
**Presentation:** 2
**Contribution:** 3
**Rating:** 5
**Confidence:** 3

**Summary:**

This paper considers the problem of learning the value ($Q$) function under q^{\pi} realizability and concentration assumption. The major contribution is to use trajectory data instead of independent samples to learn the target function, where negative results have been proven with independent samples.

**Strengths:**

The problem is definite challenging, in particular given the negative results with independent samples.

**Weaknesses:**

The presentation is a problem of this work. The notation system is not reading friendly, and there is no algorithm block. I suggest to add an algorithm to make the input and output clear.

The high-level idea is not clear. The authors spend too much space on introducing their methods, and there is no explanation about the hard term in learning with independent terms. For example, why should we skip the states with small range(s)? It is hard for readers to verify the result through touching the high-level intuition.

Another concern is about the computational efficiency. Seems there is no evidence the optimization problem could be resolved efficiently.

**Questions:**

I do not have any further questions.

**Limitations:**

Yes.

---

> ### Author Rebuttal · Authors · 2024-08-07
>
> We thank the reviewer for their helpful feedback.
>
> > The presentation is a problem of this work. The notation system is not reading friendly, and there is no algorithm block. I suggest to add an algorithm to make the input and output clear.
>
> As in the camera ready we can have an additional page, assuming the paper gets accepted, we will be happy to add an algorithm (solving the optimization problem and outputting the policy defined in Eq. (15)) to subsection 4.4 ("Learner") with clear inputs and outputs, thanks for pointing this out.
>
> Regarding the notation, we thought a lot about how to present it in a clear and precise way, given the inherent complexity of dealing with many “modified MDPs” at once. We are keen to find ways to improve it further; could you please point to any specific notations that did not read well? Any further specific suggestions would be much appreciated.
>
> > The high-level idea is not clear.
>
> We will improve the presentation of the high-level ideas; see our response to reviewer sDBe for more details on this.
>
> > The authors spend too much space on introducing their methods, and there is no explanation about the hard term in learning with independent terms.
>
> We are having trouble understanding what you mean by "hard term in learning with independent terms". Could you clarify what you mean by this?
>
> If you mean why trajectory data is crucial for our method, this is discussed in Section 4.2: “it is because we have full length trajectories that we can transform the data available to simulate arbitrary length skipping mechanisms.” In other words, without trajectory data, it is not possible to transform the existing data to data that one would have collected had one worked with a “skipping MDP”.
>
> > For example, why should we skip the states with small range(s)?
>
> This is discussed in Section 4.3. We will improve this section to make sure the intuition that follows is clear. In a nutshell, if we could skip the states with small range(s), Lemma 4.2 shows the “modified MDP” would be approximately linear, transforming the problem to one we already know how to solve (e.g., with Eleanor [Zanette et al., 2020]). This would be great, but sadly it is hard to directly learn which states have small ranges. Instead, we use Lemma 4.2 as an analytical tool, as follows. We show that if we were to skip these states, the value function estimates in optimization problem 1 would be accurate. This guarantees that at least one near-optimal solution is considered by the optimizer. Without skipping, optimization problem 1 could optimize over the empty set, as it is possible that condition (14) would never be satisfied.
>
> > It is hard for readers to verify the result through touching the high-level intuition.
>
> We hope that including the above, as well as the high-level argument using guarantees (1) and (2) in our response to reviewer sDBe will significantly improve this shortcoming.
>
> > Another concern is about the computational efficiency. Seems there is no evidence the optimization problem could be resolved efficiently.
>
> The paper already acknowledges that computational efficiency is a concern. Through the skipping mechanism, our method inherently introduces complicated nonlinearities for value function estimations. It is an open question whether a computationally efficient solution to the problem considered in this paper exists at all. Our work only comments on statistical complexity, by discovering a polynomial-exponential complexity divide between trajectory data and individual transitions, which we found quite interesting. We think this contribution is of high interest and significance and thus warrants the paper to be published at NeurIPS even if we leave the question of efficient computation for future work.

---

> > ### Comment · Reviewer_kP5e · 2024-08-10
> >
> > Thanks for the response. I think I have the same concern as reviewer sDBe about how to eliminate the misleading value functions (which exists in the case with independent samples). It would be helpful if you can show how your methods address the hard instance in previous work [Foster et.al., 2021] and then summarize the high-level idea to generalize to other instances. Also it would be helpful to present some toy examples (e.g., tabular MDP) to show why it is necessary to skip some states.  Because I have not checked the results in detail, I would like to keep the score and reset my confidence as 2.

---

> > > ### Author Response · Authors · 2024-08-13
> > >
> > > > I think I have the same concern as reviewer sDBe about how to eliminate the misleading value functions (which exists in the case with independent samples)
> > >
> > > We gave a detailed explanation in response to reviewer sDBe about how the optimization does not necessarily eliminate all G that don’t lead to linear MDPs. Instead it guarantees (1) and (2), for which trajectory data is necessary. It would not be possible to show that guarantees (1) and (2) hold if we had general data of the form used in [Foster et al., 2021].
> > >
> > > > It would be helpful if you can show how your methods address the hard instance in previous work [Foster et.al., 2021] and then summarize the high-level idea to generalize to other instances.
> > >
> > > An important thing to note is that the lower bound constructions in [Foster et al., 2021] do not use trajectory data (otherwise our results would be a contradiction). Our method addresses the hard instance from [Foster et al., 2021] if the data is given as complete trajectories. The intuition for why our algorithm works if it has trajectory data has hopefully been addressed by our original comments to you and reviewer sDBe. Below we explain why the lower bound constructions break down if they need to use trajectory data, and why our algorithm breaks down if it doesn’t have trajectory data.
> > >
> > > The lower bound constructions in Theorems 1.1 and 1.2 of [Foster et al., 2021] were both made hard because the data collection distributions of individual transition tuples $(s, a, r, s’)$ were selected such that they reveal no (or almost no) information about the MDP instance. In both cases, receiving samples from the joint distribution of the entire trajectory makes the problem easy. In the case of Theorem 1.1, one would simply observe which states are reachable from the start state (the planted states). For Theorem 1.2, some information on whether any next-state $s’$ is planted or not would be leaking in each trajectory, in the form of being able to observe the next-state transition from exactly $s’$.
> > >
> > > A simpler example showing the root of the problem with individual transition data is as follows. Consider the toy problem of learning the value of some policy $\pi$ after taking action $a$ in state $s_1$ in a 2-stage MDP. The data is given as tuples of the form $(s_1, a, r_1^1, s_2^1, \dots, s_1, a, r_1^n, s_2^n)$  for the first stage and $(\bar s_2^1, a_2^1, r_2^1, \bar s_3^1, \dots, \bar s_2^n, a_2^n, r_2^n, \bar s_3^n)$ for the second stage. Notice there is no guarantee that $\bar s_2^j \sim P(s_1, a)$ with $j \in [n]$. We cannot infer what the rewards from the second-stage states distributed as $P(s_1, a)$ might look like from the data, making this problem hopelessly hard. In the extreme, the MDP might have infinitely many second-stage states, with the probability of any $s_2^j = \bar s_2^k$ (for any $j$ and $k$) being 0, highlighting that one cannot just “connect” and “importance weight” matching next-states $s_2^j$ of the first-stage transitions with matching start-states $\bar s_2^k$ of the second-stage transitions. In contrast, if we assume the data is such that $s_2^j = \bar s_2^j$ (notice this is exactly our “trajectory data”), this problem is immediately avoided as samples from $P(s_1, a)$, along with rewards from those states are directly handed to the learner. The learner can then simply use all of the rewards $r_2^j$ from tuples that contain the action $\pi(s_2^j)$ (which we have on average at least $~ 1/ C_\text{conc}$ of, due to concentrability) to estimate the value of policy $\pi$ after taking $s_1, a$ (solving the toy problem).
> > >
> > > Now consider our algorithm if we do not have trajectory data. In this case we are no longer able to construct least squares targets of the form needed to make use of Lemma 4.2 (discussed in Section 4.2 “The benefit of Trajectory Data”). This means that we would not be able to guarantee that our targets are linear, even under the true skipping mechanism $\bar G$, implying that $\bar G$ might not be a feasible solution to our optimization problem. Then our optimism argument that the output of the optimization problem has a value estimate at least as large as the value estimate based on $\bar G$ would no longer hold, causing our whole proof strategy to break down.
> > >
> > > We indicated to reviewer eyWj that we will add the definition of non-trajectory data (i.e individual transition tuples) and better highlight the hardness result. We will use our above explanation to better highlight the hardness result in the revised version of our paper.
> > >
> > > > Also it would be helpful to present some toy examples (e.g., tabular MDP) to show why it is necessary to skip some states.
> > >
> > > The work of [Weisz et al., 2023] originally presented this skipping mechanism. We believe that Figure 1 from their work does a good job of illustrating why skipping low-range states makes an MDP linear. We will add a similar example to our paper to make it more self contained.

---

### Decision · Program_Chairs · 2024-09-25

**Decision:**

Accept (poster)

**Comment:**

There is general consensus about the value of the results in the current submission for the theoretical RL community, hence my recommendation for acceptance. The paper relies on the recent findings of Weisz et al., who first noticed how trajectory data would make it possible to prove positive learning complexity results in online RL under less restrictive assumptions. This paper leverages those technical results to do a similar treatment in the offline RL case. While the technical contribution of the paper may not be extremely novel, showing how leveraging trajectory data (instead of simple transitions) unlock polynomial sample complexity is particularly important from a conceptual point of view. Unfortunately, as pointed out by the reviewers, there remain several limitations, in particular on the “practicality” of the current algorithmic approach, which is not computationally efficient. Nonetheless, I believe there is enough interest in the formal result to propose acceptance. I still encourage the authors to integrate their rebuttal into the final version of the paper, in particular in making algorithmic choices and technical arguments as clear and accessible as possible.